# Minimax Optimal and Computationally Efficient Algorithms for Distributionally Robust Offline Reinforcement Learning

**Zhishuai Liu**
Duke University
zhishuai.liu@duke.edu

**Pan Xu**
Duke University
pan.xu@duke.edu

## Abstract

Distributionally robust offline reinforcement learning (RL), which seeks robust policy training against environment perturbation by modeling dynamics uncertainty, calls for function approximations when facing large state-action spaces. However, the consideration of dynamics uncertainty introduces essential nonlinearity and computational burden, posing unique challenges for analyzing and practically employing function approximation. Focusing on a basic setting where the nominal model and perturbed models are linearly parameterized, we propose minimax optimal and computationally efficient algorithms realizing function approximation and initiate the study on instance-dependent suboptimality analysis in the context of robust offline RL. Our results uncover that function approximation in robust offline RL is essentially distinct from and probably harder than that in standard offline RL. Our algorithms and theoretical results crucially depend on a novel function approximation mechanism incorporating variance information, a new procedure of suboptimality and estimation uncertainty decomposition, a quantification of the robust value function shrinkage, and a meticulously designed family of hard instances, which might be of independent interest.

## 1 Introduction

Offline reinforcement learning (RL) [17, 18], which aims to learn an optimal policy achieving maximum expected cumulative reward from a pre-collected dataset, plays an important role in critical domains where online exploration is infeasible due to high cost or ethical issues, such as precision medicine [49, 11, 22, 21] and autonomous driving [32, 43]. The foundational assumption of offline RL [18, 15, 53] is that the offline dataset is collected from the same environment where learned policies are intended to be deployed. However, this assumption can be violated in practice due to temporal changes in dynamics. In such cases, standard offline RL could face catastrophic failures [10, 31, 64]. To address this issue, the robust offline RL [28, 30] focuses on robust policy training against the environment perturbation, which serves as a promising solution. Existing empirical successes of robust offline RL rely heavily on expressive function approximations [37, 36, 25, 45, 63, 16], as the omnipresence of applications featuring large state and action spaces necessitates powerful function representations to enhance generalization capability of decision-making in RL.

To theoretically understand robust offline RL with function approximation, the distributionally robust Markov decision process (DRMDP) [39, 30, 13] provides an established framework. In stark contrast to the standard MDP, DRMDP specifically tackles the *model uncertainty* by forming an uncertainty set around the nominal model, and takes a max-min formulation aiming to maximize the value function corresponding to a policy, uniformly across all perturbed models in the uncertainty set [55, 52, 57, 35, 41, 59, 40]. The core of DRMDPs lies in achieving an amenable combination of uncertainty set design and corresponding techniques to solve the inner optimization over the

uncertainty set. However, this consideration of model uncertainty introduces fundamental challenges to function approximation in terms of computational and statistical efficiency, particularly given the need to maximally exploit essential information in the offline dataset. For instance, in cases where the state and action spaces are large, the commonly used $(s, a)$-rectangular uncertainty set can make the inner optimization computationally intractable for function approximation [66]. Additionally, the distribution shifts, arising from the mismatch between the behavior policy and the target policy, as well as the mismatch between the nominal model and perturbed models, complicate the statistical analysis [41, 4]. Several recent works attempt to conquer these challenges. Panaganti et al. [35] studied the $(s, a)$-rectangularity, and their algorithm may suffer from the above mentioned computational issue. Additionally, the $(s, a)$-rectangular uncertainty set may contain transitions that would never happen in reality, and thus leads to conservative policies; Blanchet et al. [4] proposed a novel double pessimism principle, while their algorithm requires strong oracles, which is not practically implementable. Meanwhile, a line of works study function approximation in the online setting [44, 38, 51, 3, 20] or with a simulator [66], which are not applicable to offline RL. Thus, the following question arises:

*Is it possible to design a computationally efficient and minimax optimal algorithm for robust offline RL with function approximation?*

To answer the above question, we focus on a basic setting of *d-rectangular linear DRMDP*, where the nominal model is a standard linear MDP, and all perturbed models are parameterized in a linearly structured uncertainty set. We provide the first *instance-dependent* suboptimality analysis in the DRMDP literature with function approximation, which offers insights into the problem's intrinsic characteristics and challenges. Concretely, our contributions are summarized as follows.

- We propose a computationally efficient algorithm, Distributionally Robust Pessimistic Value Iteration (DRPVI), based on the pessimism principle [15, 53, 41] with a new *function approximation* mechanism explicitly devised for $d$-rectangular linear DRMDPs. We show that DRPVI achieves the following instance-dependent upper bound on the suboptimality gap:

$$\beta_1 \cdot \sup_{P \in \mathcal{U}^\rho(P^0)} \sum_{h=1}^H \mathbb{E}^{\pi^\star, P} \big[ \sum_{i=1}^d \|\phi_i(s_h, a_h)\mathbf{1}_i\|_{\mathbf{\Lambda}_h^{-1}} | s_1 = s \big]^1,$$

  This bound resembles those established in offline RL within standard linear MDPs [15, 62, 54]. However, there are two significant differences in our results. First, our bound depends on the *supremum* over the uncertainty set of transition kernels instead of one single transition kernel. Second, our result relies on a *diagonal-based normalization*, instead of the Mahalanobis norm of the feature vector, $\|\phi(s_h, a_h)\|_{\mathbf{\Lambda}_h^{-1}}$. See Table 1 for a clearer comparison. These two distinctions are unique to DRMDPs with function approximation, which we discuss in more details in Section 4. Moreover, our analysis provides a novel pipeline for studying instance-dependent upper bounds of computationally efficient algorithms under $d$-rectangular linear DRMDPs.
- We improve DRPVI by incorporating *variance information* into the new function approximation mechanism, resulting in the VA-DRPVI algorithm, which achieves a smaller upper bound:

$$\beta_2 \cdot \sup_{P \in \mathcal{U}^\rho(P^0)} \sum_{h=1}^H \mathbb{E}^{\pi^\star, P} \big[ \sum_{i=1}^d \|\phi_i(s_h, a_h)\mathbf{1}_i\|_{\mathbf{\Sigma}_h^{\star -1}} | s_1 = s \big]^2.$$

  This improves the result of DRPVI due to the fact that $\mathbf{\Sigma}_h^{\star -1} \preceq H^2 \mathbf{\Lambda}_h^{-1}$ by definition [60, 54]. Furthermore, when the uncertainty level $\rho = O(1)$, we show that the robust value function attains a *Range Shrinkage* property, leading to an improvement in the upper bound by an order of $H$. This explicit improvement is new in variance-aware algorithms, and is unique to DRMDPs.
- We further establish an *information-theoretic lower bound*. We prove that the upper bound of VA-DRPVI matches the information-theoretic lower bound up to $\beta_2$, which implies that VA-DRPVI is near-optimal in the sense of information theory. Importantly, both DRPVI and VA-DRPVI are computationally efficient and do not suffer from the high computational burden, as discussed above in settings with the $(s, a)$-rectangular uncertainty set, due to a decoupling property of the $d$-rectangular uncertainty set (see Remark 4.1 for more details). Thus, we confirm that, for robust offline RL with function approximation, both the *computational efficiency* and *minimax optimality* are achievable under the setting of $d$-rectangular linear DRMDPs.

---

[1]Here, $d$ is the feature dimension, $H$ is the horizon length, $\beta_1 = \tilde{O}(\sqrt{d}H)$ is a tunning parameter in DRPVI, $\mathcal{U}^\rho(P^0)$ is the uncertainty set with radius $\rho$, $\pi^\star$ is the optimal robust policy, $\phi(\cdot, \cdot) : \mathcal{S} \times \mathcal{A} \to \mathbb{R}^d$ is the instance-dependent feature vector, and $\mathbf{\Lambda}_h$ is the covariance matrix defined in (4.3).

[2]$\beta_2 = \tilde{O}(\sqrt{d})$ is a hyperparameter in VA-DRPVI; $\mathbf{\Sigma}_h^\star$ is the variance-weighted covariance matrix, see (5.5).

Table 1: Summary of instance-dependent results in offline RL with linear function approximation. $\mathbf{\Lambda}_h$ and $\mathbf{\Sigma}_h^\star$ are the empirical covariance matrix defined in (4.3) and (5.5) respectively. Note that $\pi^\star$ means the optimal policy in standard MDPs and the optimal robust policy in DRMDPs. The definition of $\mathbf{\Sigma}_h^\star$ also depends on the corresponding definition of $\pi^\star$.

| Algorithm | Setting | Instance-dependent upper bound on the suboptimality gap |
|---|---|---|
| PEVI [15] | MDP | $dH \cdot \sum_{h=1}^H \mathbb{E}^{\pi^\star, P}\big[\|\phi(s_h, a_h)\|_{\mathbf{\Lambda}_h^{-1}} \mid s_1 = s\big]$ |
| LinPEVI-ADV [54] | MDP | $\sqrt{d}H \cdot \sum_{h=1}^H \mathbb{E}^{\pi^\star, P}\big[\|\phi(s_h, a_h)\|_{\mathbf{\Lambda}_h^{-1}} \mid s_1 = s\big]$ |
| LinPEVI-ADV+ [54] | MDP | $\sqrt{d} \cdot \sum_{h=1}^H \mathbb{E}^{\pi^\star, P}\big[\|\phi(s_h, a_h)\|_{\mathbf{\Sigma}_h^{\star-1}} \mid s_1 = s\big]$ |
| DRPVI (ours) | DRMDP | $\sqrt{d}H \cdot \sup_{P \in \mathcal{U}^\rho(P^0)} \sum_{h=1}^H \mathbb{E}^{\pi^\star, P}\big[\sum_{i=1}^d \|\phi_i(s_h, a_h)\mathbf{1}_i\|_{\mathbf{\Lambda}_h^{-1}} \mid s_1 = s\big]$ |
| VA-DRPVI (ours) | DRMDP | $\sqrt{d} \cdot \sup_{P \in \mathcal{U}^\rho(P^0)} \sum_{h=1}^H \mathbb{E}^{\pi^\star, P}\big[\sum_{i=1}^d \|\phi_i(s_h, a_h)\mathbf{1}_i\|_{\mathbf{\Sigma}_h^{\star-1}} \mid s_1 = s\big]$ |

Our algorithm design and theoretical analysis draw inspiration from two crucial ideas proposed in standard linear MDPs: the reference-advantage decomposition [54] and the variance-weighted ridge regression [65]. However, the unique challenges in DRMDPs necessitate novel treatments that go far beyond a combination of existing techniques. Specifically, existing analysis of standard linear MDPs highly relies on the linear dependency of the Bellman equation on the (nominal) transition kernel. This linear dependency is disrupted by the consideration of model uncertainty, which induces essential *nonlinearity* that significantly complicates the statistical analysis of estimation error. To obtain our instance-dependent upper bounds, we establish a new theoretical analysis *pipeline*. This pipeline starts with a nontrivial decomposition of the suboptimality, and employs a new uncertainty decomposition that transforms the estimation uncertainty over all perturbed models to estimation uncertainty under the nominal model.

The information-theoretic lower bound in our paper is the first of its kind in the linear DRMDP setting, which could be of independent interest to the community. Previous lower bounds, which are based on the commonly used *Assouad's method* and established under the standard linear MDP, do not consider model uncertainty. In particular, one prerequisite for applying Assouad's method is switching the initial minimax objective to a minimax risk in terms of Hamming distance. The intertwining of this prerequisite with the nonlinearity induced by the model uncertainty makes the analysis significantly more challenging. To this end, we construct a novel family of *hard instances*, carefully designed to (1) mitigate the nonlinearity caused by the model uncertainty, (2) fulfil the prerequisite for Assouad's method, and (3) be concise enough to admit matrix analysis.

**Notations:** We denote $\Delta(\mathcal{S})$ as the set of probability measures over some set $\mathcal{S}$. For any number $H \in \mathbb{Z}_+$, we denote $[H]$ as the set of $\{1, 2, \cdots, H\}$. For any function $V : \mathcal{S} \to \mathbb{R}$, we denote $[\mathbb{P}_h V](s, a) = \mathbb{E}_{s' \sim P_h(\cdot|s,a)}[V(s')]$ as the expectation of $V$ with respect to the transition kernel $P_h$, $[\text{Var}_h V](s, a) = [\mathbb{P}_h V^2](s, a) - ([\mathbb{P}_h V](s, a))^2$ as the variance of $V$, $[\mathbb{V}_h V](s, a) = \max\{1, [\text{Var}_h V](s, a)\}$ as the truncated variance of $V$, and $[V(s)]_\alpha = \min\{V(s), \alpha\}$, given a scalar $\alpha > 0$, as the truncated value of $V$. For a vector $\boldsymbol{x}$, we denote $x_j$ as its $j$-th entry. And we denote $[x_i]_{i \in [d]}$ as a vector with the $i$-th entry being $x_i$. For a matrix $A$, denote $\lambda_i(A)$ as the $i$-th eigenvalue of $A$. For two matrices $A$ and $B$, we denote $A \preceq B$ as the fact that $B - A$ is a positive semidefinite matrix. For any function $f : \mathcal{S} \to \mathbb{R}$, we denote $\|f\|_\infty = \sup_{s \in \mathcal{S}} f(s)$. Given $P, Q \in \Delta(\mathcal{S})$, the total variation divergence of $P$ and $Q$ is defined as $D(P\|Q) = 1/2 \int_{\mathcal{S}} |P(s) - Q(s)| ds$.

## 2  Most Related Work

**DRMDPs.** The DRMDP framework has been extensively studied under different settings. The works of [55, 52, 61, 26, 12] assumed precise knowledge of the environment and formulated the DRMDP as classic planning problems. The works of [67, 57, 33, 56, 42, 58] assumed access to a generative model and studied the sample complexities of DRMDPs. The works of [35, 41, 4] studied the offline setting assuming access to only an offline dataset, and established sample complexities under data coverage or concentrability assumptions. The works of [51, 3, 8, 19, 20] studied the online setting where the agent can actively interact with the nominal environment to learn the robust policy.

**DRMDPs with linear function approximation.** Tamar et al. [44], Badrinath and Kalathil [3] proposed to use linear function approximation to solve DRMDPs with large state and action spaces and established asymptotic convergence guarantees. Zhou et al. [66] studied the natural Actor-Critic with function approximation, assuming access to a simulator. Their function approximation mechanisms depend on two novel uncertainty sets, one based on double sampling and the other on an integral probability metric. Ma et al. [24] first combined the linear MDP with the $d$-rectangular uncertainty set [12], and proposed the setting dubbed as the $d$-rectangular linear DRMDP, which naturally admits linear representations of the robust Q-functions[3]. Panaganti et al. [34] leverages the $d$-rectangular linear DRMDP framework to address the distribution shift problem in offline linear MDPs. Blanchet et al. [4] studied the offline $d$-rectangular linear DRMDP setting, for which the provable efficiency is established under a double pessimism principle. Liu and Xu [20] then studied the online $d$-rectangular linear DRMDP setting and pointed out that the intrinsic nonlinearity of DRMDPs might pose additional challenges for linear function approximation. After the release of our work, a concurrent study [48] emerged, which independently investigated offline DRMDPs with linear function approximation. Their algorithms attained the same instance-dependent suboptimalities as our proposed algorithms DRPVI and VA-DRPVI. Their algorithm DROP also achieved the same order of worst-case suboptimality, $\tilde{O}(dH^2/\sqrt{K})$, as our DRPVI. However, we further demonstrated that our algorithm VA-DRPVI can strictly improve this result to $\tilde{O}(dH\min\{1/\rho, H\}/\sqrt{K})$. Moreover, we introduced a novel hard instance and established the first information-theoretic lower bound for offline DRMDPs with linear function approximation. We also note that there is a line of works [4, 35] studied general function approximation under DRMDPs with the commonly studied $(s,a)$-rectangularity uncertainty sets, where no further structure is applied except the rectangularity.

## 3 Problem Formulation

In this section, we provide the preliminary of $d$-rectangular linear DRMDPs, and describe the dataset as well as the learning goal in offline reinforcement learning.

**Standard MDPs.** We start with the standard MDP, which constitutes the basic of DRMDPs. A finite horizon Markov decision process is denoted by $\text{MDP}(\mathcal{S}, \mathcal{A}, H, P, r)$, where $\mathcal{S}$ and $\mathcal{A}$ are the state and action spaces, $H \in \mathbb{Z}_+$ is the horizon length, $P = \{P_h\}_{h=1}^H$ denotes the set of probability transition kernels, $r = \{r_h\}_{h=1}^H$ denotes the reward functions. More specifically, for any $(h, s, a) \in [H] \times \mathcal{S} \times \mathcal{A}$, the transition kernel $P_h(\cdot|s, a)$ is a probability function over the state space $\mathcal{S}$, and the reward function $r_h : \mathcal{S} \times \mathcal{A} \to [0, 1]$ is assumed to be deterministic for simplicity. A sequence of deterministic policies is denoted as $\pi = \{\pi_h\}_{h=1}^H$, where $\pi_h : \mathcal{S} \to \mathcal{A}$ is the policy for step $h \in [H]$. Given any policy $\pi$ and transition $P$, for all $(s, a, h) \in \mathcal{S} \times \mathcal{A} \times [H]$, the corresponding value function $V_h^{\pi,P}(s) := \mathbb{E}^{\pi,P}\left[\sum_{t=h}^H r_t(s_t, a_t)\big|s_h = s\right]$ and Q-function $Q_h^{\pi,P}(s, a) := \mathbb{E}^{\pi,P}\left[\sum_{t=h}^H r_t(s_t, a_t)\big|s_h = s, a_h = a\right]$ characterize the expected cumulative rewards starting from step $h$, and both of them are bounded in $[0, H]$.

**Distributionally robust MDPs.** A finite horizon distributionally robust Markov decision process is denoted by $\text{DRMDP}(\mathcal{S}, \mathcal{A}, H, \mathcal{U}^\rho(P^0), r)$, where $P^0 = \{P_h^0\}_{h=1}^H$ is the set of nominal transition kernels, and $\mathcal{U}^\rho(P^0) = \bigotimes_{h\in[H]} \mathcal{U}_h^\rho(P_h^0)$ is the uncertainty set of transitions, where each $\mathcal{U}_h^\rho(P_h^0)$ is usually defined as a ball centered at $P^0$ with radius/uncertainty level $\rho \geq 0$ based on some probability divergence measures [13, 57, 56]. To account for the model uncertainty, the robust value function $V_h^{\pi,\rho}(s) := \inf_{P\in\mathcal{U}^\rho(P^0)} V_h^{\pi,P}(s), \forall(h,s) \in [H] \times \mathcal{S}$ is defined as the value function under the worst possible transition kernel within the uncertainty set $\mathcal{U}^\rho(P^0)$. Similarly, the robust Q-function is defined as $Q_h^{\pi,\rho}(s,a) = \inf_{P\in\mathcal{U}^\rho(P^0)} Q_h^{\pi,P}(s,a)$, for any $(h,s,a) \in [H] \times \mathcal{S} \times \mathcal{A}$. Further, we define the optimal robust value function and the optimal robust Q-function as

$$V_h^{\star,\rho}(s) = \sup_{\pi\in\Pi} V_h^{\pi,\rho}(s), \quad Q_h^{\star,\rho}(s,a) = \sup_{\pi\in\Pi} Q_h^{\pi,\rho}(s,a), \quad \forall(h,s,a) \in [H] \times \mathcal{S} \times \mathcal{A}.$$

---

[3]Ma et al. [24] study the offline $d$-rectangular linear DRMDPs with Kullback-Leibler (KL) uncertainty sets. We remark that 1) the proofs of their main lemmas (Lemma D.1 and Lemma D.2) related to suboptimality decomposition and the proof of theorems have technique flaws; 2) The formulation of their assumption 4.4 on the dual variable of the dual formulation of the KL-divergence is ambiguous and may be too strong to be realistic. Thus, the fundamental challenges of $d$-rectangular linear DRMDPs remain unresolved.

where $\Pi$ is the set of all policies. The optimal robust policy $\pi^\star = \{\pi_h^\star\}_{h=1}^H$ is defined as the policy that achieves the optimal robust value function: $\pi_h^\star(s) = \arg\sup_{\pi \in \Pi} V_h^{\pi,\rho}(s), \forall (h,s) \in [H] \times \mathcal{S}$.

$d$**-rectangular linear DRMDPs.** A $d$-rectangular linear DRMDP is a DRMDP where the nominal environment is a special case of linear MDP with a simplex feature space [14, Example 2.2] and the uncertainty set $\mathcal{U}_h^\rho(P_h^0)$ is defined based on the linear structure of the nominal transition kernel $P_h^0$. In particular, we make the following assumption about the nominal environment.

**Assumption 3.1.** Let $\phi : \mathcal{S} \times \mathcal{A} \to \mathbb{R}^d$ be a state-action feature mapping such that $\sum_{i=1}^d \phi_i(s,a) = 1$, $\phi_i(s,a) \geq 0$, for any $(i,s,a) \in [d] \times \mathcal{S} \times \mathcal{A}$. For any $(h,s,a) \in [H] \times \mathcal{S} \times \mathcal{A}$, the reward function and the nominal transition kernels have a linear representation: $r_h(s,a) = \langle \phi(s,a), \theta_h \rangle$, and $P_h^0(\cdot|s,a) = \langle \phi(s,a), \mu_h^0(\cdot) \rangle$, where $\|\theta_h\|_2 \leq \sqrt{d}$, and $\mu_h^0 = (\mu_{h,1}^0, \ldots, \mu_{h,d}^0)^\top$ are unknown probability measures over $\mathcal{S}$.

With notations in Assumption 3.1, we define the factor uncertainty sets as $\mathcal{U}_{h,i}^\rho(\mu_{h,i}^0) = \{\mu : \mu \in \Delta(\mathcal{S}), D(\mu\|\mu_{h,i}^0) \leq \rho\}, \forall (h,i) \in [H] \times [d]$, where $D(\cdot\|\cdot)$ is specified as the total variation (TV) divergence in this work. The uncertainty set is defined as $\mathcal{U}_h^\rho(P_h^0) = \bigotimes_{(s,a) \in \mathcal{S} \times \mathcal{A}} \mathcal{U}_h^\rho(s,a;\mu_h^0)$, where $\mathcal{U}_h^\rho(s,a;\mu_h^0) = \{\sum_{i=1}^d \phi_i(s,a)\mu_{h,i}(\cdot) : \mu_{h,i}(\cdot) \in \mathcal{U}_{h,i}^\rho(\mu_{h,i}^0), \forall i \in [d]\}$. A notable feature of this design is that the factor uncertainty sets $\{\mathcal{U}_{h,i}^\rho(\mu_{h,i}^0)\}_{h,i=1}^{H,d}$ are decoupled from the state-action pair $(s,a)$ and also independent with each other. As demonstrated later, this decoupling property results in a computationally efficient regime for function approximation.

**Robust Bellman equation.** Under the setting of $d$-rectangular linear DRMDPs, it is proved that the robust value function and the robust Q-function satisfy the robust Bellman equations [20]:

$$Q_h^{\pi,\rho}(s,a) = r_h(s,a) + \inf_{P_h(\cdot|s,a) \in \mathcal{U}_h^\rho(s,a;\mu_h^0)}[\mathbb{P}_h V_{h+1}^{\pi,\rho}](s,a), \tag{3.1a}$$

$$V_h^{\pi,\rho}(s) = \mathbb{E}_{a \sim \pi_h(\cdot|s)}[Q_h^{\pi,\rho}(s,a)], \tag{3.1b}$$

and the optimal robust policy $\pi^\star$ is deterministic. Thus, we can restrict the policy class $\Pi$ to the deterministic one. This leads to the robust Bellman optimality equations:

$$Q_h^{\star,\rho}(s,a) = r_h(s,a) + \inf_{P_h(\cdot|s,a) \in \mathcal{U}_h^\rho(s,a;\mu_h^0)}[\mathbb{P}_h V_{h+1}^{\star,\rho}](s,a), \tag{3.2a}$$

$$V_h^{\star,\rho}(s) = \max_{a \in \mathcal{A}} Q_h^\star(s,a). \tag{3.2b}$$

**Offline Dataset and the Learning Goal.** Let $\mathcal{D}$ denote an offline dataset consisting of $K$ i.i.d trajectories generated from the nominal environment $MDP(\mathcal{S}, \mathcal{A}, H, P^0, r)$ by a behavior policy $\pi^b = \{\pi_h^b\}_{h=1}^H$. In concrete, for each $\tau \in [K]$, the trajectory $\{(s_h^\tau, a_h^\tau, r_h^\tau)\}_{h=1}^H$ satisfies that $a_h^\tau \sim \pi_h^b(\cdot|s_h^\tau)$, $r_h^\tau = r_h(s_h^\tau, a_h^\tau)$, and $s_{h+1}^\tau \sim P_h^0(\cdot|s_h^\tau, a_h^\tau)$ for any $h \in [H]$. The goal of the robust offline RL is to learn the optimal robust policy $\pi^\star$ using the offline dataset $\mathcal{D}$. We define the suboptimality gap between any policy $\hat{\pi}$ and the optimal robust policy $\pi^\star$ as

$$\text{SubOpt}(\hat{\pi}, s_1, \rho) := V_1^{\star,\rho}(s_1) - V_1^{\hat{\pi},\rho}(s_1). \tag{3.3}$$

Then the goal of an algorithm in distributionally robust offline reinforcement learning is to learn a robust policy $\hat{\pi}$ that minimizes the suboptimality gap $\text{SubOpt}(\hat{\pi}, s, \rho)$, for any $s \in \mathcal{S}$.

## 4 Warmup: Robust Pessimistic Value Iteration

In this section, we first propose a simple algorithm in Algorithm 1 as a warm start, and provide an instance-dependent upper bound on its suboptimality gap in Theorem 4.4.

The optimal robust Bellman equation (3.2) implies that the optimal robust policy $\pi^\star$ is greedy with respect to the optimal robust Q-function. Therefore, it suffices to estimate $Q_h^{\star,\rho}$ to approximate $\pi^\star$. To this end, we estimate the optimal robust Q-function by iteratively performing an empirical version of the optimal robust Bellman equation similar to (3.2). In concrete, given the estimators at step $h+1$, denoted by $\widehat{Q}_{h+1}(s,a)$ and $\widehat{V}_{h+1}(s) = \max_{a \in \mathcal{A}} \widehat{Q}_{h+1}(s,a)$, Liu and Xu [20] show that applying one step backward induction similar to (3.2) leads to

$$Q_h(s,a) = r_h(s,a) + \inf_{P_h(\cdot|s,a) \in \mathcal{U}_h^\rho(s,a;\mu_h^0)}[\mathbb{P}_h \widehat{V}_{h+1}](s,a) = \langle \phi(s,a), \theta_h + \nu_h^\rho \rangle, \tag{4.1}$$

where $\nu_{h,i}^\rho := \max_{\alpha \in [0,H]} \{z_{h,i}(\alpha) - \rho(\alpha - \min_{s'}[\widehat{V}_{h+1}(s')]_\alpha)\}$, $z_{h,i}(\alpha) := \mathbb{E}^{\mu_{h,i}^0}[\widehat{V}_{h+1}(s')]_\alpha$, $\forall i \in [d]$, $[\widehat{V}_{h+1}(s')]_\alpha = \min\{\widehat{V}_{h+1}(s'), \alpha\}$, and $\alpha$ is a dual variable stemming from the dual formulation (see Proposition H.1). To estimate $Q_h(s,a)$, it suffices to estimate vectors $\boldsymbol{z}_h(\alpha) = [z_{h,1}(\alpha), \ldots, z_{h,d}(\alpha)]$ and $\boldsymbol{\nu}_h^\rho$ as follows.

- *Estimate $\boldsymbol{z}_h(\alpha)$:* note that $[\mathbb{P}_h^0[V_{h+1}]_\alpha](s,a) = \langle \boldsymbol{\phi}(s,a), \boldsymbol{z}_h(\alpha) \rangle$ by Assumption 3.1, where the expectation is taken with respect to the nominal kernel $P_h^0(\cdot|s,a)$. Given the estimator $\widehat{V}_{h+1}(s)$, it is natural to estimate $\boldsymbol{z}_h(\alpha)$ by solving the following ridge regression on the offline dataset $\mathcal{D}$.

$$\hat{\boldsymbol{z}}_h(\alpha) = \text{argmin}_{\boldsymbol{z} \in \mathbb{R}^d} \sum_{\tau=1}^K \left([\widehat{V}_{h+1}(s_{h+1}^\tau)]_\alpha - \boldsymbol{\phi}_h^{\tau\top} \boldsymbol{z}\right)^2 + \lambda \|\boldsymbol{z}\|_2^2$$
$$= \boldsymbol{\Lambda}_h^{-1} \left[\sum_{\tau=1}^K \boldsymbol{\phi}_h^\tau [\widehat{V}_{h+1}(s_{h+1}^\tau)]_\alpha\right], \tag{4.2}$$

where $\lambda > 0$, $\boldsymbol{\phi}_h^\tau$ is a shorthand notation for $\boldsymbol{\phi}(s_h^\tau, a_h^\tau)$, and $\boldsymbol{\Lambda}_h$ is the covariance matrix:

$$\boldsymbol{\Lambda}_h = \sum_{\tau=1}^K \boldsymbol{\phi}_h^\tau (\boldsymbol{\phi}_h^\tau)^\top + \lambda \mathbf{I}. \tag{4.3}$$

- *Estimate $\hat{\boldsymbol{\nu}}_h^\rho$:* based on $\hat{z}_{h,i}(\alpha)$, we can estimate $\hat{\nu}_{h,i}^\rho$ as follows.

$$\hat{\nu}_{h,i}^\rho = \max_{\alpha \in [0,H]} \{\hat{z}_{h,i}(\alpha) - \rho(\alpha - \min_{s'}[\widehat{V}_{h+1}^\rho(s')]_\alpha)\}, \forall i \in [d]. \tag{4.4}$$

After these two steps, we immediately obtain the estimated robust Q-function at step $h$,

$$\widehat{Q}_h(s,a) = \langle \boldsymbol{\phi}(s,a), \boldsymbol{\theta}_h + \hat{\boldsymbol{\nu}}_h^\rho \rangle. \tag{4.5}$$

Note that these estimations are constructed based on an offline dataset $\mathcal{D}$, which is known to cause distributional shift. We propose to incorporate a penalty term in the estimator (4.5) following the pessimism principle in the face of uncertainty [15, 53, 41].

---

**Algorithm 1** Distributionally Robust Pessimistic Value Iteration (DRPVI)

---

**Require:** Input dataset $\mathcal{D}$ and parameter $\beta_1$; $\widehat{V}_{H+1}^\rho(\cdot) = 0$.
1: **for** $h = H, \cdots, 1$ **do**
2:     $\boldsymbol{\Lambda}_h \leftarrow \sum_{\tau=1}^K \boldsymbol{\phi}_h^\tau \boldsymbol{\phi}_h^{\tau\top} + \lambda \boldsymbol{I}$
3:     **for** $i = 1, \cdots, d$ **do**
4:         Update $\hat{\nu}_{h,i}^\rho$ according to (4.4)
5:     **end for**
6:     $\Gamma_h(\cdot,\cdot) \leftarrow \beta_1 \sum_{i=1}^d \|\phi_i(\cdot,\cdot)\mathbf{1}_i\|_{\boldsymbol{\Lambda}_h^{-1}}$
7:     $\widehat{Q}_h^\rho(\cdot,\cdot) \leftarrow \{\boldsymbol{\phi}(\cdot,\cdot)^\top(\boldsymbol{\theta}_h + \hat{\boldsymbol{\nu}}_h^\rho) - \Gamma_h(\cdot,\cdot)\}_{[0,H-h+1]}$
8:     $\hat{\pi}_h(\cdot|\cdot) \leftarrow \text{argmax}_{\pi_h} \langle \widehat{Q}_h^\rho(\cdot,\cdot), \pi_h(\cdot|\cdot) \rangle_{\mathcal{A}}$, and $\widehat{V}_h^\rho(\cdot) \leftarrow \langle \widehat{Q}_h^\rho(\cdot,\cdot), \hat{\pi}_h(\cdot|\cdot) \rangle_{\mathcal{A}}$
9: **end for**

---

**Remark 4.1.** In Algorithm 1, the pessimism is achieved by subtracting a robust penalty term, $\sum_{i=1}^d \|\phi_i(\cdot,\cdot)\mathbf{1}_i\|_{\boldsymbol{\Lambda}_h^{-1}}$, from the robust Q-function estimation, which is derived from bounding the robust estimation uncertainty arising from $d$ ridge regressions. In particular, at step $h \in [H]$, denoting $\alpha_i^\star = \text{argmax}_{[0,H]} \{\hat{z}_{h,i}(\alpha) - \rho(\alpha - \min_{s'}[\widehat{V}_{h+1}^\rho(s')]_\alpha)\}, \forall i \in [d]$, we solve $d$ separate ridge regressions to obtain different coordinates of $\hat{\boldsymbol{\nu}}_h^\rho$. This design is tailored for the $d$-rectangular linear DRMDP, as we will see, leading to a distinct instance-dependent upper bound in Theorem 4.4.

**Remark 4.2.** Notably, to solve the optimization problem with respect to $\alpha \in [0,H]$ in (4.4), one will repeatedly invoke the closed form solution (4.2) for different values of $\alpha$. Moreover, the optimization is decoupled from the state-action pair, due to the decoupling property of $d$-rectangular uncertainty set. Similar algorithm designs have also appeared in [24] for Kullback-Leibler divergence based linear DRMDPs and in [20] for online linear DRMDPs. As for the computational tractability, we note that the minimization over $\alpha$ in (4.4) has been implemented in [20] using the *minimize* function in the *Nelder-Mead* method [29] in the Python module *scipy.optimize*. The minimization over the state space is avoided under a 'fail-state' assumption, common in applications such as robotics and healthcare (see Assumption 4.1 and Remark 4.2 in their paper). Without this assumption, we can also use the Nelder-Mead method to solve it. Thus, Algorithm 1 is in general computationally tractable.

Before presenting the theoretical guarantee of DRPVI, we make the following data coverage assumption, which is standard for offline linear MDPs [50, 9, 60, 54].

**Assumption 4.3.** We assume $\kappa := \min_{h \in [H]} \lambda_{\min}(\mathbb{E}^{\pi^b, P^0}[\phi(s_h, a_h)\phi(s_h, a_h)^\top]) > 0$ for the behavior policy $\pi^b$ and the nominal transition kernel $P^0$.

Assumption 4.3 requires the behavior policy to sufficiently explore the state-action space under the nominal environment. Indeed, it implicitly assumes that the nominal and perturbed environments share the same state-action space, and that the full information of this space is accessible through the nominal environment and the behavior policy $\pi^b$. Assumption 4.3 rules out cases where new states emerge in perturbed environments that can never be queried under the nominal environment as a result of the distribution shift. Now we present the theoretical guarantee for Algorithm 1.

**Theorem 4.4.** Under Assumptions 3.1 and 4.3, $\forall K > \max\{512 \log(2dH^2/\delta)/\kappa^2, 20449d^2H^2/\kappa\}$ and $\delta \in (0, 1)$, if we set $\lambda = 1$ and $\beta_1 = \tilde{O}(\sqrt{d}H)$ in Algorithm 1, then with probability at least $1 - \delta, \forall s \in \mathcal{S}$, the suboptimality of DRPVI satisfies

$$\text{SubOpt}(\hat{\pi}, s, \rho) \leq \beta_1 \cdot \sup_{P \in \mathcal{U}^\rho(P^0)} \sum_{h=1}^{H} \mathbb{E}^{\pi^\star, P}\Big[\sum_{i=1}^{d} \|\phi_i(s_h, a_h)\mathbf{1}_i\|_{\mathbf{\Lambda}_h^{-1}} \big| s_1 = s\Big], \qquad (4.6)$$

where $\mathbf{\Lambda}_h$ is the empirical covariance matrix defined in (4.3).

The result in Theorem 4.4 resembles existing instance-dependent bounds for standard linear MDPs [15, 54] (see Table 1 for a detailed comparison). However, there are two major distinctions between these results. First, our result depends on the weighted sum of diagonal elements $\sum_{i=1}^{d} \|\phi_i(s_h, a_h)\mathbf{1}_i\|_{\mathbf{\Lambda}_h^{-1}}$, dubbed as the *d-rectangular robust estimation error*, instead of the Mahalanobis norm of the feature vector $\|\phi(s_h, a_h)\|_{\mathbf{\Lambda}_h^{-1}}$. As discussed in Remark 4.1, this term primarily arises due to the necessity to solve $d$ distinct ridge regressions in each step, which presents a unique challenge in our analysis. Second, we consider the *supremum expectation* of $d$-rectangular robust estimation error with respect to all transition kernels in the uncertainty set, which measures the *worst case coverage* of the covariance matrix $\mathbf{\Lambda}_h$ under the optimal robust policy $\pi^\star$.

To connect with existing literature [4], we further show that under Assumption 4.3, the instance-dependent suboptimality bound can be simplified as follows.

**Corollary 4.5.** Under the same assumptions and settings as Theorem 4.4, with probability at least $1 - \delta$, for all $s \in \mathcal{S}$, the suboptimality of DRPVI satisfies $\text{SubOpt}(\hat{\pi}, s, \rho) = \tilde{O}(\sqrt{d}H^2/(\sqrt{\kappa \cdot K}))$.

**Remark 4.6.** Since $\|\phi(\cdot, \cdot)\|_2 \leq 1$ by Assumption 3.1, the coverage parameter $\kappa$ is trivially upper bounded by $1/d$. Assuming that $\kappa = c^\dagger/d$ for a constant $0 < c^\dagger < 1$, then we have $\text{SubOpt}(\hat{\pi}, s, \rho) = \tilde{O}(dH^2/(c^\dagger \cdot \sqrt{K}))$. This bound improves the state-of-the-art, [4, Theorem 6.3], by $O(d)$.

## 5  Distributionally Robust Variance-Aware Pessimistic Value Iteration

The instance-dependent bound in Theorem 4.4 has an explicit dependency on $H$, which arises from the fact that $Q_h^\rho(s, a) \in [0, H]$ for any $(h, \rho) \in [H] \times (0, 1]$ and the Hoeffding-type self-normalized concentration inequality used in our analysis. We will show in this section that the range of any robust value function could be much smaller under a refined analysis. Consequently, we can leverage variance information to improve Algorithm 1 and achieve a strengthened upper bound.

**Intuition** In the robust Bellman equation (3.1), the worst-case transition kernel would put as much mass as possible on the minimizer of $V_{h+1}^{\pi,\rho}(s)$, denoted by $s_{\min}$. Based on this observation, we conjecture that the robust Bellman equation (3.1) recursively reduces the maximal value of robust value functions, and thus shrinks its range. To see this, we define $\check{\mu}_{h,i} = (1-\rho)\mu_{h,i}^0 + \rho\delta_{s_{\min}}$, where $\delta_{s_{\min}}$ is the Dirac measure at $s_{\min}$, and we assume $V_{h+1}^{\pi,\rho}(s_{\min}) = 0$ for any $(\pi, h) \in \Pi \times [H]$ just for illustration. It is easy to verify that $\check{\mu}_{h,i} \in \mathcal{U}_{h,i}^\rho(\mu_{h,i}^0)$ and is indeed the worst-case factor kernel. Then by (3.1) we have $V_h^{\pi,\rho}(s) = \mathbb{E}_{a \sim \pi}[r_h(s, a) + (1-\rho)[\mathbb{P}_h^0 V_{h+1}^{\pi,\rho}](s, a)]$, which immediately implies $\max_{s \in \mathcal{S}} V_h^{\pi,\rho}(s) \leq 1 + (1-\rho)\max_{s' \in \mathcal{S}} V_{h+1}^{\pi,\rho}(s')$. This justifies our conjecture that the range of the robust value functions shrinks over stage. We dub this phenomenon as *Range Shrinkage* and summarize it in the following lemma, with a more formal proof postponed to Appendix G.5.

**Algorithm 2** Distributionally Robust and Variance Aware Pessimistic Value Iteration (VA-DRPVI)

**Require:** Input dataset $\mathcal{D}, \mathcal{D}'$ and $\beta_2$; $\widehat{V}_{H+1}^\rho(\cdot) = 0$

1: Run Algorithm 1 using dataset $\mathcal{D}'$ to get $\{\widehat{V}_h^{'\rho}\}_{h \in [H]}$
2: **for** $h = H, \cdots, 1$ **do**
3:     Construct variance estimator $\widehat{\sigma}_h^2(\cdot, \cdot; \alpha)$ using $\mathcal{D}'$ by (5.2) and (5.3)
4:     $\mathbf{\Sigma}_h(\alpha) = \sum_{\tau=1}^K \phi_h^\tau \phi_h^{\tau\top}/\widehat{\sigma}_h^2(s_h^\tau, a_h^\tau; \alpha) + \lambda \mathbf{I}$
5:     $\hat{z}_h(\alpha) = \mathbf{\Sigma}_h^{-1}(\alpha)\left( \sum_{\tau=1}^K \phi_h^\tau [\widehat{V}_{h+1}^\rho(s_{h+1}^\tau)]_\alpha / \widehat{\sigma}_h^2(s_h^\tau, a_h^\tau; \alpha) \right)$
6:     $\alpha_i = \mathrm{argmax}_{\alpha \in [0,H]}\{\hat{z}_{h,i}(\alpha) - \rho(\alpha - \min_{s'}[\widehat{V}_{h+1}^\rho(s')]_\alpha)\}, \ \forall i \in [d]$
7:     $\hat{\nu}_{h,i}^\rho = \hat{z}_{h,i}(\alpha_i) - \rho(\alpha_i - \min_{s'}[\widehat{V}_{h+1}^\rho(s')]_{\alpha_i}), \ \forall i \in [d]$
8:     $\Gamma_h(\cdot, \cdot) \leftarrow \beta_2 \sum_{i=1}^d \|\phi_i(\cdot, \cdot)\mathbf{1}_i\|_{\mathbf{\Sigma}_h^{-1}(\alpha_i)}$
9:     $\widehat{Q}_h^\rho(\cdot, \cdot) = \{\phi(\cdot, \cdot)^\top(\theta_h + \hat{\nu}_h^\rho) - \Gamma_h(\cdot, \cdot)\}_{[0, H-h+1]}$
10:    $\hat{\pi}_h(\cdot|\cdot) \leftarrow \mathrm{argmax}_{\pi_h}\langle \widehat{Q}_h^\rho(\cdot, \cdot), \pi_h(\cdot|\cdot)\rangle_\mathcal{A}, \ \widehat{V}_h^\rho(\cdot) \leftarrow \langle \widehat{Q}_h^\rho(\cdot, \cdot), \hat{\pi}_h(\cdot|\cdot)\rangle_\mathcal{A}$
11: **end for**

**Lemma 5.1** (Range Shrinkage). *For any $(\rho, \pi, h) \in (0, 1] \times \Pi \times [H]$, we have*

$$\max_{s \in \mathcal{S}} V_h^{\pi,\rho}(s) - \min_{s \in \mathcal{S}} V_h^{\pi,\rho}(s) \leq \frac{1 - (1-\rho)^{H-h+1}}{\rho}. \tag{5.1}$$

This phenomenon only appears in DRMDPs since the range of value function is generally $[0, H]$ in standard MDPs. A similar phenomenon is first observed in infinite horizon tabular DRMDPs [42, Lemma 7]. One important implication of Lemma 5.1 is that the conditional variance of any value function shrinks accordingly. In particular, when $\rho = O(1)$, the range of any robust value function would shrink to constant order, which leads to constant order conditional variances. This motivates us to leverage the variance information in both algorithm design and theoretical analysis. Inspired by the variance-weighted ridge regression in standard linear MDPs [65, 27, 60, 54], we propose to improve the vanilla ridge regression in (4.2) by incorporating variance weights. To this end, we first propose an appropriate variance estimator, whose form is specifically motivated by our theoretical analysis framework, to quantify the variance information.

**Variance estimation** We first run Algorithm 1 using an offline dataset $\mathcal{D}'$ that is independent of $\mathcal{D}$ to obtain estimators of the optimal robust value functions $\{\widehat{V}_h^{'\rho}\}_{h \in [H]}$. By Assumption 3.1, the variance of $[\widehat{V}_{h+1}^{'\rho}]_\alpha$ under the nominal environment is $[\mathrm{Var}_h[\widehat{V}_{h+1}^{'\rho}]_\alpha](s, a) = [\mathbb{P}_h^0[\widehat{V}_{h+1}^{'\rho}]_\alpha^2](s, a) - ([\mathbb{P}_h^0[\widehat{V}_{h+1}^{'\rho}]_\alpha](s, a))^2 = \langle\phi(s, a), z_{h,2}\rangle - (\langle\phi(s, a), z_{h,1}\rangle)^2$. We estimate $z_{h,1}$ and $z_{h,2}$ via ridge regression similarly as in (4.2):

$$\tilde{z}_{h,2}(\alpha) = \mathrm{argmin}_{z \in \mathbb{R}^d} \sum_{\tau=1}^K \left([\widehat{V}_{h+1}^{'\rho}(s_{h+1}^\tau)]_\alpha^2 - \phi_h^{\tau\top} z\right)^2 + \lambda\|z\|_2^2, \tag{5.2a}$$

$$\tilde{z}_{h,1}(\alpha) = \mathrm{argmin}_{z \in \mathbb{R}^d} \sum_{\tau=1}^K \left([\widehat{V}_{h+1}^{'\rho}(s_{h+1}^\tau)]_\alpha - \phi_h^{\tau\top} z\right)^2 + \lambda\|z\|_2^2. \tag{5.2b}$$

We then construct the following truncated variance estimator

$$\widehat{\sigma}_h^2(s, a; \alpha) := \max\left\{1, [\phi(s, a)^\top \tilde{z}_{h,2}(\alpha)]_{[0,H^2]} - [\phi(s, a)^\top \tilde{z}_{h,1}(\alpha)]_{[0,H]}^2 - \tilde{O}\left(\frac{dH^3}{\sqrt{K\kappa}}\right)\right\}, \tag{5.3}$$

where the last term is a penalty to achieve pessimistic estimations of conditional variances.

**Variance-Aware Function Approximation Mechanism** Similar to the two-step estimation procedure of Algorithm 1, we first estimate $z_h(\alpha)$ by the following variance-weighted ridge regression under the nominal environment:

$$\begin{aligned}\hat{z}_h(\alpha) &= \mathrm{argmin}_{z \in \mathbb{R}^d} \sum_{\tau=1}^K \frac{([\widehat{V}_{h+1}^\rho(s_{h+1}^\tau)]_\alpha - \phi_h^{\tau\top} z)^2}{\widehat{\sigma}_h^2(s_h^\tau, a_h^\tau; \alpha)} + \lambda\|z\|_2^2 \\ &= \mathbf{\Sigma}_h^{-1}(\alpha)\left[\sum_{\tau=1}^K \frac{\phi_h^\tau [\widehat{V}_{h+1}^\rho(s_{h+1}^\tau)]_\alpha}{\widehat{\sigma}_h^2(s_h^\tau, a_h^\tau; \alpha)}\right],\end{aligned} \tag{5.4}$$

where $\boldsymbol{\Sigma}_h(\alpha) = \sum_{\tau=1}^{K} \phi_h^\tau \phi_h^{\tau\top}/\widehat{\sigma}_h^2(s_h^\tau, a_h^\tau; \alpha) + \lambda \mathbf{I}$ is the empirical variance-weighted covariance matrix, which can be deemed as an estimator of the following variance-weighted covariance matrix

$$\boldsymbol{\Sigma}_h^\star = \sum_{\tau=1}^{K} \phi_h^\tau \phi_h^{\tau\top}/[\mathbb{V}_h V_{h+1}^{\star,\rho}](s_h^\tau, a_h^\tau) + \lambda \mathbf{I}. \tag{5.5}$$

In the second step, we estimate $\nu_{h,i}^\rho, \forall i \in [d]$ in the same way as (4.4). We then add a pessimism penalty based on $\boldsymbol{\Sigma}_h(\alpha)$. We present the full algorithm details in Algorithm 2.

**Theorem 5.2.** Under Assumptions 3.1 and 4.3, for $K > \max\{\tilde{O}(d^2 H^6/\kappa), \tilde{O}(H^4/\kappa^2)\}$ and $\delta \in (0,1)$, if we set $\lambda = 1/H^2$ and $\beta_2 = \tilde{O}(\sqrt{d})$ in Algorithm 2, then with probability at least $1 - \delta$, the suboptimality of VA-DRPVI satisfies

$$\text{SubOpt}(\hat{\pi}, s, \rho) \leq \beta_2 \cdot \sup_{P \in \mathcal{U}^\rho(P^0)} \sum_{h=1}^{H} \mathbb{E}^{\pi^\star, P} \left[ \sum_{i=1}^{d} \|\phi_i(s_h, a_h)\mathbf{1}_i\|_{\boldsymbol{\Sigma}_h^{\star-1}} \Big| s_1 = s \right], \tag{5.6}$$

where $\boldsymbol{\Sigma}_h^\star$ is the population variance-weighted covariance matrix defined as in (5.5).

Note that the bound in Theorem 5.2 does not explicitly depend on $H$ anymore compared with that in Theorem 4.4. A naive observation is that $[\mathbb{V}_h V_{h+1}^{\star,\rho}](s, a) \in [1, H^2]$. By comparing the definitions in (4.3) and (5.5), we have $\boldsymbol{\Sigma}_h^{\star-1} \preceq H^2 \boldsymbol{\Lambda}_h^{-1}$. Thus the upper bound of Algorithm 2 is never worse than that of Algorithm 1. This improvement brought by variance information is similar to that in standard linear MDPs [54, Theorem 2]. However, thanks to the range shrinkage phenomenon, we can further show that VA-DRPVI is strictly better than DRPVI when the uncertainty level is of constant order.

**Corollary 5.3.** Under the same assumptions and settings as Theorem 5.2, given the uncertainty level $\rho$, we have with probability at least $1 - \delta$, for all $s \in \mathcal{S}$, the suboptimality of VA-DRPVI satisfies

$$\text{SubOpt}(\hat{\pi}, s, \rho) \leq \beta_2 \cdot \frac{(1 - (1-\rho)^H)}{\rho} \cdot \sup_{P \in \mathcal{U}^\rho(P^0)} \sum_{h=1}^{H} \mathbb{E}^{\pi^\star, P} \left[ \sum_{i=1}^{d} \|\phi_i(s_h, a_h)\mathbf{1}_i\|_{\boldsymbol{\Lambda}_h^{-1}} \Big| s_1 = s \right].$$

**Remark 5.4.** Note that $(1 - (1-\rho)^H)/\rho = \boldsymbol{\Theta}(\min\{1/\rho, H\})$. When $\rho = O(1)$, the suboptimality of Algorithm 2 is strictly smaller than that of Algorithm 1 by $H$. With a similar argument as in Remark 4.6, if we assume there exist a constant $0 < c^\dagger < 1$, such that $\kappa = c^\dagger/d$ in Assumption 4.3, then the instance-dependent upper bound can be simplified to $\tilde{O}(dH \min\{1/\rho, H\}/(c^\dagger \cdot \sqrt{K}))$, which improves the state-of-the-art [4, Theorem 6.3] by $O(dH)$ when $\rho = O(1)$.

# 6 Information-Theoretic Lower Bound

For a matrix $\mathbf{A} \in \mathbb{R}^{d \times d}$ and a state $s \in \mathcal{S}$, we define function $\Phi(\cdot, \cdot) : \mathbb{R}^{d \times d} \times \mathcal{S} \to \mathbb{R}$ as follows.

$$\Phi(\mathbf{A}, s) = \sup_{P \in \mathcal{U}^\rho(P^0)} \sum_{h=1}^{H} \mathbb{E}^{\pi^\star, P} \left[ \sum_{i=1}^{d} \|\phi_i(s_h, a_h)\mathbf{1}_i\|_{\mathbf{A}} \Big| s_1 = s \right]. \tag{6.1}$$

It can be seen our upper bounds in previous sections primarily depend on quantities such as $\Phi(\boldsymbol{\Lambda}_h^{-1}, s)$ and $\Phi(\boldsymbol{\Sigma}_h^{\star-1}, s)$. Roughly speaking, these quantities characterize the discrepancy between the (weighted) covariance matrix of the offline dataset and the state action pairs generated from the transition probability in the uncertainty set. Hence we call $\Phi(\cdot, \cdot)$ the uncertainty function.

We now establish an information-theoretic lower bound to show that the uncertainty function is unavoidable for $d$-rectangular linear DRMDPs. Let $\mathcal{M}$ be a class of DRMDPs and we define $\text{SubOpt}(M, \hat{\pi}, s, \rho)$ as the suboptimality gap specific to one DRMDP instance $M \in \mathcal{M}$.

**Theorem 6.1.** Given uncertainty level $\rho \in (0, 3/4)$, dimension $d$, horizon length $H$ and sample size $K > \tilde{O}(d^6)$, there exists a class of $d$-rectangular linear DRMDPs $\mathcal{M}$ and an offline dataset $\mathcal{D}$ of size $K$ such that for all $s \in \mathcal{S}$, with probability at least $1 - \delta$, $\inf_{\hat{\pi}} \sup_{M \in \mathcal{M}} \text{SubOpt}(M, \hat{\pi}, s, \rho) \geq c \cdot \Phi(\boldsymbol{\Sigma}_h^{\star-1}, s)$, where $c$ is a universal constant.

Theorem 6.1 shows that the uncertainty function $\Phi(\boldsymbol{\Sigma}_h^{\star-1}, s)$ is intrinsic to the information-theoretic lower bound, and thus is inevitable. It is noteworthy that the lower bound in Theorem 6.1 aligns with the upper bound in Theorem 5.2 up to a factor of $\beta_2$, which implies that VA-DRPVI is minimax

optimal in the sense of information theory, but with a small gap of $\tilde{O}(\sqrt{d})$. Consequently, we affirm that, in the context of robust offline reinforcement learning with function approximation, both the computational efficiency and minimax optimality are achievable under the setting of $d$-rectangular linear DRMDPs with TV uncertainty sets. Moreover, Theorem 6.1 also suggests that achieving a good robust policy necessitates the worst case coverage of the offline dataset over the entire uncertainty set of transition models, which is significantly different from standard linear MDPs where a good coverage under the nominal model is enough [15, 60, 54]. Such a distinction indicates that learning in linear DRMDPs may be more challenging in comparison to standard linear MDPs.

Further, we highlight that the hard instances we constructed also satisfy Assumption 4.3. It remains an interesting direction to explore what would happen if the nominal and perturbed environments don't share exactly the same state-action space. We conjecture that since there could be absolutely new states emerging in perturbed environments that can never be explored in the nominal environment, the policy learned merely using data collected from the nominal environment could be arbitrarily bad.

**Challenges and novelties in construction of hard instances**  Existing tight lower bound analysis in standard linear MDPs [62, 60, 54] generally depends on the Assouad's method and a family of hard instances indexed by $\boldsymbol{\xi} \in \{-1, 1\}^{dH}$. However, they do not consider model uncertainty, which largely hinders the derivation of explicit formulas for the robust value functions. Further, one prerequisite of the Assouad's method is switching the initial minimax suboptimality $\inf_{\hat{\pi}} \max_{M \in \mathcal{M}} \text{SubOpt}(\hat{\pi}, s, \rho)$ to a risk of the form $\inf_{\boldsymbol{\xi}'} \max_{\boldsymbol{\xi}} D_H(\boldsymbol{\xi}, \boldsymbol{\xi}')$, where $D_H(\cdot, \cdot)$ is the Hamming distance. The model uncertainty significantly complicates this procedure, as the nonlinearity involved disrupts the linear dependency between the value function and the index $\boldsymbol{\xi}$. At the core of Theorem 6.1 is a novel class of hard instances $\mathcal{M}$. At a high-level, the hard instances should (1) fulfill the $d$-rectangular linear DRMDP conditions, (2) mitigate the nonlinearity caused by model uncertainty, (3) achieve the prerequisite for Assouad's method, and (4) be concise enough to admit matrix analysis. We postpone details on the construction of hard instances and the proof of Theorem 6.1 to Appendix F.

As a side product of Theorem 6.1, we show in the following corollary an information-theoretic lower bound in terms of the instance-dependent uncertainty function $\Phi(\boldsymbol{\Lambda}_h^{-1}, s)$ in Theorem 4.4.

**Corollary 6.2.** Under the same setting in Theorem 6.1, the class of hard instances $\mathcal{M}$ and offline dataset $\mathcal{D}$ in Theorem 6.1 also suggests that, with probability at least $1 - \delta$, $\inf_{\hat{\pi}} \sup_{M \in \mathcal{M}} \text{SubOpt}(\hat{\pi}, s, \rho) \geq c \cdot \Phi(\boldsymbol{\Lambda}_h^{-1}, s)$, where $c$ is a universal constant.

This implies that the uncertainty function $\Phi(\boldsymbol{\Lambda}_h^{-1}, s)$ in Theorem 4.4 also arises from the information-theoretic lower bound. We note the lower bound in Corollary 6.2 matches the upper bound in Theorem 4.4 up to $\beta_1$, thus DRPVI is also minimax optimal in the sense of information theory, but with a larger gap of $\tilde{O}(\sqrt{d}H)$. Moreover, the only difference between Theorem 6.1 and Corollary 6.2 is the covariance matrix. Due to the fact that $\boldsymbol{\Lambda}_h^{-1} \preceq \boldsymbol{\Sigma}_h^{\star, -1}$, the information-theoretic lower bound in Theorem 6.1 is indeed tighter than that in Corollary 6.2.

# 7  Conclusions

We studied robust offline RL with function approximation under the setting of $d$-rectangular linear DRMDPs with TV uncertainty sets. We first proposed the DRPVI algorithm and built up a theoretical analysis pipeline to establish the first instance-dependent upper bound on the suboptimality gap in the context of robust offline RL. We then showed an interesting range shrinkage phenomenon specific to DRMDPs, and we proposed the VA-DRPVI algorithm, which leverages the conditional variance information of the optimal robust value function. Based on the analysis pipeline built above, we show that the upper bound of VA-DRPVI achieves sharp dependence on the horizon length $H$. In addition, we found that an uncertainty function consisting of two crucial quantities–a supremum over uncertainty set and a diagonal-based normalization–appears in all upper bounds. We further established an information-theoretic lower bound to prove that the uncertainty function is unavoidable for robust offline RL under the setting of $d$-rectangular linear DRMDPs.

It remains an interesting future research question whether the computational and provable efficiency can be achieved in other settings for robust offline RL with function approximation. Another interesting future direction is to explore the unique challenges of applying general function approximation techniques in standard offline RL [6] to DRMDPs.

## Acknowledgments

We would like to thank the anonymous reviewers for their helpful comments. ZL and PX was supported in part by the National Science Foundation (DMS-2323112) and the Whitehead Scholars Program at the Duke University School of Medicine. The views and conclusions in this paper are those of the authors and should not be interpreted as representing any funding agency.

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

# A    Additional Related Work

**Offline Linear MDPs.**    Our work focuses on the offline linear MDP setting where the nominal transition kernel, from which the offline dataset is collected, admits the linear MDP structure. Numerous works have studied the provable efficiency and statistical limits of algorithms under this setting [15, 62, 53, 60, 54]. The most relevant study to ours is the recent work of [54], which established the minimax optimality of offline linear MDPs. At the core of their analysis is an advantage-reference technique designed for offline RL under linear function approximation, together with a variance aware pessimism-based algorithm. However, the offline linear MDP setting still remains understudied in the context of DRMDPs.

**Transfer-Learning in Low Rank MDPs.**    Besides the distributionally robust perspective to solve the planning problem in a nearly unknown target environment, another line of work focuses on transfer learning in low-rank MDPs [7, 23, 2, 5]. Specifically, the problem setup assumes that the agent has access to information of several source tasks. The agent learns a common representation from the source domains and then leverages the learned representation to learn a policy performing well in the target tasks with limited information. This setting is in stark contrast to DRMDPs, where the agent only has access to the information of a single source domain, without any available information of the target domain, assuming the same task is being performed. This motivates the pessimistic principle of the distributionally robust perspective. Among the aforementioned works, Bose et al. [5] studied the offline multi-task RL, which is the most closely related to our setting. In particular, they investigate the representation transfer error in their Theorem 1, stating that the learned representation can lead to a transition kernel that is close to the target kernel in terms of the TV divergence. Note that the uncertainty is induced by the representation estimation error, which is different from our setting assuming that the uncertainty comes from perturbations on underlying factor distributions. Nevertheless, this work provides evidence that TV divergence is a reasonable measure to quantify the uncertainty in transition kernels and motivates a future research direction in learning robust policies that are robust to the uncertainty induced by the representation estimation error.

# B    A More Computationally Efficient Variant of VA-DRPVI

In this section, we propose a modified version of Algorithm 2, which reduces the computation cost in the ridge regressions for variance estimation and achieves the same theoretical guarantees.

**Variance Estimator.**    In Section 5, we estimate the variance of the truncated robust value function $[\widehat{V}_{h+1}^{'\rho}]_\alpha$. Thus, for different $\alpha$, we need to establish different variance estimators, which significantly increases the computational burden. The theoretical analysis of Algorithm 2 suggests that it suffices to estimate the the variance of $\widehat{V}_{h+1}^{'\rho}$, instead of the truncated one. In particular, we know
$$[\text{Var}_h \widehat{V}_{h+1}^{'\rho}](s,a) = [\mathbb{P}_h^0 (\widehat{V}_{h+1}^{'\rho})^2](s,a) - ([\mathbb{P}_h^0 \widehat{V}_{h+1}^{'\rho}](s,a))^2 = \langle \phi(s,a), z_{h,2} \rangle - (\langle \phi(s,a), z_{h,1} \rangle)^2.$$
Then we estimate $z_{h,1}$ and $z_{h,2}$ via ridge regression:

$$\tilde{z}_{h,2} = \underset{z \in \mathbb{R}^d}{\text{argmin}} \sum_{\tau=1}^K \left( \left( \widehat{V}_{h+1}^{'\rho}(s_{h+1}^\tau) \right)^2 - \phi_h^{\tau\top} z \right)^2 + \lambda \|z\|_2^2, \tag{B.1a}$$

$$\tilde{z}_{h,1} = \underset{z \in \mathbb{R}^d}{\text{argmin}} \sum_{\tau=1}^K \left( \widehat{V}_{h+1}^{'\rho}(s_{h+1}^\tau) - \phi_h^{\tau\top} z \right)^2 + \lambda \|z\|_2^2. \tag{B.1b}$$

We construct the following truncated variance estimator:

$$\widehat{\sigma}_h^2(s,a) := \max \left\{ 1, \left[ \phi(s,a)^\top \tilde{z}_{h,2} \right]_{[0,H^2]} - \left[ \phi(s,a)^\top \tilde{z}_{h,1} \right]_{[0,H]}^2 - \tilde{O}\left( \frac{dH^3}{\sqrt{K}\kappa} \right) \right\}. \tag{B.2}$$

The modified variance-aware algorithm is presented in Algorithm 3 and the theoretical guarantee is presented in Theorem B.1.

**Theorem B.1.**   Under Assumptions 3.1 and 4.3, for $K > \max\{\tilde{O}(d^2 H^6/\kappa), \tilde{O}(H^4/\kappa^2)\}$ and $\delta \in (0,1)$, if we set $\lambda = 1/H^2$ and $\beta_2 = \tilde{O}(\sqrt{d})$ in Algorithm 3, then with probability at least $1 - \delta$, for

**Algorithm 3** Modified VA-DRPVI

---

**Require:** Input dataset $\mathcal{D}, \mathcal{D}'$ and $\beta_2$; $\widehat{V}_{H+1}^\rho(\cdot) = 0$
1: Run Algorithm 1 using dataset $\mathcal{D}'$ to get $\{\widehat{V}_h^{'\rho}\}_{h\in[H]}$
2: **for** $h = H, \cdots, 1$ **do**
3:     Construct variance estimator $\widehat{\sigma}_h^2(\cdot,\cdot)$ using $\mathcal{D}'$ by (B.1) and (B.2)
4:     $\boldsymbol{\Sigma}_h = \sum_{\tau=1}^K \boldsymbol{\phi}_h^\tau \boldsymbol{\phi}_h^{\tau\top}/\widehat{\sigma}_h^2(s_h^\tau, a_h^\tau) + \lambda\mathbf{I}$
5:     $\hat{\boldsymbol{z}}_h(\alpha) = \boldsymbol{\Sigma}_h^{-1}\Big( \sum_{\tau=1}^K \boldsymbol{\phi}_h^\tau \big[\widehat{V}_{h+1}^\rho(s_{h+1}^\tau)\big]_\alpha/\widehat{\sigma}_h^2(s_h^\tau, a_h^\tau) \Big)$
6:     $\alpha_i = \operatorname{argmax}_{\alpha\in[0,H]}\{\hat{z}_{h,i}(\alpha) - \rho(\alpha - \min_{s'}[\widehat{V}_{h+1}^\rho(s')]_\alpha)\}, \forall i \in [d]$
7:     $\hat{\nu}_{h,i}^\rho = \hat{z}_{h,i}(\alpha_i) - \rho(\alpha_i - \min_{s'}[\widehat{V}_{h+1}^\rho(s')]_{\alpha_i}), \forall i \in [d]$
8:     $\Gamma_h(\cdot,\cdot) \leftarrow \beta_2 \sum_{i=1}^d \|\phi_i(\cdot,\cdot)\mathbf{1}_i\|_{\boldsymbol{\Sigma}_h^{-1}}$
9:     $\widehat{Q}_h^\rho(\cdot,\cdot) = \{\boldsymbol{\phi}(\cdot,\cdot)^\top(\boldsymbol{\theta}_h + \hat{\boldsymbol{\nu}}_h^\rho) - \Gamma_h(\cdot,\cdot)\}_{[0,H-h+1]}$
10:    $\hat{\pi}_h(\cdot|\cdot) \leftarrow \operatorname{argmax}_{\pi_h}\langle\widehat{Q}_h^\rho(\cdot,\cdot), \pi_h(\cdot|\cdot)\rangle_{\mathcal{A}}, \widehat{V}_h^\rho(\cdot) \leftarrow \langle\widehat{Q}_h^\rho(\cdot,\cdot), \hat{\pi}_h(\cdot|\cdot)\rangle_{\mathcal{A}}$
11: **end for**

---

all $s \in \mathcal{S}$, the suboptimality of VA-DRPVI satisfies

$$\text{SubOpt}(\hat{\pi}, s, \rho) \le \beta_2 \cdot \sup_{P\in\mathcal{U}^\rho(P^0)} \sum_{h=1}^H \mathbb{E}^{\pi^\star, P}\Big[ \sum_{i=1}^d \|\phi_i(s_h, a_h)\mathbf{1}_i\|_{\boldsymbol{\Sigma}_h^{\star-1}}|s_1 = s\Big], \quad \text{(B.3)}$$

where $\boldsymbol{\Sigma}_h^\star = \sum_{\tau=1}^K \boldsymbol{\phi}_h^\tau \boldsymbol{\phi}_h^{\tau\top}/[\mathbb{V}_h V_{h+1}^\star](s_h^\tau, a_h^\tau) + \lambda\mathbf{I}$.

**Remark B.2.** The computation cost of Algorithm 3 is much smaller than Algorithm 2, as the variance estimators are not related to $\alpha$ anymore. Notably, Algorithm 3 shares the same upper bound as Algorithm 2. According to Theorem 6.1, we know the modified algorithm is also minimax optimal.

## C Experiments

We conduct numerical experiments to illustrate the performances of our proposed algorithms, DRPVI and VA-DRPVI, and compare it with the their non-robust counterpart, PEVI [15]. All numerical experiments were conducted on a MacBook Pro with a 2.6 GHz 6-Core Intel CPU. The implementation of our DRPVI algorithm is available at https://github.com/panxulab/Offline-Linear-DRMDP.

**Construction of the simulated linear MDP** We leverage the simulated linear MDP setting proposed by Liu and Xu [20] and modify it as an offline RL problem. In particular, the source and target linear MDP environment are shown in Figure 1(a) and Figure 1(b). The state space is set to be $\mathcal{S} = \{x_1, \cdots, x_5\}$ and the action space is to be $\mathcal{A} = \{-1, 1\}^4 \subset \mathbb{R}^4$. At each episode, the state always starts with $x_1$, and then transits to $x_2, x_4, x_5$ with probability defined in the figures. $x_2$ is an intermediate state, and it can transit to $x_3, x_4, x_5$ with probability defined on the lines. Moreover, Both $x_4$ and $x_5$ are absorbing states. $x_4$ ($x_5$) is the fail state (goal state), and the reward starting from which is always 0 (1). The reward functions and transition probabilities are designed to depend on the hyperparameter $\boldsymbol{\xi} \in \mathbb{R}^4$ as shown in the figure. The target environment is constructed by only perturbing the transition probability at $x_1$ of the source environment, and the extend of perturbation is controlled by the hyperparameter $q \in (0, 1)$. We refer more details on the construction of the simulated linear DRMDP to the Supplementary A.1 of [20].

**Implementation** We simply use the random policy that chooses actions uniformly at random at any $(s, a, h) \in \mathcal{S} \times \mathcal{A} \times [H]$ to collect offline dataset. The offline dataset containing 100 trajectories collected by the behavior policy from the source environment. We conduct ablation study by setting the hyperpameter $\boldsymbol{\xi} = (1/\|\boldsymbol{\xi}\|_1, 1/\|\boldsymbol{\xi}\|_1, 1/\|\boldsymbol{\xi}\|_1, 1/\|\boldsymbol{\xi}\|_1)^\top$ and consider different choices of $\|\boldsymbol{\xi}\|_1 \in \{0.1, 0.2, 0.3\}$. Following [20], we use heterogeneous uncertainty level for our two algorithms. Specifically, we set $\rho_{1,4} = 0.5$ and $\rho_{h,i} = 0$ for all other cases. The experiment results are shown in Figure 2.

Figure 2 shows the performances of the learned policies of three algorithms. We conclude that both of our proposed algorithms are robust to environmental perturbation compared to the non-robust PEVI.

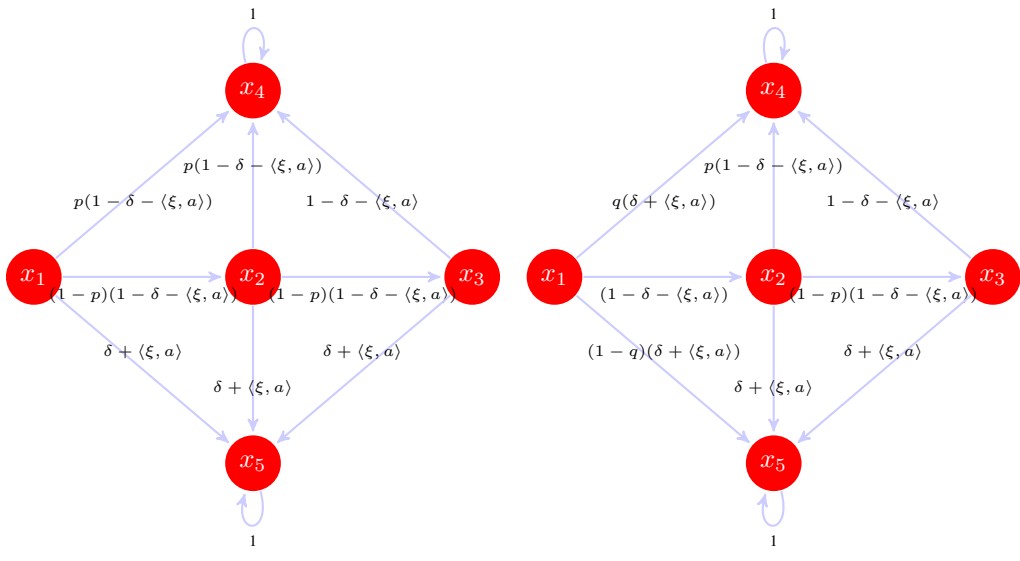

(a) The source MDP environment.  (b) The target MDP environment.

Figure 1: The source and the target linear MDP environments. The value on each arrow represents the transition probability. For the source MDP, there are five states and three steps, with the initial state being $x_1$, the fail state being $x_4$, and $x_5$ being an absorbing state with reward 1. The target MDP on the right is obtained by perturbing the transition probability at the first step of the source MDP, with others remaining the same.

Furthermore, VA-DRPVIslightly outperforms DRPVI in most settings. These numerical results are consistent with our theoretical findings.

# D  Proof of Theorem 4.4

Our analysis mainly deals with the challenges induced by the model uncertainty, $\inf_{P \in \mathcal{U}^\rho(P^0)}$, and the need to maximally exploit the information in the offline dataset. More specifically, the proof of Theorem 4.4 mainly constitutes of two steps.

**Step 1: suboptimality decomposition.**   We first decompose the suboptimality gap in the following lemma to connect it with the estimation error, the full proof of which can be found in Appendix G.1.

**Lemma D.1** (Suboptimality Decomposition for DRMDP). If the following holds

$$\left| \inf_{P_h(\cdot|s,a) \in \mathcal{U}_h^\rho(s,a;\boldsymbol{\mu}_{h,i}^0)} [\mathbb{P}_h \widehat{V}_{h+1}^\rho](s,a) - \boldsymbol{\phi}(s,a)\hat{\boldsymbol{\nu}}_h^\rho \right| \le \Gamma_h(s,a), \forall (s,a,h) \in \mathcal{S} \times \mathcal{A} \times [H], \quad \text{(D.1)}$$

then we have $\mathrm{SubOpt}(\hat{\pi}, s, \rho) \le 2 \sup_{P \in \mathcal{U}^\rho(P^0)} \sum_{h=1}^H \mathbb{E}^{\pi^\star, P} \big[ \Gamma_h(s_h, a_h) | s_1 = s \big].$

The main challenge in deriving Lemma D.1 lies in the dependency of the robust Bellman equation (3.1) on the nominal kernel $P^0$, which is not linear and does not even have an explicit form. It should be noted that the term $\big| \inf_{P_h(\cdot|s,a) \in \mathcal{U}_h^\rho(s,a;\boldsymbol{\mu}_{h,i}^0)} [\mathbb{P}_h \widehat{V}_{h+1}^\rho](s,a) - \boldsymbol{\phi}(s,a)^\top \hat{\boldsymbol{\nu}}_h^\rho \big|$ in condition (D.1) stands for the estimation error of the estimated robust Q-function in (4.5), which we refer to as the robust estimation uncertainty. Lemma D.1 shows that under the condition that the robust estimation uncertainty is bounded by $\Gamma_h(s,a)$, the suboptimality gap can be upper bounded in terms of $\Gamma_h(s,a)$. To conclude the proof, it remains to derive $\Gamma_h(s,a)$ and then substitute it back into the result in Lemma D.1.

**Step 2: bounding the robust estimation uncertainty.**   We now bound the robust estimation uncertainty in Lemma D.1 by the following result, the full proof of which can be found in Appendix G.2.

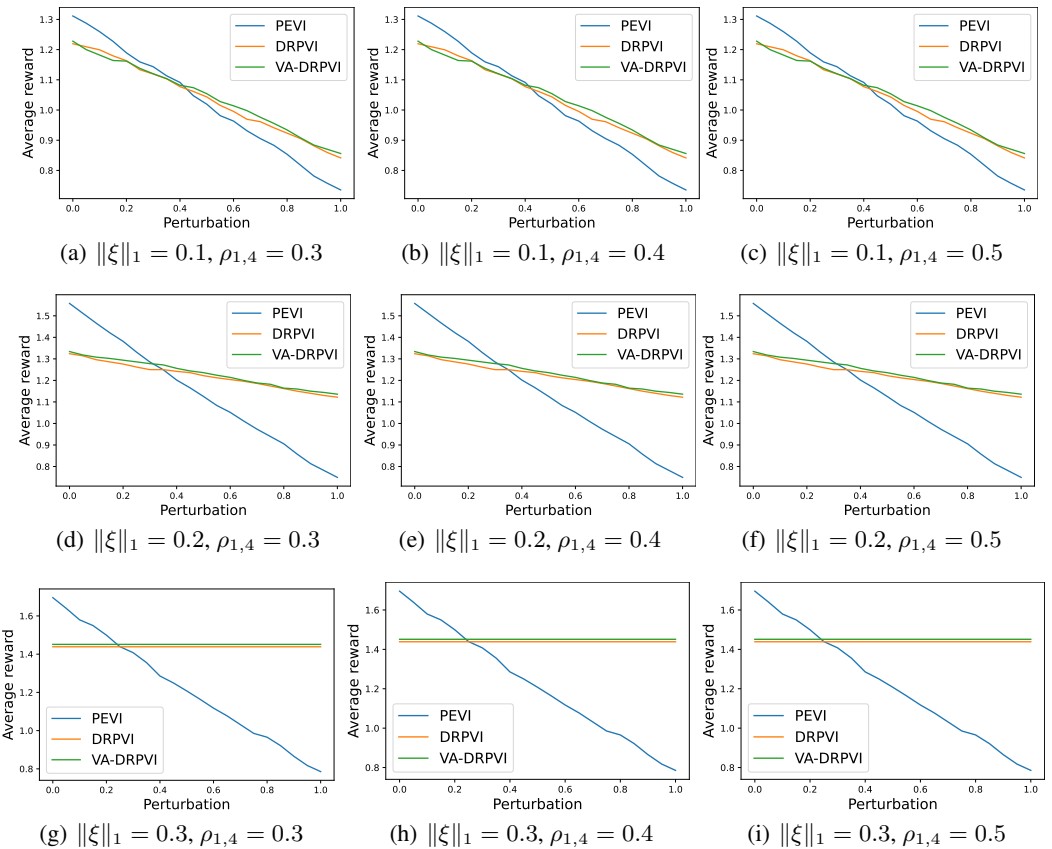

Figure 2: Simulation results under different source domains. The $x$-axis represents the perturbation level corresponding to different target environments. $\rho_{1,4}$ is the input uncertainty level for our VA-DRPVI algorithm. $\|\xi\|_1$ is the hyperparameter of the linear DRMDP environment.

**Lemma D.2** (Robust Estimation Uncertainty Bound). For any sufficiently large sample size $K$ satisfying $K > \max\{512\log(2dH^2/\delta)/\kappa^2, 20449d^2H^2/\kappa\}$, and any fixed $\delta \in (0, 1)$, if we set $\lambda = 1$ in Algorithm 1, then with probability at least $1 - \delta$, for all $(s, a, h) \in \mathcal{S} \times \mathcal{A} \times [H]$, we have

$$\left| \inf_{P_h(\cdot|s,a)\in\mathcal{U}_h^\rho(s,a;\boldsymbol{\mu}_{h,i}^0)} [\mathbb{P}_h \widehat{V}_{h+1}^\rho](s,a) - \boldsymbol{\phi}(s,a)^\top \hat{\boldsymbol{\nu}}_h^\rho \right| \leq \Gamma_h(s,a), \tag{D.2}$$

where $\Gamma_h(s,a) = 4\sqrt{d}H\sqrt{\iota}\sum_{i=1}^d \|\phi_i(s,a)\mathbf{1}_i\|_{\boldsymbol{\Lambda}_h^{-1}}$ and $\iota = \log(2dH^2K/\delta)$.

$\Gamma_h(s, a)$ provides an explicit bound for the robust estimation uncertainty, which also serves as the penalty term in Line 6 of Algorithm 1. The main challenge of deriving Lemma D.2 lies in inferring the worst-case behavior using information merely from the nominal environment. Our idea is to first transform the robust estimation uncertainty to the estimation uncertainty of ridge regressions (4.2) on the nominal model $P^0$, where the samples are collected and statistical control is available. We then adopt a reference-advantage decomposition technique, which is new in the linear DRMDP literature, to further decompose the estimation uncertainty on the nominal model into the reference uncertainty and the advantage uncertainty. The remaining proof is to bound the reference uncertainty and advantage uncertainty respectively using concentration and union bound arguments under an induction framework to address the temporal dependency. We highlight that all these arguments are specifically designed for the unique problem of DRMDP, which is novel and nontrivial.

# E   Proof of the Suboptimality Upper Bounds

In this section, we prove the main results in Corollary 4.5, Remark 4.6, Theorem 5.2, and Corollary 5.3, which give out the instance-dependent upper bounds of the proposed algorithms. Before the proof, we introduce some useful notations. For any function $f : \mathcal{S} \to [0, H-1]$, define

$$\widehat{\inf_{P_h(\cdot|s,a)\in\mathcal{U}_h^\rho(s,a;\boldsymbol{\mu}_{h,i}^0)}}[\mathbb{P}_h f](s,a) := \boldsymbol{\phi}(s,a)^\top \hat{\boldsymbol{\nu}}_h^\rho(f), \tag{E.1}$$

where for each $i \in [d]$, we have

$$\hat{\nu}_{h,i}^\rho(f) = \max_{\alpha\in[0,H]}\left\{\hat{\mathbb{E}}^{\mu_{h,i}^0}[f(s)]_\alpha - \rho(\alpha - \min_{s'\in\mathcal{S}}[f(s')]_\alpha)\right\},$$

$$\hat{\mathbb{E}}^{\mu_{h,i}^0}[f(s)]_\alpha = \left[\boldsymbol{\Lambda}_h^{-1}\sum_{\tau=1}^K\boldsymbol{\phi}_h^\tau[f(s_{h+1}^\tau)]_\alpha\right]_i.$$

## E.1   Proof of Corollary 4.5

The proof of Corollary 4.5 is straightforward given our result in Theorem 4.4.

*Proof.* Define $\tilde{\boldsymbol{\Lambda}}_h = \mathbb{E}^{\pi^b,P^0}[\boldsymbol{\phi}(s_h,a_h)\boldsymbol{\phi}(s_h,a_h)^\top], \forall h \in [H]$. By Assumption 4.3, we have $\tilde{\boldsymbol{\Lambda}}_h \succeq \kappa \cdot \mathbf{I}$. We further bound (6.1) as follows,

$$\sup_{P\in\mathcal{U}^\rho(P^0)}\sum_{h=1}^H\mathbb{E}^{\pi^\star,P}\left[\sum_{i=1}^d\|\phi_i(s_h,a_h)\mathbf{1}_i\|_{\boldsymbol{\Lambda}_h^{-1}}\Big|s_1=s\right]$$

$$\leq \sup_{P\in\mathcal{U}^\rho(P^0)}\frac{2}{\sqrt{K}}\mathbb{E}^{\pi^\star,P}\left[\sum_{h=1}^H\sum_{i=1}^d\|\phi_i(s_h,a_h)\mathbf{1}_i\|_{\tilde{\boldsymbol{\Lambda}}_h^{-1}}\Big|s_1=s\right] \tag{E.2}$$

$$= \sup_{P\in\mathcal{U}^\rho(P^0)}\frac{2}{\sqrt{K}}\mathbb{E}^{\pi^\star,P}\left[\sum_{h=1}^H\sum_{i=1}^d\phi_i(s,a)\sqrt{\mathbf{1}_i^\top\tilde{\boldsymbol{\Lambda}}_h^{-1}\mathbf{1}_i}\Big|s_1=s\right]$$

$$\leq \sup_{P\in\mathcal{U}^\rho(P^0)}\frac{2}{\sqrt{K}}\mathbb{E}^{\pi^\star,P}\left[\sum_{h=1}^H\sum_{i=1}^d\phi_i(s,a)\sqrt{\lambda_{\max}(\tilde{\boldsymbol{\Lambda}}_h^{-1})}\Big|s_1=s\right] \tag{E.3}$$

$$= \sup_{P\in\mathcal{U}^\rho(P^0)}\frac{2}{\sqrt{K}}\mathbb{E}^{\pi^\star,P}\left[\sum_{h=1}^H\sum_{i=1}^d\phi_i(s,a)\sqrt{\frac{1}{\lambda_{\min}(\tilde{\boldsymbol{\Lambda}}_h)}}\Big|s_1=s\right]$$

$$\leq \sup_{P\in\mathcal{U}^\rho(P^0)}\frac{2}{\sqrt{K}}\mathbb{E}^{\pi^\star,P}\left[\sum_{h=1}^H\sqrt{\frac{1}{\kappa}}\right] \tag{E.4}$$

$$= \frac{2H}{\sqrt{K\cdot\kappa}},$$

where (E.2) is due to Lemma I.3, (E.3) is due to the fact that for any matrix $\boldsymbol{A}$, $\lambda_{\min} \leq \boldsymbol{A}_{ii} \leq \lambda_{\max}$, where $\boldsymbol{A}_{ii}$ is the $i$-th diagonal element of $\boldsymbol{A}$. (E.3) holds due to Assumption 4.3 and the fact that $\sum_{i=1}^d\phi_i(s,a) = 1$. We conclude the proof by invoking Theorem 4.4. $\square$

## E.2   Proof of Theorem 5.2

The proof idea is similar to that of Theorem 4.4, except that we additionally analyze the variance estimation and apply the Bernstein-type self-normalized concentration inequality to bound the reference uncertainty, which is the dominant term. We start from analyzing the estimation error of conditional variances in the following lemma.

**Lemma E.1.** Under Assumptions 3.1 and 4.3, when $K \geq \tilde{O}(H^4/\kappa^2)$, then with probability at least $1-\delta$, for all $(s,a,h) \in \mathcal{S} \times \mathcal{A} \times [H]$ and any fixed $\alpha$, we have

$$\left[\mathbb{V}_h[V_{h+1}^{\star,\rho}]_\alpha\right](s,a) - \tilde{O}\left(\frac{dH^3}{\sqrt{K\kappa}}\right) \leq \hat{\sigma}_h^2(s,a;\alpha) \leq \left[\mathbb{V}_h[V_{h+1}^{\star,\rho}]_\alpha\right](s,a).$$

The following lemma bounds the estimation error by reference-advantage decomposition.

**Lemma E.2** (Variance-Aware Reference-Advantage Decomposition). There exist $\{\alpha_i\}_{i\in[d]}$, where $\alpha_i \in [0, H], \forall i \in [d]$, such that

$$
\Bigg| \inf_{P_h(\cdot|s,a)\in\mathcal{U}_h^\rho(s,a;\boldsymbol{\mu}_{h,i}^0)} [\mathbb{P}_h \widehat{V}_{h+1}^\rho](s,a) - \widehat{\inf_{P_h(\cdot|s,a)\in\mathcal{U}_h^\rho(s,a;\boldsymbol{\mu}_{h,i}^0)}} [\mathbb{P}_h \widehat{V}_{h+1}^\rho](s,a) \Bigg|
$$

$$
\leq \lambda \underbrace{\sum_{i=1}^d \|\phi_i(s,a)\mathbf{1}_i\|_{\boldsymbol{\Sigma}_h^{-1}(\alpha_i)} \|\mathbb{E}^{\boldsymbol{\mu}_h^0}[V_{h+1}^{\star,\rho}(s)]_{\alpha_i}\|_{\boldsymbol{\Sigma}_h^{-1}(\alpha_i)}}_{\text{i}}
$$

$$
+ \underbrace{\sum_{i=1}^d \|\phi_i(s,a)\mathbf{1}_i\|_{\boldsymbol{\Sigma}_h^{-1}(\alpha_i)} \Bigg\| \sum_{\tau=1}^K \frac{\phi_h^\tau \eta_h^\tau([V_{h+1}^{\star,\rho}]_{\alpha_i})}{\widehat{\sigma}_h^2(s_h^\tau, a_h^\tau; \alpha_i)} \Bigg\|_{\boldsymbol{\Sigma}_h^{-1}(\alpha_i)}}_{\text{ii}}
$$

$$
+ \lambda \underbrace{\sum_{i=1}^d \|\phi_i(s,a)\mathbf{1}_i\|_{\boldsymbol{\Sigma}_h^{-1}(\alpha_i)} \Big\| \mathbb{E}^{\boldsymbol{\mu}_h^0}\big[[\widehat{V}_{h+1}^\rho(s)]_{\alpha_i} - [V_{h+1}^{\star,\rho}(s)]_{\alpha_i}\big] \Big\|_{\boldsymbol{\Sigma}_h^{-1}(\alpha_i)}}_{\text{iii}}
$$

$$
+ \underbrace{\sum_{i=1}^d \|\phi_i(s,a)\mathbf{1}_i\|_{\boldsymbol{\Sigma}_h^{-1}(\alpha_i)} \Bigg\| \sum_{\tau=1}^K \frac{\phi_h^\tau \eta_h^\tau([\widehat{V}_{h+1}^\rho(s)]_{\alpha_i} - [V_{h+1}^{\star,\rho}(s)]_{\alpha_i})}{\widehat{\sigma}_h^2(s_h^\tau, a_h^\tau; \alpha_i)} \Bigg\|_{\boldsymbol{\Sigma}_h^{-1}(\alpha_i)}}_{\text{iv}},
$$

where $\eta_h^\tau([f]_{\alpha_i}) = \big([\mathbb{P}_h^0[f]_{\alpha_i}](s_h^\tau, a_h^\tau) - [f(s_{h+1}^\tau)]_{\alpha_i}\big)$, for any function $f : \mathcal{S} \to [0, H-1]$.

Now we are ready to prove Theorem 5.2

*Proof of Theorem 5.2.* To prove this theorem, we bound the estimation error by $\Gamma_h(s, a)$, then invoke Lemma D.1 to get the result. First, we bound terms i-iv in Lemma E.2 to deduce $\Gamma_h(s, a)$ at each step $h \in [H]$, respectively.

**Bound i and iii:** We set $\lambda = 1/H^2$ to ensure that for all $(s, a, h) \in \mathcal{S} \times \mathcal{A} \times [H]$, we have

$$
\text{i} + \text{iii} \leq \sqrt{\lambda}\sqrt{d}H \sum_{i=1}^d \|\phi_i(s,a)\mathbf{1}_i\|_{\boldsymbol{\Sigma}_h^{-1}(\alpha_i)} = \sqrt{d} \sum_{i=1}^d \|\phi_i(s,a)\mathbf{1}_i\|_{\boldsymbol{\Sigma}_h^{-1}(\alpha_i)}. \tag{E.5}
$$

**Bound ii:** For all $(s, a, \alpha) \in \mathcal{S} \times \mathcal{A} \times [0, H]$, by definition we have $\widehat{\sigma}_h(s, a; \alpha) \geq 1$. Thus, for all $(h, \tau, i) \in [H] \times [K] \times [d]$, we have $|\eta_h^\tau([V_{h+1}^{\star,\rho}]_{\alpha_i})/\widehat{\sigma}_h(s_h^\tau, a_h^\tau, \alpha_i)| \leq H$. Note that $V_{H+1}^{\star,\rho}$ is independent of $\mathcal{D}$, we can directly apply Bernstein-type self-normalized concentration inequality Lemma I.2 and a union bound to obtain the upper bound. In concrete, we define the filtration $\mathcal{F}_{\tau-1,h} = \sigma(\{(s_h^j, a_h^j)\}_{j=1}^\tau \cup \{s_{h+1}^j\}_{j=1}^{\tau-1})$. Since $V_{h+1}^{\star,\rho}$ and $\widehat{\sigma}_h(s, a; \alpha)$ are independent of $\mathcal{D}$, thus $\eta_h^\tau([V_{h+1}^{\star,\rho}]_{\alpha_i})/\widehat{\sigma}_h(s_h^\tau, a_h^\tau, \alpha_i)$ is mean-zero conditioned on the filtration $\mathcal{F}_{\tau-1,h}$. Further, we have

$$
\mathbb{E}\Bigg[ \Big( \frac{\eta_h^\tau([V_{h+1}^{\star,\rho}]_{\alpha_i})}{\widehat{\sigma}_h(s_h^\tau, a_h^\tau; \alpha_i)} \Big)^2 \Big| \mathcal{F}_{\tau-1,h} \Bigg] = \frac{[\text{Var}[V_{h+1}^{\star,\rho}]_{\alpha_i}](s_h^\tau, a_h^\tau)}{\widehat{\sigma}_h^2(s_h^\tau, a_h^\tau; \alpha_i)} \tag{E.6}
$$

$$
\leq \frac{[\mathbb{V}[V_{h+1}^{\star,\rho}]_{\alpha_i}](s_h^\tau, a_h^\tau)}{\widehat{\sigma}_h^2(s_h^\tau, a_h^\tau; \alpha_i)}
$$

$$
= \frac{[\mathbb{V}[V_{h+1}^{\star,\rho}]_{\alpha_i}](s_h^\tau, a_h^\tau) - \tilde{O}(dH^3/\sqrt{K\kappa})}{\widehat{\sigma}_h^2(s_h^\tau, a_h^\tau; \alpha_i)} + \frac{\tilde{O}(dH^3/\sqrt{K\kappa})}{\widehat{\sigma}_h^2(s_h^\tau, a_h^\tau; \alpha_i)}
$$

$$
\leq 1 + \frac{\tilde{O}(dH^3/\sqrt{K\kappa})}{\widehat{\sigma}_h^2(s_h^\tau, a_h^\tau; \alpha_i) - \tilde{O}(dH^3/\sqrt{K\kappa})} \tag{E.7}
$$

$$
\leq 1 + 2\tilde{O}\Big( \frac{dH^3}{\sqrt{K\kappa}} \Big), \tag{E.8}
$$

where (E.6) holds by the fact that $\widehat{\sigma}_h^2(\cdot, \cdot; \cdot)$ is independent of $\mathcal{D}$ and $(s_h^\tau, a_h^\tau)$ is $\mathcal{F}_{\tau-1,h}$ measurable. (E.7) holds by Lemma E.1, and (E.8) holds by setting $K \geq \tilde{\Omega}(d^2 H^6/\kappa)$ such that $\widehat{\sigma}_h^2(s_h^\tau, a_h^\tau; \alpha_i) - \tilde{O}(dH^3/\sqrt{K\kappa}) \geq 1 - \tilde{O}(dH^3/\sqrt{K\kappa}) \geq 1/2$. Further, by (E.8), our choice of $K$ also ensures that $\mathbb{E}\left[\left(\eta_h^\tau([V_{h+1}^{\star,\rho}]_{\alpha_i})\right)^2 | \mathcal{F}_{\tau-1,h}\right] = O(1)$. Then by Lemma I.2, we have

$$\left\| \sum_{\tau=1}^K \frac{\phi_h^\tau \eta_h^\tau([V_{h+1}^{\star,\rho}]_{\alpha_i})}{\widehat{\sigma}_h^2(s_h^\tau, a_h^\tau; \alpha_i)} \right\|_{\boldsymbol{\Sigma}_h^{-1}(\alpha_i)} \leq \tilde{O}(\sqrt{d}).$$

This implies

$$\text{ii} \leq \tilde{O}(\sqrt{d}) \sum_{i=1}^d \|\phi_i(s,a)\mathbf{1}_i\|_{\boldsymbol{\Sigma}_h^{-1}(\alpha_i)}. \tag{E.9}$$

**Bound iv:** Following the same induction analysis procedure, we have $\|[\widehat{V}_{h+1}^\rho]_{\alpha_i} - [V_{h+1}^{\star,\rho}]_{\alpha_i}\| \leq \tilde{O}(\sqrt{d}H^2/\sqrt{K\kappa})$. Then, using standard $\epsilon$-covering number argument and Lemma I.1, we have

$$\text{iv} \leq \tilde{O}\left(\frac{d^{3/2}H^2}{\sqrt{K\kappa}}\right) \sum_{i=1}^d \|\phi_i(s,a)\mathbf{1}_i\|_{\boldsymbol{\Sigma}_h^{-1}(\alpha_i)}. \tag{E.10}$$

To make it non-dominant, we require $K \geq \tilde{\Omega}(d^2 H^4/\kappa)$. By Lemma E.1, for any $\alpha \in [0, H]$, we have

$$\widehat{\sigma}_h^2(s_h^\tau, a_h^\tau; \alpha) \leq [\mathbb{V}_h[V_{h+1}^{\star,\rho}]_\alpha](s_h^\tau, a_h^\tau) \leq [\mathbb{V}_h V_{h+1}^{\star,\rho}](s_h^\tau, a_h^\tau),$$

this implies that

$$\left( \sum_{\tau=1}^K \frac{\phi_h^\tau \phi_h^{\tau\top}}{\widehat{\sigma}_h^2(s_h^\tau, a_h^\tau; \alpha_i)} + \lambda \mathbf{I} \right)^{-1} \preceq \left( \sum_{\tau=1}^K \frac{\phi_h^\tau \phi_h^{\tau\top}}{[\mathbb{V}_h V_{h+1}^{\star,\rho}](s_h^\tau, a_h^\tau)} + \lambda \mathbf{I} \right)^{-1} := \boldsymbol{\Sigma}_h^{\star-1}.$$

Combining (E.5), (E.9) and (E.10), we have

$$\left| \inf_{P_h(\cdot|s,a) \in \mathcal{U}_h^\rho(s,a;\boldsymbol{\mu}_{h,i}^0)} [\mathbb{P}_h \widehat{V}_{h+1}^\rho](s,a) - \widehat{\inf_{P_h(\cdot|s,a) \in \mathcal{U}_h^\rho(s,a;\boldsymbol{\mu}_{h,i}^0)}} [\mathbb{P}_h \widehat{V}_{h+1}^\rho](s,a) \right|$$

$$\leq \tilde{O}(\sqrt{d}) \sum_{i=1}^d \|\phi_i(s,a)\mathbf{1}_i\|_{\boldsymbol{\Sigma}_h^{\star-1}}.$$

Define $\Gamma_h(s,a) = \tilde{O}(\sqrt{d}) \sum_{i=1}^d \|\phi_i(s,a)\mathbf{1}_i\|_{\boldsymbol{\Sigma}_h^{\star-1}}$, we concludes the proof by invoking Lemma D.1.
$\square$

### E.3 Proof of Corollary 5.3

In this section, we prove Corollary 5.3. We start with an interesting phenomenon, we call 'range shrinkage', stated in the following lemma.

**Lemma E.3** (Range Shrinkage). For any $(\rho, \pi, h) \in (0,1] \times \Pi \times [H]$, we have

$$\max_{s \in \mathcal{S}} V_h^{\pi,\rho}(s) - \min_{s \in \mathcal{S}} V_h^{\pi,\rho}(s) \leq \frac{1 - (1-\rho)^{H-h+1}}{\rho}. \tag{E.11}$$

*Proof of Corollary 5.3.* By the fact that the variance of a random variable can be upper bounded by the square of its range and Lemma E.3, for all $(s, a, h) \in \mathcal{S} \times \mathcal{A} \times [H]$, we have

$$[\mathbb{V} V_{h+1}^\star](s,a) \leq \left( \frac{1 - (1-\rho)^{H-h+1}}{\rho} \right)^2 \leq \left( \frac{1 - (1-\rho)^H}{\rho} \right)^2.$$

Then we have

$$\sum_{\tau=1}^K \frac{\phi_h^\tau \phi_h^{\tau\top}}{[\mathbb{V}_h V_{h+1}^\star](s_h^\tau, a_h^\tau)} + \frac{1}{H^2} \mathbf{I} \succeq \sum_{\tau=1}^K \frac{\phi_h^\tau \phi_h^{\tau\top}}{\left(\frac{1-(1-\rho)^H}{\rho}\right)^2} + \frac{1}{H^2} \mathbf{I}.$$

Thus we have

$$\Sigma_h^{\star-1} = \Big( \sum_{\tau=1}^{K} \frac{\phi_h^\tau \phi_h^{\tau\top}}{[\mathbb{V}_h V_{h+1}^\star](s_h^\tau, a_h^\tau)} + \frac{1}{H^2}\mathbf{I} \Big)^{-1} \preceq \Big( \frac{1-(1-\rho)^H}{\rho} \Big)^2 \Big( \sum_{\tau=1}^{K} \phi_h^\tau \phi_h^{\tau\top} + \frac{1}{H^2}\mathbf{I} \Big)^{-1}.$$

By Theorem 5.2, we have

$$\text{SubOpt}(\hat{\pi}, s, \rho) \leq \tilde{O}(\sqrt{d}) \cdot \sup_{P \in \mathcal{U}^\rho(P^0)} \sum_{h=1}^{H} \mathbb{E}^{\pi^\star, P} \Big[ \sum_{i=1}^{d} \|\phi_i(s_h, a_h)\mathbf{1}_i\|_{\Sigma_h^{\star-1}} \big| s_1 = s \Big]$$

$$\leq \tilde{O}(\sqrt{d}) \cdot \frac{1-(1-\rho)^H}{\rho} \sup_{P \in \mathcal{U}^\rho(P^0)} \sum_{h=1}^{H} \mathbb{E}^{\pi^\star, P} \Big[ \sum_{i=1}^{d} \|\phi_i(s_h, a_h)\mathbf{1}_i\|_{\Lambda_h^{-1}} \big| s_1 = s \Big].$$

This concludes the proof. $\qquad\square$

## F  Proof of the Information-Theoretic Lower Bound

In this section, we prove the information-theoretic lower bound. We first introduce the construction of hard instances in Appendix F.1, then we prove Theorem 6.1 in Appendix F.2, and prove Corollary 6.2 in Appendix F.3.

### F.1  Construction of Hard Instances

We design a family of $d$-rectangular linear DRMDPs parameterized by a Boolean vector $\boldsymbol{\xi} = \{\boldsymbol{\xi}_h\}_{h \in [H]}$, where $\boldsymbol{\xi}_h \in \{-1, 1\}^d$. For a given $\boldsymbol{\xi}$ and uncertainty level $\rho \in (0, 3/4]$, the corresponding $d$-rectangular linear DRMDP $M_{\boldsymbol{\xi}}^\rho$ has the following structure. The state space $\mathcal{S} = \{x_1, x_2\}$ and the action space $\mathcal{A} = \{0, 1\}^d$. The initial state distribution $\mu_0$ is defined as

$$\mu_0(x_1) = \frac{d+1}{d+2} \quad \text{and} \quad \mu_0(x_2) = \frac{1}{d+2}.$$

The feature mapping $\phi : \mathcal{S} \times \mathcal{A} \to \mathbb{R}^{d+2}$ is defined as

$$\phi(x_1, a)^\top = \Big( \frac{a_1}{d}, \frac{a_2}{d}, \cdots, \frac{a_d}{d}, 1 - \sum_{i=1}^{d} \frac{a_i}{d}, 0 \Big)$$

$$\phi(x_2, a)^\top = \big( 0, 0, \cdots, 0, 0, 1 \big),$$

which satisfies $\phi_i(s, a) \geq 0$ and $\sum_{i=1}^{d} \phi_i(s, a) = 1$. The factor distributions $\{\boldsymbol{\mu}_h\}_{h \in [H]}$ are defined as

$$\boldsymbol{\mu}_h^\top = \big( \underbrace{\delta_{x_1}, \delta_{x_1}, \cdots, \delta_{x1}, \delta_{x_1}}_{d+1}, \delta_{x_2} \big), \forall h \in [H],$$

so the transition is homogeneous and does not depend on action but only on state. The reward parameters $\{\boldsymbol{\theta}_h\}_{h \in [H]}$ are defined as

$$\boldsymbol{\theta}_h^\top = \delta \cdot \Big( \frac{\xi_{h1}+1}{2}, \frac{\xi_{h2}+1}{2}, \cdots, \frac{\xi_{hd}+1}{2}, \frac{1}{2}, 0 \Big), \forall h \in [H],$$

where $\delta$ is a parameter to control the differences among instances, which is to be determined later. The reward $r_h$ is generated from the normal distribution $r_h \sim \mathcal{N}(r_h(s_h, a_h), 1)$, where $r_h(s, a) = \phi(s, a)^\top \boldsymbol{\theta}_h$. Note that

$$r_h(x_1, a) = \phi(x_1, a)^\top \boldsymbol{\theta}_h = \frac{\delta}{2d}(\langle \boldsymbol{\xi}_h, a \rangle + d) \geq 0 \quad \text{and} \quad r_h(x_2, a) = \phi(x_2, a)^\top \boldsymbol{\theta}_h = 0, \ \forall a \in \mathcal{A},$$

which means that $x_2$ is a worst state in terms of the mean reward. Thus, the worst case transition kernel should have the highest possible transition probability to $x_2$. This construction is pivotal in achieving a concise expression of robust value function. Further, we only consider model uncertainty

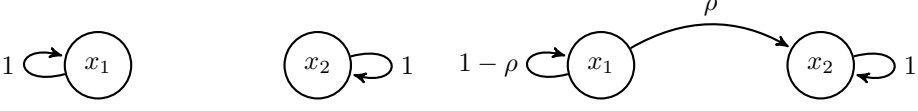

(a) The nominal environment.    (b) The worst case transition at the first step.

Figure 3: The nominal environment and the worst case environment. The value on each arrow represents the transition probability. The MDP has two states and $H$ steps. For the nominal environment, both $x_1$ and $x_2$ are absorbing states, which means that the state will always stay at the initial state in the nominal environment. The worst case environment on the right is obtained by perturbing the transition probability at the first step of the nominal environment, with others remain the same.

in the first step. By the fact that $x_2$ is the worse state, we know the worst case factor distribution for the first step is

$$\check{\boldsymbol{\mu}}_1^\top = \big((1-\rho)\delta_{x_1}+\rho\delta_{x_2},(1-\rho)\delta_{x_1}+\rho\delta_{x_2},\cdots,(1-\rho)\delta_{x_1}+\rho\delta_{x_2},(1-\rho)\delta_{x_1}+\rho\delta_{x_2},\delta_{x_2}\big).$$

We illustrate the designed $d$-rectangular linear DRMDP $M_{\boldsymbol{\xi}}^\rho$ in Figure 3(a) and Figure 3(b).

Finally, we design the procedure for collecting the offline dataset. We assume the $K$ trajectories are collected by a behavior policy $\pi^b = \{\pi_h^b\}_{h\in[H]}$ defined as

$$\pi_h^b \sim \text{Unif}\big(\{\boldsymbol{e}_1,\cdots,\boldsymbol{e}_d,\mathbf{0}\}\big), \forall h \in [H],$$

where $\{\boldsymbol{e}_i\}_{i\in[d]}$ are the canonical basis vectors in $\mathbb{R}^d$. The initial state is generated according to $\mu_0$. It is straightforward to check that the constructed hard instances satisfy Assumption 4.3. We denote the offline dataset as $\mathcal{D}$.

### F.2 Proof of Theorem 6.1

With this family of hard instances, we are ready to prove the information-theoretic lower bound. First, we define some notations. For any $\boldsymbol{\xi} \in \{-1,1\}^{dH}$, let $\mathbb{Q}_{\boldsymbol{\xi}}$ denote the distribution of dataset $\mathcal{D}$ collected from the MDP $M_{\boldsymbol{\xi}}$. Denote the family of parameters as $\Omega = \{-1,1\}^{dH}$ and the family of hard instances as $\mathcal{M} = \{M_{\boldsymbol{\xi}} : \boldsymbol{\xi} \in \Omega\}$.

*Proof of Theorem 6.1.* The proof constitutes three steps. In the first step, we lower bound the minimax suboptimality gap by testing error in the following Lemma F.1, the full proof of which can be found in Appendix G.6.

**Lemma F.1** (Reduction to testing). For the given family of $d$-rectangular linear DRMDPs, we have

$$\inf_{\hat{\pi}} \sup_{M\in\mathcal{M}} \text{SubOpt}(\hat{\pi},x_1,\rho) \geq (1-\rho)\cdot\frac{\delta dH}{8d}\cdot \min_{\substack{\boldsymbol{\xi},\boldsymbol{\xi}'\in\Omega \\ D_H(\boldsymbol{\xi},\boldsymbol{\xi}')=1}} \inf_{\psi}\Big[\mathbb{Q}_{\boldsymbol{\xi}}(\psi(\mathcal{D})\neq\boldsymbol{\xi})+\mathbb{Q}_{\boldsymbol{\xi}'}(\psi(\mathcal{D})\neq\boldsymbol{\xi}')\Big],$$

(F.1)

where for fixed indices $\boldsymbol{\xi}$ and $\boldsymbol{\xi}'$, $\psi$ is any test function taking value in $\{\boldsymbol{\xi},\boldsymbol{\xi}'\}$.

In the second step, we lower bound the testing error on the right hand side of (F.1) in the following Lemma F.2, the full proof of which can be found in Appendix G.7.

**Lemma F.2** (Lower bound on testing error). For the given family of $d$-rectangular linear DRMDPs, let $\delta = d^{3/2}/\sqrt{2K}$, then we have

$$\min_{\substack{\boldsymbol{\xi},\boldsymbol{\xi}' \\ D_H(\boldsymbol{\xi},\boldsymbol{\xi}')=1}} \inf_{\psi}\Big[\mathbb{Q}_{\boldsymbol{\xi}}(\psi(\mathcal{D})\neq\boldsymbol{\xi})+\mathbb{Q}_{\boldsymbol{\xi}'}(\psi(\mathcal{D})\neq\boldsymbol{\xi}')\Big] \geq \frac{1}{2}.$$

By Lemma F.1 and Lemma F.2, we have

$$\inf_{\hat{\pi}} \sup_{M\in\mathcal{M}} \text{SubOpt}(\hat{\pi},x_1,\rho) \geq \frac{d^{3/2}H}{128\sqrt{K}}.$$

(F.2)

In the last step, we upper bound the uncertainty function $\Phi(\Sigma_h^\star,s)$ in the following Lemma F.3, the full proof of which can be found in Appendix G.8.

**Lemma F.3.** For all $M_{\boldsymbol{\xi}} \in \mathcal{M}$, when $K \geq \tilde{O}(d^4)$, then with probability at least $1 - \delta$, we have

$$\sup_{P \in \mathcal{U}^\rho(P^0)} \sum_{h=1}^{H} \mathbb{E}^{\pi^\star, P}\Big[\sum_{i=1}^{d} \|\phi_i(s_h, a_h)\mathbf{1}_i\|_{\boldsymbol{\Sigma}_h^{\star-1}}\big|s_1 = x_1\Big] \leq \frac{4d^{3/2}H}{\sqrt{K}}.$$

By Lemma F.3 and (F.2), we know that with probability at least $1 - \delta$, there exist a universal constant $c$, such that

$$\inf_{\hat{\pi}} \sup_{M \in \mathcal{M}} \mathrm{SubOpt}(\hat{\pi}, x_1, \rho) \geq c \cdot \sup_{P \in \mathcal{U}^\rho(P^0)} \sum_{h=1}^{H} \mathbb{E}^{\pi^\star, P}\Big[\sum_{i=1}^{d} \|\phi_i(s_h, a_h)\mathbf{1}_i\|_{\boldsymbol{\Sigma}_h^{\star-1}}\big|s_1 = x_1\Big].$$

This concludes the proof. □

### F.3 Proof of Corollary 6.2

*Proof.* The result in Corollary 6.2 directly follows from the fact shown in (G.38): for the constructed hard instances, we have $\boldsymbol{\Sigma}_h^\star = \boldsymbol{\Lambda}_h$. Thus, we complete the proof by directly substituting $\boldsymbol{\Sigma}_h^\star$ in the result of Theorem 6.1 by $\boldsymbol{\Lambda}_h$. □

## G   Proof of Technical Lemmas

### G.1   Proof of Lemma D.1

*Proof.* First, we decompose $\mathrm{SubOpt}(\hat{\pi}, s, \rho)$ as follows

$$\mathrm{SubOpt}(\hat{\pi}, s, \rho) = \underbrace{V_1^{\pi^\star, \rho}(s) - \widehat{V}_1^\rho(s)}_{\mathrm{I}} + \underbrace{\widehat{V}_1^\rho(s) - V_1^{\hat{\pi}, \rho}(s)}_{\mathrm{II}},$$

then we bound term I and term II, respectively.

**Bounding term I:**   Note that

$$
\begin{aligned}
V_h^{\pi^\star, \rho}(s) - \widehat{V}_h^\rho(s) &= Q_h^{\pi^\star, \rho}(s, \pi_h^\star(s)) - \widehat{Q}_h^\rho(s, \hat{\pi}_h(s)) \\
&= Q_h^{\pi^\star, \rho}(s, \pi_h^\star(s)) - \widehat{Q}_h^\rho(s, \pi_h^\star(s)) + \widehat{Q}_h^\rho(s, \pi_h^\star(s)) - \widehat{Q}_h^\rho(s, \hat{\pi}_h(s)) \\
&\leq Q_h^{\pi^\star, \rho}(s, \pi_h^\star(s)) - \widehat{Q}_h^\rho(s, \pi_h^\star(s)). \qquad (\mathrm{G.1})
\end{aligned}
$$

Here (G.1) holds by the fact that $\hat{\pi}_h(s)$ is the greedy policy corresponding to $\widehat{Q}_h^\rho(s, a)$, which leads to $\widehat{Q}_h^\rho(s, \pi_h^\star(s)) - \widehat{Q}_h^\rho(s, \hat{\pi}_h(s)) \leq 0$. Further, by the robust Bellman equation (3.1), we have

$$Q_h^{\pi^\star, \rho}(s, \pi_h^\star(s)) - \widehat{Q}_h^\rho(s, \pi_h^\star(s))$$

$$= r_h(s, \pi_h^\star(s)) + \inf_{P_h(\cdot|s,a) \in \mathcal{U}_h^\rho(s,a;\boldsymbol{\mu}_{h,i}^0)} [\mathbb{P}_h V_{h+1}^{\pi^\star, \rho}](s, \pi_h^\star(s)) - \widehat{Q}_h^\rho(s, \pi_h^\star(s))$$

$$= r_h(s, \pi_h^\star(s)) + \inf_{P_h(\cdot|s,a) \in \mathcal{U}_h^\rho(s,a;\boldsymbol{\mu}_{h,i}^0)} [\mathbb{P}_h V_{h+1}^{\pi^\star, \rho}](s, \pi_h^\star(s)) - r_h(s, \pi_h^\star(s))$$

$$- \inf_{P_h(\cdot|s,a) \in \mathcal{U}_h^\rho(s,a;\boldsymbol{\mu}_{h,i}^0)} [\mathbb{P}_h \widehat{V}_{h+1}^\rho](s, \pi_h^\star(s)) + r_h(s, \pi_h^\star(s)) + \inf_{P_h(\cdot|s,a) \in \mathcal{U}_h^\rho(s,a;\boldsymbol{\mu}_{h,i}^0)} [\mathbb{P}_h \widehat{V}_{h+1}^\rho](s, \pi_h^\star(s))$$

$$- \widehat{Q}_h^\rho(s, \pi_h^\star(s)).$$

To proceed, we define the robust Bellman update error as follows

$$\zeta_h^\rho(s, a) = r_h(s, a) + \inf_{P_h(\cdot|s,a) \in \mathcal{U}_h^\rho(s,a;\boldsymbol{\mu}_{h,i}^0)} [\mathbb{P}_h \widehat{V}_{h+1}^\rho](s, a) - \widehat{Q}_h^\rho(s, a),$$

and denote the worst case transition kernel with respect to the estimated robust value function as $\widehat{P} = \{\widehat{P}_h\}_{h \in [H]}$, where $\widehat{P}_h(\cdot|s, a) = \arg\inf_{P_h(\cdot|s,a) \in \mathcal{U}_h^\rho(s,a;\boldsymbol{\mu}_{h,i}^0)} [\mathbb{P}_h \widehat{V}_{h+1}^\rho](s, a), \forall (s, a) \in \mathcal{S} \times \mathcal{A}$. Then we have

$$Q_h^{\pi^\star, \rho}(s, \pi_h^\star(s)) - \widehat{Q}_h^\rho(s, \pi_h^\star(s))$$

$$
\begin{aligned}
&= \inf_{P_h(\cdot|s,a)\in\mathcal{U}_h^\rho(s,a;\boldsymbol{\mu}_{h,i}^0)}[\mathbb{P}_h V_{h+1}^{\pi^\star;\rho}](s,\pi_h^\star(s)) - \inf_{P_h(\cdot|s,a)\in\mathcal{U}_h^\rho(s,a;\boldsymbol{\mu}_{h,i}^0)}[\mathbb{P}_h \widehat{V}_{h+1}^\rho](s,\pi_h^\star(s)) + \zeta_h^\rho(s,\pi_h^\star(s)) \\
&\leq \left[\widehat{\mathbb{P}}_h(V_{h+1}^{\pi^\star;\rho} - \widehat{V}_{h+1}^\rho)\right](s,\pi_h^\star(s)) + \zeta_h^\rho(s,\pi_h^\star(s)). \tag{G.2}
\end{aligned}
$$

Combining (G.1) and (G.2), we have for any $h\in[H]$,

$$
V_h^{\pi^\star,\rho}(s) - \widehat{V}_h^\rho(s) \leq \left[\widehat{\mathbb{P}}_h(V_{h+1}^{\pi^\star;\rho} - \widehat{V}_{h+1}^\rho)\right](s,\pi_h^\star(s)) + \zeta_h^\rho(s,\pi_h^\star(s)). \tag{G.3}
$$

Recursively applying (G.3), we have

$$
V_1^{\pi^\star,\rho}(s) - \widehat{V}_1^\rho(s) \leq \sum_{h=1}^H \mathbb{E}^{\pi^\star,\widehat{P}}\left[\zeta_h^\rho(s_h,a_h)|s_1=s\right]. \tag{G.4}
$$

**Bounding term II:** Note that $\widehat{V}_h^\rho(s) - V_h^{\hat{\pi},\rho}(s) = \widehat{Q}_h^\rho(s,\hat{\pi}_h(s)) - Q_h^{\hat{\pi},\rho}(s,\hat{\pi}_h(s))$, by the robust Bellman equation (3.1), we have

$$
\begin{aligned}
&\widehat{V}_h^\rho(s) - V_h^{\hat{\pi},\rho}(s) \\
&= \widehat{Q}_h^\rho(s,\hat{\pi}_h(s)) - r_h(s,\hat{\pi}_h(s)) - \inf_{P_h(\cdot|s,a)\in\mathcal{U}_h^\rho(s,a;\boldsymbol{\mu}_{h,i}^0)}[\mathbb{P}_h \widehat{V}_{h+1}^\rho](s,\hat{\pi}_h(s)) \\
&\quad + \inf_{P_h(\cdot|s,a)\in\mathcal{U}_h^\rho(s,a;\boldsymbol{\mu}_{h,i}^0)}[\mathbb{P}_h \widehat{V}_{h+1}^\rho](s,\hat{\pi}_h(s)) - \inf_{P_h(\cdot|s,a)\in\mathcal{U}_h^\rho(s,a;\boldsymbol{\mu}_{h,i}^0)}[\mathbb{P}_h V_{h+1}^{\hat{\pi},\rho}](s,\hat{\pi}_h(s)) \\
&= -\zeta_h^\rho(s,\hat{\pi}_h(s)) + \inf_{P_h(\cdot|s,a)\in\mathcal{U}_h^\rho(s,a;\boldsymbol{\mu}_{h,i}^0)}[\mathbb{P}_h \widehat{V}_{h+1}^\rho](s,\hat{\pi}_h(s)) - \inf_{P_h(\cdot|s,a)\in\mathcal{U}_h^\rho(s,a;\boldsymbol{\mu}_{h,i}^0)}[\mathbb{P}_h V_{h+1}^{\hat{\pi},\rho}](s,\hat{\pi}_h(s)).
\end{aligned}
$$

To proceed, we denote the worst case transition kernel with respect to the robust value function of $\hat{\pi}$ as $P^{\hat{\pi}} = \{P_h^{\hat{\pi}}\}_{h\in[H]}$, where $P_h^{\hat{\pi}}(\cdot|s,a) = \arg\inf_{P_h(\cdot|s,a)\in\mathcal{U}_h^\rho(s,a;\boldsymbol{\mu}_{h,i}^0)}[\mathbb{P}_h V_{h+1}^{\hat{\pi},\rho}](s,a)$, then we have

$$
\widehat{V}_h^\rho(s) - V_h^{\hat{\pi},\rho}(s) \leq -\zeta_h^\rho(s,\hat{\pi}_h(s)) + \left[\mathbb{P}_h^{\hat{\pi}}(\widehat{V}_{h+1}^\rho - V_{h+1}^{\hat{\pi},\rho})\right](s,\hat{\pi}_h(s)). \tag{G.5}
$$

Applying (G.5) recursively, we have

$$
\widehat{V}_1^\rho(s) - V_1^{\hat{\pi},\rho}(s) \leq \sum_{h=1}^H \mathbb{E}^{\hat{\pi},P^{\hat{\pi}}}\left[-\zeta_h^\rho(s_h,a_h)|s_1=s\right]. \tag{G.6}
$$

Now it remains to bound the robust Bellman error $\zeta_h^\rho(\cdot,\cdot)$. In particular, we aim to show that for all $(s,a,h)\in\mathcal{S}\times\mathcal{A}\times[H]$,

$$
0 \leq \zeta_h^\rho(s,a) \leq 2\Gamma_h(s,a).
$$

Note that $\zeta_h^\rho(s,a) = r_h(s,a) + \inf_{P_h(\cdot|s,a)\in\mathcal{U}_h^\rho(s,a;\boldsymbol{\mu}_{h,i}^0)}[\mathbb{P}_h\widehat{V}_{h+1}^\rho](s,a) - \widehat{Q}_h^\rho(s,a)$. Recall the definition of $\widehat{Q}_h^\rho(s,a)$ in Algorithm 1 and the notation in (E.1), and we have

$$
\begin{aligned}
\zeta_h^\rho(s,a) = r_h(s,a) + &\inf_{P_h(\cdot|s,a)\in\mathcal{U}_h^\rho(s,a;\boldsymbol{\mu}_{h,i}^0)}[\mathbb{P}_h\widehat{V}_{h+1}^\rho](s,a) \\
&- \max\left\{r_h(s,a) + \widehat{\inf_{P_h(\cdot|s,a)\in\mathcal{U}_h^\rho(s,a;\boldsymbol{\mu}_{h,i}^0)}}[\mathbb{P}_h\widehat{V}_{h+1}^\rho](s,a) - \Gamma_h(s,a), 0\right\}.
\end{aligned}
$$

If $r_h(s,a) + \widehat{\inf}_{P_h(\cdot|s,a)\in\mathcal{U}_h^\rho(s,a;\boldsymbol{\mu}_{h,i}^0)}[\mathbb{P}_h\widehat{V}_{h+1}^\rho](s,a) - \Gamma_h(s,a) \leq 0$, then $\zeta_h^\rho(s,a) = r_h(s,a) + \inf_{P_h(\cdot|s,a)\in\mathcal{U}_h^\rho(s,a;\boldsymbol{\mu}_{h,i}^0)}[\mathbb{P}_h\widehat{V}_{h+1}^\rho](s,a) \geq 0$. If $r_h(s,a) + \widehat{\inf}_{P_h(\cdot|s,a)\in\mathcal{U}_h^\rho(s,a;\boldsymbol{\mu}_{h,i}^0)}[\mathbb{P}_h\widehat{V}_{h+1}^\rho](s,a) - \Gamma_h(s,a) > 0$, then we have $\zeta_h^\rho(s,a) = r_h(s,a) + \inf_{P_h(\cdot|s,a)\in\mathcal{U}_h^\rho(s,a;\boldsymbol{\mu}_{h,i}^0)}[\mathbb{P}_h\widehat{V}_{h+1}^\rho](s,a) - r_h(s,a) - \widehat{\inf}_{P_h(\cdot|s,a)\in\mathcal{U}_h^\rho(s,a;\boldsymbol{\mu}_{h,i}^0)}[\mathbb{P}_h\widehat{V}_{h+1}^\rho](s,a) + \Gamma_h(s,a) \geq -\Gamma_h(s,a) + \Gamma_h(s,a) = 0$, where we used the condition in (D.1). In conclusion, we have $\zeta_h^\rho(s,a) \geq 0$.

On the other hand, we always have

$$
\begin{aligned}
\zeta_h^\rho(s,a) &\leq \inf_{P_h(\cdot|s,a)\in\mathcal{U}_h^\rho(s,a;\boldsymbol{\mu}_{h,i}^0)}[\mathbb{P}_h\widehat{V}_{h+1}^\rho](s,a) - \widehat{\inf_{P_h(\cdot|s,a)\in\mathcal{U}_h^\rho(s,a;\boldsymbol{\mu}_{h,i}^0)}}[\mathbb{P}_h\widehat{V}_{h+1}^\rho](s,a) + \Gamma_h(s,a) \\
&\leq 2\Gamma_h(s,a).
\end{aligned}
$$

Thus, for all $(s, a, h) \in \mathcal{S} \times \mathcal{A} \times [H]$, we have

$$0 \le \zeta_h^\rho(s, a) \le 2\Gamma_h(s, a). \tag{G.7}$$

Combining (G.4), (G.6) and (G.7), we have

$$
\begin{aligned}
\mathrm{SubOpt}(\hat\pi, s, \rho) &\le \sum_{h=1}^{H} \mathbb{E}^{\pi^\star, \widehat{P}}\big[\zeta_h(s_h, a_h)|s_1 = s\big] + \sum_{h=1}^{H} \mathbb{E}^{\hat\pi, P^{\hat\pi}}\big[-\zeta_h^\rho(s_h, a_h)|s_1 = s\big] \\
&\le \sum_{h=1}^{H} \mathbb{E}^{\pi^\star, \widehat{P}}\big[\zeta_h^\rho(s_h, a_h)|s_1 = s\big] \\
&\le \sup_{P \in \mathcal{U}^\rho(P^0)} \sum_{h=1}^{H} \mathbb{E}^{\pi^\star, P}\big[\zeta_h^\rho(s_h, a_h)|s_1 = s\big] \\
&\le 2 \sup_{P \in \mathcal{U}^\rho(P^0)} \sum_{h=1}^{H} \mathbb{E}^{\pi^\star, P}\big[\Gamma_h(s_h, a_h)|s_1 = s\big].
\end{aligned}
$$

This concludes the proof. $\qquad\square$

## G.2 Proof of Lemma D.2

In this section, we prove Lemma D.2. Before the proof, we first present several auxiliary lemmas.

**Lemma G.1** (Reference-Advantage Decomposition). There exist real values $\{\alpha_i\}_{i \in [d]}$, where $\alpha_i \in [0, H], \forall i \in [d]$, such that

$$
\begin{aligned}
&\left| \inf_{P_h(\cdot|s,a) \in \mathcal{U}_h^\rho(s,a;\boldsymbol{\mu}_{h,i}^0)} [\mathbb{P}_h \widehat{V}_{h+1}^\rho](s,a) - \widehat{\inf_{P_h(\cdot|s,a) \in \mathcal{U}_h^\rho(s,a;\boldsymbol{\mu}_{h,i}^0)}} [\mathbb{P}_h \widehat{V}_{h+1}^\rho](s,a) \right| \\
&\le \underbrace{\lambda \sum_{i=1}^{d} \|\phi_i(s,a)\mathbf{1}_i\|_{\boldsymbol{\Lambda}_h^{-1}} \|\mathbb{E}^{\boldsymbol{\mu}_h^0}[V_{h+1}^{\star,\rho}(s)]_{\alpha_i}\|_{\boldsymbol{\Lambda}_h^{-1}}}_{\text{i}} + \underbrace{\sum_{i=1}^{d} \|\phi_i(s,a)\mathbf{1}_i\|_{\boldsymbol{\Lambda}_h^{-1}} \left\| \sum_{\tau=1}^{K} \boldsymbol{\phi}_h^\tau \eta_h^\tau([V_{h+1}^{\star,\rho}]_{\alpha_i}) \right\|_{\boldsymbol{\Lambda}_h^{-1}}}_{\text{ii}} \\
&\quad + \underbrace{\lambda \sum_{i=1}^{d} \|\phi_i(s,a)\mathbf{1}_i\|_{\boldsymbol{\Lambda}_h^{-1}} \left\| \mathbb{E}^{\boldsymbol{\mu}_h^0}\big[[\widehat{V}_{h+1}^\rho(s)]_{\alpha_i} - [V_{h+1}^{\star,\rho}(s)]_{\alpha_i}\big] \right\|_{\boldsymbol{\Lambda}_h^{-1}}}_{\text{iii}} \\
&\quad + \underbrace{\sum_{i=1}^{d} \|\phi_i(s,a)\mathbf{1}_i\|_{\boldsymbol{\Lambda}_h^{-1}} \left\| \sum_{\tau=1}^{K} \boldsymbol{\phi}_h^\tau \eta_h^\tau([\widehat{V}_{h+1}^\rho(s)]_{\alpha_i} - [V_{h+1}^{\star,\rho}(s)]_{\alpha_i}) \right\|_{\boldsymbol{\Lambda}_h^{-1}}}_{\text{iv}},
\end{aligned}
$$

where $\eta_h^\tau([f]_{\alpha_i}) = \big([\mathbb{P}_h^0[f]_{\alpha_i}](s_h^\tau, a_h^\tau) - [f(s_{h+1}^\tau)]_{\alpha_i}\big)$, for any function $f : \mathcal{S} \to [0, H-1]$.

**Lemma G.2** (Bound of Weights). For any $h \in [H]$, denote the weight $\boldsymbol{w}_h^\rho = \boldsymbol{\theta}_h + \hat{\boldsymbol{\nu}}_h^\rho$ in Algorithm 1, then $\boldsymbol{w}_h^\rho$ satisfies

$$\|\boldsymbol{w}_h^\rho\|_2 \le 2H\sqrt{dK/\lambda}.$$

**Lemma G.3.** [15, Lemma B.2] Let $f : \mathcal{S} \to [0, R-1]$ be any fixed function. For any $\delta \in (0, 1)$, we have

$$\mathbb{P}\left( \left\| \sum_{\tau=1}^{K} \boldsymbol{\phi}_h^\tau \cdot \eta_h^\tau(f) \right\|_{\boldsymbol{\Lambda}_h^{-1}}^2 \ge R^2 \left( 2\log\left(\frac{1}{\delta}\right) + d\log\left(1 + \frac{K}{\lambda}\right) \right) \right) \le \delta.$$

**Lemma G.4** (Covering number of function class $\mathcal{V}_h$). For any $h \in [H]$, let $\mathcal{V}_h$ denote a class of functions mapping from $\mathcal{S}$ to $\mathbb{R}$ with the following parametric form

$$\mathcal{V}_h(s) = \max_{a \in \mathcal{A}} \left\{ \boldsymbol{\phi}(s,a)^\top \boldsymbol{\theta} - \beta \sum_{i=1}^{d} \sqrt{\phi_i(s,a)\mathbf{1}_i^\top \boldsymbol{\Sigma}_h^{-1} \phi_i(s,a)\mathbf{1}_i} \right\}_{[0, H-h+1]},$$

where the parameters $(\boldsymbol{\theta}, \beta, \boldsymbol{\Sigma}_h)$ satisfy $\|\boldsymbol{\theta}\| \leq L$, $\beta \in [0, B]$, $\lambda_{\min}(\boldsymbol{\Sigma}_h) \geq \lambda$. Assume $\|\boldsymbol{\phi}(s,a)\| \leq 1$ for all (s,a) pairs, and let $\mathcal{N}_h(\epsilon)$ be the $\epsilon$-covering number of $\mathcal{V}$ with respect to the distance $\text{dist}(V_1, V_2) = \sup_x |V_1(x) - V_2(x)|$. Then

$$\log \mathcal{N}_h(\epsilon) \leq d \log(1 + 4L/\epsilon) + d^2 \log\left[1 + 8d^{1/2}B^2/(\lambda\epsilon^2)\right].$$

**Lemma G.5.** [47, Covering number of an interval] Denote the $\epsilon$-covering number of the closed interval $[a, b]$ for some real number $b > a$ with respect to the distance metric $d(\alpha_1, \alpha_2) = |\alpha_1 - \alpha_2|$ as $\mathcal{N}_\epsilon([a, b])$. Then we have $\mathcal{N}_\epsilon([a, b]) \leq 3(b - a)/\epsilon$.

*Proof of Lemma D.2.* To prove this lemma, we bound terms i-iv in Lemma G.1 at each step $h \in [H]$, respectively. To deal with the temporal dependency, we follow the induction procedure proposed in [54] and make essential adjustments to adapt to the robust setting.

**The base case.** We start from the last step $H$. By the fact that any robust value function is upper bounded by $H$, then with $\lambda = 1$, for all $(s, a) \in \mathcal{S} \times \mathcal{A}$, we have

$$\text{i} + \text{iii} \leq 2H \sum_{i=1}^{d} \|\phi_i(s,a)\mathbf{1}_i\|_{\boldsymbol{\Lambda}_h^{-1}}. \tag{G.8}$$

Next, we bound term ii. Note that $V_{H+1}^{\star,\rho}$ is independent of $\mathcal{D}$, we can directly apply Hoeffding-type self-normalized concentration inequality Lemma I.1 and a union bound to obtain the upper bound. In concrete, we define the filtration $\mathcal{F}_{\tau-1,h} = \sigma(\{(s_h^j, a_h^j)\}_{j=1}^{\tau} \cup \{s_{h+1}^j\}_{j=1}^{\tau-1})$. Since $V_{H+1}^{\star,\rho}$ is independent of $\mathcal{D}$ and is upper bounded by $H$, thus we have $\eta_H^\tau([V_{H+1}^{\star,\rho}]_{\alpha_i})|\mathcal{F}_{\tau-1,H}$ is mean zero, i.e., $\mathbb{E}[\eta_H^\tau([V_{H+1}^{\star,\rho}]_{\alpha_i})|\mathcal{F}_{\tau-1,H}] = 0$ and $H$-subGaussian. By Lemma I.1, for any fixed index $i \in [d]$, with probability at least $1 - \delta/2dH^2$, we have

$$\left\| \sum_{\tau=1}^{K} \phi_H^\tau \eta_H^\tau([V_{H+1}^{\star,\rho}]_{\alpha_i}) \right\|_{\boldsymbol{\Lambda}_h^{-1}}^2 \leq 2H^2 \log\left( \frac{2dH^2 \det(\boldsymbol{\Lambda}_h)^{1/2}}{\delta \det(\lambda\mathbf{I})^{1/2}} \right).$$

By the proof of Lemma B.2 in [15], we know $\det(\boldsymbol{\Lambda}_h) \leq (\lambda + K)^d$. Thus, we have

$$\left\| \sum_{\tau=1}^{K} \phi_H^\tau \eta_H^\tau([V_{H+1}^{\star,\rho}]_{\alpha_i}) \right\|_{\boldsymbol{\Lambda}_h^{-1}}^2 \leq 2H^2 \left( \frac{d}{2} \log \frac{\lambda + K}{\lambda} + \log \frac{2dH^2}{\delta} \right) \leq dH^2 \log \frac{2dH^2 K}{\delta}.$$

Then by a union bound over $i \in [d]$, with probability at least $1 - \delta/2H^2$, we have

$$\text{ii} \leq \sqrt{d}H\sqrt{\iota} \sum_{i=1}^{d} \|\phi_i(s,a)\mathbf{1}_i\|_{\boldsymbol{\Lambda}_h^{-1}}, \tag{G.9}$$

where $\iota = \log(2dH^2K/\delta) \geq 1$. As for the term iv, by construction we have $V_{H+1}^{\star,\rho} = \widehat{V}_{H+1}^\rho = 0$ with probability 1. Thus, we trivially have

$$\text{iv} \leq \sqrt{d}H\sqrt{\iota} \sum_{i=1}^{d} \|\phi_i(s,a)\mathbf{1}_i\|_{\boldsymbol{\Lambda}_h^{-1}}. \tag{G.10}$$

Combining (G.8), (G.9) and (G.10), for all $(s, a) \in \mathcal{S} \times \mathcal{A}$, with probability at least $1 - \delta/2H^2$, we have

$$\left| \inf_{P_H(\cdot|s,a) \in \mathcal{U}_H^\rho(s,a;\boldsymbol{\mu}_{H,i}^0)} [\mathbb{P}_H \widehat{V}_{H+1}^\rho](s,a) - \widehat{\inf_{P_H(\cdot|s,a) \in \mathcal{U}_H^\rho(s,a;\boldsymbol{\mu}_{H,i}^0)} [\mathbb{P}_H \widehat{V}_{H+1}^\rho]}(s,a) \right|$$

$$\leq 4\sqrt{d}H\sqrt{\iota} \sum_{i=1}^{d} \|\phi_i(s,a)\mathbf{1}_i\|_{\boldsymbol{\Lambda}_h^{-1}}. \tag{G.11}$$

Thus, we define $\Gamma_H(s,a) := 4\sqrt{d}H\sqrt{\iota} \sum_{i=1}^{d} \|\phi_i(s,a)\mathbf{1}_i\|_{\boldsymbol{\Lambda}_h^{-1}}$. By the definition of $\widehat{Q}_H^\rho(s,a)$ in Algorithm 1, we have

$$\widehat{Q}_H^\rho(s,a) = \left\{ r_H(s,a) - \Gamma_H(s,a) \right\}_{[0,1]} \leq r_H(s,a) = Q_H^{\star,\rho}(s,a),$$

which implies that a pessimistic estimation is achieved at step $H$, i.e., $V_H^{\star,\rho}(s) \geq \widehat{V}_H^\rho(s), \forall s \in \mathcal{S}$. Next, we study $V_H^{\star,\rho}(s) - \widehat{V}_H^\rho(s)$. The intuition is that given the estimation error bound in (G.11), with sufficient data, the difference between $V_H^{\star,\rho}(s)$ and $\widehat{V}_H^\rho(s)$ should be small. Specifically, we have

$$
\begin{aligned}
V_H^{\star,\rho}(s) - \widehat{V}_H^\rho(s) &= Q_H^{\star,\rho}(s, \pi_H^\star(s)) - \widehat{Q}_H^\rho(s, \pi_H^\star(s)) + \widehat{Q}_H^\rho(s, \pi_H^\star(s)) - \widehat{Q}_H^\rho(s, \hat\pi(s)) \\
&\leq r_H(s, \pi_H^\star(s)) + \inf_{P_H(\cdot|s,a)\in\mathcal{U}_H^\rho(s,a;\boldsymbol{\mu}_{H,i}^0)} [\mathbb{P}_H \widehat{V}_{H+1}^\rho](s,a) - \\
&\quad\ r_H(s, \pi_H^\star(s)) - \widehat{\inf_{P_H(\cdot|s,a)\in\mathcal{U}_H^\rho(s,a;\boldsymbol{\mu}_{H,i}^0)}} [\mathbb{P}_H \widehat{V}_{H+1}^\rho](s,a) + \Gamma_H(s, \pi_H^\star(s))
\end{aligned}
\tag{G.12}
$$

$$
\leq 2\Gamma_H(s, \pi_H^\star(s)),
$$

where (G.12) holds by the robust Bellman equation (3.1) and the fact that $\widehat{Q}_H^\rho(s, \pi_H^\star(s)) - \widehat{Q}_H^\rho(s, \hat\pi(s)) \leq 0$. Then we bound the pessimism term $\Gamma_H(s,a)$ in terms of the sample size $K$. By Lemma I.3, when $K \geq \max\{512 \log(2dH^2/\delta)/\kappa^2, 4/\kappa\}$, with probability at least $1 - \delta/2H^2$, we have

$$
2\Gamma_H(s,a) = 8\sqrt{d}H\sqrt{\iota} \sum_{i=1}^d \|\phi_i(s,a)\mathbf{1}_i\|_{\Lambda_h^{-1}} \leq \frac{16\sqrt{d}H\sqrt{\iota}}{\sqrt{K}} \sum_{i=1}^d \phi_i(s,a)(\tilde{\Lambda}_H^{-1})_{ii}^{1/2},
$$

where $\tilde{\Lambda}_H = \mathbb{E}^{\pi^b, P^0}[\phi(s_H, a_H)\phi(s_H, a_H)^\top]$. Note that for any positive definite matrix $A$, we know $\lambda_{\min}(A) \leq A_{ii} \leq \lambda_{\max}(A)$. Thus, by Assumption 4.3, we have

$$
2\Gamma_H(s,a) \leq \frac{16\sqrt{d}H \cdot 1\sqrt{\iota}}{\sqrt{K\kappa}} := R_H.
\tag{G.13}
$$

To summarize, we define the event

$$
\mathcal{E}_H = \left\{0 \leq V_H^{\star,\rho}(s) - \widehat{V}_H^\rho(s) \leq R_H, \forall s \in \mathcal{S}\right\}.
$$

Then by a union bound over (G.11) and (G.13), we know $\mathcal{E}_H$ holds with probability at least $1 - \delta_H = 1 - \delta/H^2$. This concludes the proof of the base case.

**Inductive Hypothesis.** Suppose with probability at least $1 - \delta_{h+1}$, we have

$$
\left| \inf_{P_{h+1}(\cdot|s,a)\in\mathcal{U}_{h+1}^\rho(s,a;\boldsymbol{\mu}_{h+1,i}^0)} [\mathbb{P}_{h+1} \widehat{V}_{h+2}^\rho](s,a) - \widehat{\inf_{P_{h+1}(\cdot|s,a)\in\mathcal{U}_{h+1}^\rho(s,a;\boldsymbol{\mu}_{h+1,i}^0)}} [\mathbb{P}_{h+1} \widehat{V}_{h+2}^\rho](s,a) \right|
$$

$$
\leq 4\sqrt{d}H\sqrt{\iota} \sum_{i=1}^d \|\phi_i(s,a)\mathbf{1}_i\|_{\Lambda_{h+1}^{-1}} := \Gamma_{h+1}(s,a),
\tag{G.14}
$$

and

$$
\mathcal{E}_{h+1} = \left\{0 \leq V_{h+1}^\star(s) - \widehat{V}_{h+1}(s) \leq R_{h+1} := \frac{16\sqrt{d}H(H-h)\sqrt{\iota}}{\sqrt{K\kappa}}, \forall s \in \mathcal{S}\right\}.
\tag{G.15}
$$

**Inductive Step.** Next, we establish the result for step $h$. First, terms i, ii and iii at step $h$ can be similarly bounded as in the base case, i.e., we have

$$
\text{i} + \text{ii} + \text{iii} \leq 3\sqrt{d}H\sqrt{\iota} \sum_{i=1}^d \|\phi_i(s,a)\mathbf{1}_i\|_{\Lambda_h^{-1}},
\tag{G.16}
$$

with probability at least $1 - \delta/3H^2$. It remains to bound the term iv and ensure it is non-dominating. Here, we need to deal with the temporal dependency, as $[\widehat{V}_{h+1}^\rho(s)]_\alpha - [V_{h+1}^{\star,\rho}(s)]_\alpha$ is correlated to $\{(s_h^\tau, a_h^\tau, s_{h+1}^\tau)\}_{\tau=1}^K$, thus we need a uniform concentration argument. Consider the function class

$$
\mathcal{V}_h(D, B, \lambda) = \{V_h(s; \theta, \beta, \Sigma) : \mathcal{S} \to [0, H] \text{ with } \|\theta\| \leq D, \beta \in [0, B], \Sigma \succeq \lambda I\},
$$

where $V_h(s;\theta,\beta,\boldsymbol{\Sigma}) = \max_{a\in\mathcal{A}}\{\boldsymbol{\phi}(s,a)^\top\theta - \beta\sum_{i=1}^d\sqrt{\phi_i(s,a)\mathbf{1}_i^\top\boldsymbol{\Sigma}^{-1}\phi_i(s,a)\mathbf{1}_i}\}_{[0,H-h+1]}$. For simplicity, we denote $f_{\alpha_i}(s) := [\widehat{V}_{h+1}^\rho(s)]_{\alpha_i} - [V_{h+1}^{\star,\rho}(s)]_{\alpha_i}$, then $f_{\alpha_i}\in\mathcal{F}_{h+1}(\alpha_i)$, where

$$\mathcal{F}_{h+1}(\alpha) := \big\{[\widehat{V}_{h+1}^\rho(s)]_\alpha - [V_{h+1}^{\star,\rho}(s)]_\alpha : \widehat{V}_{h+1}^\rho(s)\in\mathcal{V}_{h+1}(D_0,B_0,\lambda)\big\}.$$

Note that for any fixed $\alpha$, the covering number of $\mathcal{F}_{h+1}(\alpha)$ is the same as that of $\mathcal{V}_h(D_0,B_0,\lambda)$. By Lemma G.2, we have $D_0 = H\sqrt{Kd/\lambda}$. By the induction assumption (G.14), we have $B_0 = 4\sqrt{d}H\sqrt{\iota}$. Denote the $\epsilon$-covering of the interval $[0,H]$ with respect to the distance $\mathrm{dist}(\alpha_1,\alpha_2) = |\alpha_1 - \alpha_2|$ as $\mathcal{N}_{[0,H]}(\epsilon)$, and its $\epsilon$-covering number as $|\mathcal{N}_{[0,H]}(\epsilon)|$. For each $\alpha\in[0,H]$, we can find $\alpha_\epsilon\in\mathcal{N}_{[0,H]}(\epsilon)$ such that $|\alpha - \alpha_\epsilon|\le\epsilon$. For any fixed $\alpha\in[0,H]$, we denote the $\epsilon$-covering of $\mathcal{F}_{h+1}(\alpha)$ with respect to the distance $\mathrm{dist}(f_1,f_2) = \sup_x|f_1(x) - f_2(x)|$ as $\mathcal{N}_{h+1}(\epsilon)$ (short for $\mathcal{N}_{h+1}(\epsilon;D,B,\lambda)$) and its $\epsilon$-covering number as $|\mathcal{N}_{h+1}(\epsilon)|$. For each $f_\alpha\in\mathcal{F}_{h+1}(\alpha)$, we can find $f_\alpha^\epsilon\in\mathcal{N}_{h+1}(\epsilon)$ such that $\sup_s|f_\alpha(s) - f_\alpha^\epsilon(s)|\le\epsilon$. It follows that

$$\Big\|\sum_{k=1}^K\boldsymbol{\phi}_h^\tau\eta_h^\tau(f_{\alpha_i})\Big\|_{\boldsymbol{\Lambda}_h^{-1}}^2\cdot\mathbb{1}\big\{\|f_{\alpha_i}\|_\infty\le R_{h+1}\big\}$$

$$\le 2\Big\|\sum_{\tau=1}^K\boldsymbol{\phi}_h^\tau\eta_h^\tau(f_{\alpha_{i_\epsilon}})\Big\|_{\boldsymbol{\Lambda}_h^{-1}}^2\cdot\mathbb{1}\big\{\|f_{\alpha_{i_\epsilon}}\|_\infty\le R_{h+1}+\epsilon\big\} + 2\Big\|\sum_{k=1}^K\boldsymbol{\phi}_h^\tau\big(\eta_h^\tau(f_{\alpha_i}) - \eta_h^\tau(f_{\alpha_{i_\epsilon}})\big)\Big\|_{\boldsymbol{\Lambda}_h^{-1}}^2.$$

Note that

$$2\Big\|\sum_{k=1}^K\boldsymbol{\phi}_h^\tau\big(\eta_h^\tau(f_{\alpha_i}) - \eta_h^\tau(f_{\alpha_{i_\epsilon}})\big)\Big\|_{\boldsymbol{\Lambda}_h^{-1}}^2 \le 2\epsilon^2\sum_{\tau,\tau'=1}^K\big|\boldsymbol{\phi}_h^\tau\boldsymbol{\Lambda}_h^{-1}\boldsymbol{\phi}_h^{\tau'}\big| \le 2\epsilon^2 K^2/\lambda.$$

Then we have

$$\Big\|\sum_{k=1}^K\boldsymbol{\phi}_h^\tau\eta_h^\tau(f_{\alpha_i})\Big\|_{\boldsymbol{\Lambda}_h^{-1}}^2\cdot\mathbb{1}\big\{\|f_{\alpha_i}\|_\infty\le R_{h+1}\big\}$$

$$\le 4\Big\|\sum_{\tau=1}^K\boldsymbol{\phi}_h^\tau\eta_h^\tau(f_{\alpha_{i_\epsilon}}^\epsilon)\Big\|_{\boldsymbol{\Lambda}_h^{-1}}^2\cdot\mathbb{1}\big\{\|f_{\alpha_{i_\epsilon}}^\epsilon\|_\infty\le R_{h+1}+2\epsilon\big\}$$

$$+ 4\Big\|\sum_{k=1}^K\boldsymbol{\phi}_h^\tau\big(\eta_h^\tau(f_{\alpha_{i_\epsilon}}) - \eta_h^\tau(f_{\alpha_{i_\epsilon}}^\epsilon)\big)\Big\|_{\boldsymbol{\Lambda}_h^{-1}}^2 + \frac{2\epsilon^2 K^2}{\lambda}$$

$$\le 4\Big\|\sum_{\tau=1}^K\boldsymbol{\phi}_h^\tau\eta_h^\tau(f_{\alpha_{i_\epsilon}}^\epsilon)\Big\|_{\boldsymbol{\Lambda}_h^{-1}}^2\cdot\mathbb{1}\big\{\|f_{\alpha_{i_\epsilon}}^\epsilon\|_\infty\le R_{h+1}+2\epsilon\big\} + \frac{6\epsilon^2 K^2}{\lambda},$$

where the last inequality holds by the fact that

$$4\Big\|\sum_{k=1}^K\boldsymbol{\phi}_h^\tau\big(\eta_h^\tau(f_{\alpha_{i_\epsilon}}) - \eta_h^\tau(f_{\alpha_{i_\epsilon}}^\epsilon)\big)\Big\|_{\boldsymbol{\Lambda}_h^{-1}}^2 \le 4\epsilon^2\sum_{\tau,\tau'=1}^K\big|\boldsymbol{\phi}_h^\tau\boldsymbol{\Lambda}_h^{-1}\boldsymbol{\phi}_h^{\tau'}\big| \le 4\epsilon^2 K^2/\lambda.$$

With a union bound over $\mathcal{N}_{h+1}(\epsilon)$ and $\mathcal{N}_{[0,H]}(\epsilon)$ and by Lemma G.3, we have

$$\mathbb{P}\bigg\{\sup_{\substack{\alpha_{i_\epsilon}\in\mathcal{N}_{[0,H]}(\epsilon)\\ f_{\alpha_{i_\epsilon}}^\epsilon\in\mathcal{N}_{h+1}(\epsilon)}}\Big\|\sum_{\tau=1}^K\boldsymbol{\phi}_h^\tau\eta_h^\tau(f_{\alpha_{i_\epsilon}}^\epsilon)\Big\|_{\boldsymbol{\Lambda}_h^{-1}}^2\cdot\mathbb{1}\big\{\|f_{\alpha_{i_\epsilon}}^\epsilon\|_\infty\le R_{h+1}+2\epsilon\big\}$$

$$> (R_{h+1}+2\epsilon)^2\Big(2\log\frac{3dH^2|\mathcal{N}_{h+1}(\epsilon)||\mathcal{N}_{[0,H]}(\epsilon)|}{\delta} + d\log\Big(1+\frac{K}{\lambda}\Big)\Big)\bigg\} \le \frac{\delta}{3dH^2}.$$

Then with probability at least $1 - \delta/3dH^2$, for all $f_{\alpha_i}\in\mathcal{F}_{h+1}(\alpha_i)$, we have

$$\Big\|\sum_{k=1}^K\boldsymbol{\phi}_h^\tau\eta_h^\tau(f_{\alpha_i})\Big\|_{\boldsymbol{\Lambda}_h^{-1}}^2\cdot\mathbb{1}\big\{\|f_{\alpha_i}\|_\infty\le R_{h+1}\big\}$$

$$\leq 4 \inf_{\epsilon > 0} \left\{ (R_{h+1} + 2\epsilon)^2 \left( 2 \log \left( \frac{3dH^2 |\mathcal{N}_{h+1}(\epsilon)| |\mathcal{N}_{[0,H]}(\epsilon)|}{\delta} \right) + d \log \left( 1 + \frac{K}{\lambda} \right) \right) + \frac{6\epsilon^2 K^2}{\lambda} \right\}.$$

By Lemma G.4 and Lemma G.5 together with $D_0 = H\sqrt{Kd/\lambda}$ and $B_0 = 4\sqrt{d}H\sqrt{\iota}$, setting $\epsilon = d^{3/2}H^2/(K^{3/2}\sqrt{\kappa})$ and $K \geq \sqrt{d}H/(32\sqrt{\kappa}\iota)$, we have $\log |\mathcal{N}_{h+1}(\epsilon)| \leq 2d^2 \log(512K^3\iota/d^{3/2}H^2)$. Thus, we have

$$\left\| \sum_{k=1}^K \phi_h^\tau \eta_h^\tau(f_{\alpha_i}) \right\|_{\mathbf{\Lambda}_h^{-1}}^2 \cdot \mathbb{1}\left\{ \|f_{\alpha_i}\|_\infty \leq R_{h+1} \right\} \leq \frac{512dH^4\iota}{K\kappa} \left( 2\log \frac{2dH^2}{\delta} + 4d^2 \log \frac{512K^3\iota}{d^{3/2}H^2} \right)$$

$$\leq \frac{20480d^3H^4\iota^2}{K\kappa}.$$

Then, with a union bound over $i \in [d]$, we have

$$\mathbb{P}\left( \sup_{i \in [d]} \left\| \sum_{\tau=1}^K \phi_h^\tau \eta_h^\tau([\widehat{V}_{h+1}^\rho]_{\alpha_i} - [V_{h+1}^{\star,\rho}]_{\alpha_i}) \right\|_{\mathbf{\Lambda}_h^{-1}} > \frac{143d^{3/2}H^2\iota}{\sqrt{K\kappa}} \right)$$

$$\leq \mathbb{P}\left( \sup_{i \in [d]} \left\| \sum_{\tau=1}^K \phi_h^\tau \eta_h^\tau([\widehat{V}_{h+1}^\rho]_{\alpha_i} - [V_{h+1}^{\star,\rho}]_{\alpha_i}) \right\|_{\mathbf{\Lambda}_h^{-1}} \mathbb{1}\left\{ \left\| [\widehat{V}_{h+1}^\rho]_{\alpha_i} - [V_{h+1}^{\star,\rho}]_{\alpha_i} \right\|_\infty \leq R_{h+1} \right\} \right.$$

$$\left. > \frac{143d^{3/2}H^2\iota}{\sqrt{K\kappa}} \right) + \mathbb{P}\left( \mathbb{1}\left\{ \left\| [\widehat{V}_{h+1}^\rho]_{\alpha_i} - [V_{h+1}^{\star,\rho}]_{\alpha_i} \right\|_\infty > R_{h+1} \right\} \right)$$

$$\leq \frac{\delta}{3H^2} + \delta_{h+1},$$

which implies with probability at least $1 - \delta/3H^2 - \delta_{h+1}$, the term iv at step $h$ can be bounded as

$$\text{iv} \leq \frac{143d^{3/2}H^2\iota}{\sqrt{K\kappa}} \sum_{i=1}^d \|\phi_i(s,a)\mathbf{1}_i\|_{\mathbf{\Lambda}_h^{-1}}. \tag{G.17}$$

Then by a union bound over (G.16) and (G.17), if $K > 20449d^2H^2/\kappa$, then with probability at least $1 - 2\delta/3H^2 - \delta_{h+1}$ we have

$$\left| \inf_{P_h(\cdot|s,a) \in \mathcal{U}_h^\rho(s,a;\boldsymbol{\mu}_{h,i}^0)} [\mathbb{P}_h \widehat{V}_{h+1}^\rho](s,a) - \widehat{\inf_{P_h(\cdot|s,a) \in \mathcal{U}_h^\rho(s,a;\boldsymbol{\mu}_{h,i}^0)}} [\mathbb{P}_h \widehat{V}_{h+1}^\rho](s,a) \right|$$

$$\leq 4\sqrt{d}H\sqrt{\iota} \sum_{i=1}^d \|\phi_i(s,a)\mathbf{1}_i\|_{\mathbf{\Lambda}_h^{-1}} := \Gamma_h(s,a). \tag{G.18}$$

Further, when $K > \max\{512 \log(3H^2/\delta)/\kappa^2, 4/\kappa\}$, by Lemma I.3, with probability at least $1 - \delta/3H^2$, we have

$$\Gamma_h(s,a) \leq 4\sqrt{d}H\sqrt{\iota} \sum_{i=1}^d \|\phi_i(s,a)\mathbf{1}_i\|_{\mathbf{\Lambda}_h^{-1}} \leq \frac{8\sqrt{d}H\sqrt{\iota}}{\sqrt{K\kappa}}. \tag{G.19}$$

Then by a union bound over (G.18) and (G.19), under the event $\mathcal{E}_{h+1}$, with probability at least $1 - \delta/H^2 - \delta_{h+1}$, we have

$$V_h^{\star,\rho}(s) - \widehat{V}_h^\rho(s)$$

$$= Q_h^{\star,\rho}(s, \pi^\star(s)) - \widehat{Q}_h^\rho(s, \pi^\star(s)) + \widehat{Q}_h^\rho(s, \pi^\star(s)) - \widehat{Q}_h^\rho(s, \hat{\pi}(s))$$

$$\leq \inf_{P_h(\cdot|s,a) \in \mathcal{U}_h^\rho(s,a;\boldsymbol{\mu}_{h,i}^0)} [\mathbb{P}_h V_{h+1}^{\star,\rho}](s, \pi^\star(s)) - \widehat{\inf_{P_h(\cdot|s,a) \in \mathcal{U}_h^\rho(s,a;\boldsymbol{\mu}_{h,i}^0)}} [\mathbb{P}_h \widehat{V}_{h+1}^\rho](s,a) + \Gamma_h(s, \pi^\star(s))$$

$$= \inf_{P_h(\cdot|s,a) \in \mathcal{U}_h^\rho(s,a;\boldsymbol{\mu}_{h,i}^0)} [\mathbb{P}_h V_{h+1}^{\star,\rho}](s, \pi^\star(s)) - \inf_{P_h(\cdot|s,a) \in \mathcal{U}_h^\rho(s,a;\boldsymbol{\mu}_{h,i}^0)} [\mathbb{P}_h \widehat{V}_{h+1}^\rho](s, \pi^\star(s))$$

$$+ \inf_{P_h(\cdot|s,a) \in \mathcal{U}_h^\rho(s,a;\boldsymbol{\mu}_{h,i}^0)} [\mathbb{P}_h \widehat{V}_{h+1}^\rho](s, \pi^\star(s)) - \widehat{\inf_{P_h(\cdot|s,a) \in \mathcal{U}_h^\rho(s,a;\boldsymbol{\mu}_{h,i}^0)}} [\mathbb{P}_h \widehat{V}_{h+1}^\rho](s,a) + \Gamma_h(s, \pi^\star(s))$$

$$\leq R_{h+1} + 2\Gamma_h(s, \pi^\star(s)) \tag{G.20}$$

$$\leq \frac{16\sqrt{d}H(H-h)\sqrt{\iota}}{\sqrt{K\kappa}} + \frac{16\sqrt{d}H\sqrt{\iota}}{\sqrt{K\kappa}}$$

$$= \frac{16\sqrt{d}H(H-h+1)\sqrt{\iota}}{\sqrt{K\kappa}} := R_h,$$

where (G.20) holds by the following argument

$$\inf_{P_h(\cdot|s,a)\in\mathcal{U}_h^\rho(s,a;\boldsymbol{\mu}_{h,i}^0)}[\mathbb{P}_h V_{h+1}^{\star,\rho}](s,\pi^\star(s)) - \inf_{P_h(\cdot|s,a)\in\mathcal{U}_h^\rho(s,a;\boldsymbol{\mu}_{h,i}^0)}[\mathbb{P}_h\widehat{V}_{h+1}^\rho](s,\pi^\star(s))$$

$$\leq [\hat{\mathbb{P}}_h V_{h+1}^{\star,\rho}](s,\pi^\star(s)) - [\hat{\mathbb{P}}_h\widehat{V}_{h+1}^\rho](s,\pi^\star(s))$$

$$\leq \sup_s |V_{h+1}^{\star,\rho}(s) - \widehat{V}_{h+1}^\rho(s)|$$

$$\leq R_{h+1}, \tag{G.21}$$

where $\hat{P}_h(\cdot|s,a) = \arg\inf_{P_h(\cdot|s,a)\in\mathcal{U}_h^\rho(s,a;\boldsymbol{\mu}_{h,i}^0)}[\mathbb{P}_h\widehat{V}_{h+1}^\rho](s,a), \forall(s,a)\in\mathcal{S}\times\mathcal{A}$, and (G.21) is due to the induction assumption (G.15). Finally, denote

$$\mathcal{E}_h = \{0 \leq V_{h+1}^{\star,\rho}(s) - \widehat{V}_{h+1}^\rho(s) \leq R_h, \forall s\in\mathcal{S}\},$$

then we have $P(\mathcal{E}_h) \leq \delta_{h+1} + \delta/H^2 := \delta_h$.

**Generalization.** By induction and a union bound over $h\in[H]$, setting

$$\Gamma_h(s,a) = 4\sqrt{d}H\sqrt{\iota}\sum_{i=1}^d \|\phi_i(s,a)\mathbf{1}_i\|_{\boldsymbol{\Lambda}_h^{-1}},$$

then with probability at least $1 - (\delta/H^2 + 2\delta/H^2 + \cdots + H\delta/H^2) = 1 - dH(H+1)\delta/2H^2 > 1-\delta$, for all $(s,a,h)\in\mathcal{S}\times\mathcal{A}\times[H]$, we have

$$\left|\inf_{P_h(\cdot|s,a)\in\mathcal{U}_h^\rho(s,a;\boldsymbol{\mu}_{h,i}^0)}[\mathbb{P}_h\widehat{V}_{h+1}^\rho](s,a) - \widehat{\inf_{P_h(\cdot|s,a)\in\mathcal{U}_h^\rho(s,a;\boldsymbol{\mu}_{h,i}^0)}}[\mathbb{P}_h\widehat{V}_{h+1}^\rho](s,a)\right| \leq \Gamma_h(s,a).$$

This concludes the proof. $\qquad\square$

## G.3 Proof of Lemma E.1

*Proof.* Note that the conditional variance estimation does not involve any element of model uncertainty, and thus the proof follows from Lemma 5 of [54]. Recall that we estimate $[\mathbb{V}_h[V_{h+1}^\rho]_\alpha](s,a)$ based on $\mathcal{D}'$ as

$$\widehat{\sigma}_h^2(s,a;\alpha) = \max\left\{1, \left[\phi(s,a)^\top\tilde{\beta}_{h,2}(\alpha)\right]_{[0,H^2]} - \left[\phi(s,a)^\top\tilde{\beta}_{h,1}(\alpha)\right]_{[0,H]}^2 - \tilde{O}\left(\frac{\sqrt{d}H^3}{\sqrt{K\kappa}}\right)\right\}.$$

Note that

$$\left|\left[\phi(s,a)^\top\tilde{\beta}_{h,2}(\alpha)\right]_{[0,H^2]} - \left[\phi(s,a)^\top\tilde{\beta}_{h,1}(\alpha)\right]_{[0,H]}^2 - \mathbb{P}_h[\widehat{V}_{h+1}^{'\rho}]_\alpha^2](s,a) - ([\mathbb{P}_h[\widehat{V}_{h+1}^{'\rho}]_\alpha](s,a))^2\right|$$

$$\leq \left|\left[\phi(s,a)^\top\tilde{\beta}_{h,2}(\alpha)\right]_{[0,H^2]} - \mathbb{P}_h[\widehat{V}_{h+1}^{'\rho}]_\alpha^2](s,a)\right| + \left|\left[\phi(s,a)^\top\tilde{\beta}_{h,1}(\alpha)\right]_{[0,H]}^2 - ([\mathbb{P}_h[\widehat{V}_{h+1}^{'\rho}]_\alpha](s,a))^2\right|$$

$$\leq \underbrace{\left|\phi(s,a)^\top\tilde{\beta}_{h,2}(\alpha) - \mathbb{P}_h[\widehat{V}_{h+1}^{'\rho}]_\alpha^2](s,a)\right|}_{\text{i}} + 2H\underbrace{\left|\phi(s,a)^\top\tilde{\beta}_{h,1}(\alpha) - [\mathbb{P}_h[\widehat{V}_{h+1}^{'\rho}]_\alpha](s,a)\right|}_{\text{ii}}.$$

Note that the estimation error i and ii both come from regular ridge regressions with targets $[\widehat{V}_{h+1}^{'\rho}(s)]_\alpha^2$ and $[\widehat{V}_{h+1}^{'\rho}(s)]_\alpha$, respectively. Thus, the analysis is standard and for simplicity we omit the details here and focus on the results: with probability at least $1 - \delta/2$, we have

$$\left|\left[\phi(s,a)^\top\tilde{\beta}_{h,2}(\alpha)\right]_{[0,H^2]} - \left[\phi(s,a)^\top\tilde{\beta}_{h,1}(\alpha)\right]_{[0,H]}^2 - \mathbb{P}_h[\widehat{V}_{h+1}^{'\rho}]_\alpha^2](s,a) - ([\mathbb{P}_h[\widehat{V}_{h+1}^{'\rho}]_\alpha](s,a))^2\right|$$

$$\leq \tilde{O}\left(\frac{dH^2}{\sqrt{K\kappa}}\right). \tag{G.22}$$

Then by [Theorem 4.4](#) and [Lemma I.3](#), for all $(s, a, h) \in \mathcal{S} \times \mathcal{A} \times [H]$, with probability at least $1 - \delta/2$, we have

$$
\begin{aligned}
&\left| [\mathrm{Var}_h[\widehat{V}'^{\rho}_{h+1}]_\alpha](s, a) - [\mathrm{Var}_h[V^{\star,\rho}_{h+1}]_\alpha](s, a) \right| \\
&\leq \left| [\mathbb{P}_h[\widehat{V}'^{\rho}_{h+1}]^2_\alpha](s, a) - [\mathbb{P}_h[V^{\star,\rho}_{h+1}]^2_\alpha](s, a) \right| + \left| \left([\mathbb{P}_h[\widehat{V}'^{\rho}_{h+1}]_\alpha](s, a)\right)^2 - \left([\mathbb{P}_h[V^{\star,\rho}_{h+1}]_\alpha](s, a)\right)^2 \right| \\
&\leq 2H \left| [\mathbb{P}_h([\widehat{V}^{\rho}_{h+1}]_\alpha - [V^{\star,\rho}_{h+1}]_\alpha)](s, a) \right| + 2H \left| [\mathbb{P}_h([V^{\star,\rho}_{h+1}]_\alpha - [V^{\star,\rho}_{h+1}]_\alpha)](s, a) \right| \\
&\leq \tilde{O}\left(\frac{\sqrt{d}H^3}{\sqrt{K\kappa}}\right). \tag{G.23}
\end{aligned}
$$

By [(G.22)](#) and [(G.23)](#) and a union bound, we know that with probability at least $1 - \delta$, we have

$$
\begin{aligned}
&\left| \left[\phi(s, a)^\top \tilde{\beta}_{h,2}(\alpha)\right]_{[0,H^2]} - \left[\phi(s, a)^\top \tilde{\beta}_{h,1}(\alpha)\right]^2_{[0,H]} - [\mathrm{Var}_h[V^{\star,\rho}_{h+1}]_\alpha](s, a) \right| \\
&\leq \left| \left[\phi(s, a)^\top \tilde{\beta}_{h,2}(\alpha)\right]_{[0,H^2]} - \left[\phi(s, a)^\top \tilde{\beta}_{h,1}(\alpha)\right]^2_{[0,H]} - [\mathrm{Var}_h[\widehat{V}'^{\rho}_{h+1}]_\alpha](s, a) \right| \\
&\quad + \left| [\mathrm{Var}_h[\widehat{V}'^{\rho}_{h+1}]_\alpha](s, a) - [\mathrm{Var}_h[V^{\star,\rho}_{h+1}]_\alpha](s, a) \right| \\
&\leq \tilde{O}\left(\frac{dH^3}{\sqrt{K\kappa}}\right),
\end{aligned}
$$

which implies that

$$
\left[\phi(s, a)^\top \tilde{\beta}_{h,2}(\alpha)\right]_{[0,H^2]} - \left[\phi(s, a)^\top \tilde{\beta}_{h,1}(\alpha)\right]^2_{[0,H]} - \tilde{O}\left(\frac{dH^3}{\sqrt{K\kappa}}\right) \leq [\mathrm{Var}_h[V^{\star,\rho}_{h+1}]_\alpha](s, a).
$$

By the fact that the operator $\min\{1, \cdot\}$ is order preserving, thus we have

$$
\widehat{\sigma}^2_h(s, a; \alpha) \leq [\mathbb{V}_h[V^{\star,\rho}_{h+1}]_\alpha](s, a).
$$

Further, by the fact that the operator $\min\{1, \cdot\}$ is a contraction map, [(G.22)](#) and [(G.23)](#), we have

$$
\begin{aligned}
&\left| \widehat{\sigma}^2_h(s, a; \alpha) - [\mathbb{V}_h[V^{\star,\rho}_{h+1}]_\alpha](s, a) \right| \\
&\leq \left| \widehat{\sigma}^2_h(s, a; \alpha) - [\mathbb{V}_h[\widehat{V}'^{\rho}_{h+1}]_\alpha](s, a) \right| + \left| [\mathbb{V}_h[\widehat{V}'^{\rho}_{h+1}]_\alpha](s, a) - [\mathbb{V}_h[V^{\star,\rho}_{h+1}]_\alpha](s, a) \right| \\
&\leq \left| \left[\phi(s, a)^\top \tilde{\beta}_{h,2}(\alpha)\right]_{[0,H^2]} - \left[\phi(s, a)^\top \tilde{\beta}_{h,1}(\alpha)\right]^2_{[0,H]} - \tilde{O}\left(\frac{dH^3}{\sqrt{K\kappa}}\right) - [\mathrm{Var}_h[\widehat{V}'^{\rho}_{h+1}]_\alpha](s, a) \right| \\
&\quad + \left| [\mathrm{Var}_h[\widehat{V}'^{\rho}_{h+1}]_\alpha](s, a) - [\mathrm{Var}_h[V^{\star,\rho}_{h+1}]_\alpha](s, a) \right| \\
&\leq \tilde{O}\left(\frac{dH^2}{\sqrt{K\kappa}}\right) + \tilde{O}\left(\frac{dH^3}{\sqrt{K\kappa}}\right) + \tilde{O}\left(\frac{\sqrt{d}H^3}{\sqrt{K\kappa}}\right) \\
&= \tilde{O}\left(\frac{dH^3}{\sqrt{K\kappa}}\right).
\end{aligned}
$$

This concludes the proof. $\qquad\square$

## G.4 Proof of [Lemma E.2](#)

*Proof.* Note that the reference-advantage decomposition is exactly the same as that in the proof of [Lemma G.1](#), thus we have

$$
\begin{aligned}
&\inf_{P_h(\cdot|s,a) \in \mathcal{U}^\rho_h(s, a; \boldsymbol{\mu}^0_{h,i})} [\mathbb{P}_h \widehat{V}^{\rho}_{h+1}](s, a) - \widehat{\inf_{P_h(\cdot|s,a) \in \mathcal{U}^\rho_h(s, a; \boldsymbol{\mu}^0_{h,i})}} [\mathbb{P}_h \widehat{V}^{\rho}_{h+1}](s, a) \\
&\leq \underbrace{\sum_{i=1}^{d} \phi_i(s, a) \mathbf{1}^\top_i \left( \mathbb{E}^{\boldsymbol{\mu}^0_h}[V^{\star,\rho}_{h+1}(s)]_{\alpha_i} - \widehat{\mathbb{E}}^{\boldsymbol{\mu}^0_h}[V^{\star,\rho}_{h+1}(s)]_{\alpha_i} \right)}_{\text{reference uncertainty}} \\
&\quad + \underbrace{\sum_{i=1}^{d} \phi_i(s, a) \mathbf{1}^\top_i \left( \mathbb{E}^{\boldsymbol{\mu}^0_h}\left[[\widehat{V}^{\rho}_{h+1}(s)]_{\alpha_i} - [V^{\star,\rho}_{h+1}(s)]_{\alpha_i}\right] - \widehat{\mathbb{E}}^{\boldsymbol{\mu}^0_h}\left[[\widehat{V}^{\rho}_{h+1}(s)]_{\alpha_i} - [V^{\star,\rho}_{h+1}(s)]_{\alpha_i}\right] \right)}_{\text{advantage uncertainty}}.
\end{aligned}
$$

Next, we further decompose the reference uncertainty and the advantage uncertainty, respectively.

**The Reference Uncertainty.**    Specifically, we have

$$\sum_{i=1}^{d} \phi_i(s,a)\mathbf{1}_i^\top \left( \mathbb{E}^{\boldsymbol{\mu}_h^0}[V_{h+1}^{\star,\rho}(s)]_{\alpha_i} - \widehat{\mathbb{E}}^{\boldsymbol{\mu}_h^0}[V_{h+1}^{\star,\rho}(s)]_{\alpha_i} \right)$$

$$= \sum_{i=1}^{d} \phi_i(s,a)\mathbf{1}_i^\top \left( \mathbb{E}^{\boldsymbol{\mu}_h^0}[V_{h+1}^{\star,\rho}(s)]_{\alpha_i} - \boldsymbol{\Sigma}_h^{-1}(\alpha_i)\sum_{\tau=1}^{K} \frac{\boldsymbol{\phi}_h^\tau [\mathbb{P}_h^0[V_{h+1}^{\star,\rho}]_{\alpha_i}](s_h^\tau,a_h^\tau)}{\widehat{\sigma}_h^2(s_h^\tau,a_h^\tau;\alpha_i)} \right.$$

$$\left. + \boldsymbol{\Sigma}_h^{-1}(\alpha_i)\sum_{\tau=1}^{K} \frac{\boldsymbol{\phi}_h^\tau [\mathbb{P}_h^0[V_{h+1}^{\star,\rho}]_{\alpha_i}](s_h^\tau,a_h^\tau)}{\widehat{\sigma}_h^2(s_h^\tau,a_h^\tau;\alpha_i)} - \boldsymbol{\Sigma}_h^{-1}(\alpha_i)\sum_{\tau=1}^{K} \frac{\boldsymbol{\phi}_h^\tau [V_{h+1}^{\star,\rho}(s_{h+1}^\tau)]_{\alpha_i}}{\widehat{\sigma}_h^2(s_h^\tau,a_h^\tau;\alpha_i)} \right)$$

$$= \lambda\sum_{i=1}^{d} \phi_i(s,a)\mathbf{1}_i^\top \boldsymbol{\Sigma}_h^{-1}(\alpha_i)\mathbb{E}^{\boldsymbol{\mu}_h^0}[V_{h+1}^{\star,\rho}(s)]_{\alpha_i} + \sum_{i=1}^{d} \phi_i(s,a)\mathbf{1}_i^\top \boldsymbol{\Sigma}_h^{-1}(\alpha_i)\sum_{\tau=1}^{K} \frac{\boldsymbol{\phi}_h^\tau \eta_h^\tau([V_{h+1}^{\star,\rho}]_{\alpha_i})}{\widehat{\sigma}_h^2(s_h^\tau,a_h^\tau;\alpha_i)}$$

$$\leq \underbrace{\lambda\sum_{i=1}^{d} \|\phi_i(s,a)\mathbf{1}_i\|_{\boldsymbol{\Sigma}_h^{-1}(\alpha_i)}\|\mathbb{E}^{\boldsymbol{\mu}_h^0}[V_{h+1}^{\star,\rho}(s)]_{\alpha_i}\|_{\boldsymbol{\Sigma}_h^{-1}(\alpha_i)}}_{\text{i}}$$

$$+ \underbrace{\sum_{i=1}^{d} \|\phi_i(s,a)\mathbf{1}_i\|_{\boldsymbol{\Sigma}_h^{-1}(\alpha_i)}\left\| \sum_{\tau=1}^{K} \frac{\boldsymbol{\phi}_h^\tau \eta_h^\tau([V_{h+1}^{\star,\rho}]_{\alpha_i})}{\widehat{\sigma}_h^2(s_h^\tau,a_h^\tau;\alpha_i)} \right\|_{\boldsymbol{\Sigma}_h^{-1}(\alpha_i)}}_{\text{ii}}.$$

**The Advantage Uncertainty.**    Similar to the argument in decomposing the reference uncertainty, we have

$$\sum_{i=1}^{d} \phi_i(s,a)\mathbf{1}_i^\top \left( \mathbb{E}^{\boldsymbol{\mu}_h^0}\left[[\widehat{V}_{h+1}^{\rho}(s)]_{\alpha_i} - [V_{h+1}^{\star,\rho}(s)]_{\alpha_i}\right] - \widehat{\mathbb{E}}^{\boldsymbol{\mu}_h^0}\left[[\widehat{V}_{h+1}^{\rho}(s)]_{\alpha_i} - [V_{h+1}^{\star,\rho}(s)]_{\alpha_i}\right] \right)$$

$$\leq \underbrace{\lambda\sum_{i=1}^{d} \|\phi_i(s,a)\mathbf{1}_i\|_{\boldsymbol{\Sigma}_h^{-1}(\alpha_i)}\left\| \mathbb{E}^{\boldsymbol{\mu}_h^0}\left[[\widehat{V}_{h+1}^{\rho}(s)]_{\alpha_i} - [V_{h+1}^{\star,\rho}(s)]_{\alpha_i}\right] \right\|_{\boldsymbol{\Sigma}_h^{-1}(\alpha_i)}}_{\text{iii}}$$

$$+ \underbrace{\sum_{i=1}^{d} \|\phi_i(s,a)\mathbf{1}_i\|_{\boldsymbol{\Sigma}_h^{-1}(\alpha_i)}\left\| \sum_{\tau=1}^{K} \frac{\boldsymbol{\phi}_h^\tau \eta_h^\tau([\widehat{V}_{h+1}^{\rho}(s)]_{\alpha_i} - [V_{h+1}^{\star,\rho}(s)]_{\alpha_i})}{\widehat{\sigma}_h^2(s_h^\tau,a_h^\tau;\alpha_i)} \right\|_{\boldsymbol{\Sigma}_h^{-1}(\alpha_i)}}_{\text{iv}}.$$

Put terms i-iv together, we have

$$\inf_{P_h(\cdot|s,a)\in\mathcal{U}_h^\rho(s,a;\boldsymbol{\mu}_{h,i}^0)}[\mathbb{P}_h\widehat{V}_{h+1}^{\rho}](s,a) - \widehat{\inf_{P_h(\cdot|s,a)\in\mathcal{U}_h^\rho(s,a;\boldsymbol{\mu}_{h,i}^0)}}[\mathbb{P}_h\widehat{V}_{h+1}^{\rho}](s,a)$$

$$\leq \underbrace{\lambda\sum_{i=1}^{d} \|\phi_i(s,a)\mathbf{1}_i\|_{\boldsymbol{\Sigma}_h^{-1}(\alpha_i)}\|\mathbb{E}^{\boldsymbol{\mu}_h^0}[V_{h+1}^{\star,\rho}(s)]_{\alpha_i}\|_{\boldsymbol{\Sigma}_h^{-1}(\alpha_i)}}_{\text{i}}$$

$$+ \underbrace{\sum_{i=1}^{d} \|\phi_i(s,a)\mathbf{1}_i\|_{\boldsymbol{\Sigma}_h^{-1}(\alpha_i)}\left\| \sum_{\tau=1}^{K} \frac{\boldsymbol{\phi}_h^\tau \eta_h^\tau([V_{h+1}^{\star,\rho}]_{\alpha_i})}{\widehat{\sigma}_h^2(s_h^\tau,a_h^\tau;\alpha_i)} \right\|_{\boldsymbol{\Sigma}_h^{-1}(\alpha_i)}}_{\text{ii}}$$

$$+ \underbrace{\lambda\sum_{i=1}^{d} \|\phi_i(s,a)\mathbf{1}_i\|_{\boldsymbol{\Sigma}_h^{-1}(\alpha_i)}\left\| \mathbb{E}^{\boldsymbol{\mu}_h^0}\left[[\widehat{V}_{h+1}^{\rho}(s)]_{\alpha_i} - [V_{h+1}^{\star,\rho}(s)]_{\alpha_i}\right] \right\|_{\boldsymbol{\Sigma}_h^{-1}(\alpha_i)}}_{\text{iii}}$$

$$+ \underbrace{\sum_{i=1}^{d} \|\phi_i(s,a)\mathbf{1}_i\|_{\boldsymbol{\Sigma}_h^{-1}(\alpha_i)}\left\| \sum_{\tau=1}^{K} \frac{\boldsymbol{\phi}_h^\tau \eta_h^\tau([\widehat{V}_{h+1}^{\rho}(s)]_{\alpha_i} - [V_{h+1}^{\star,\rho}(s)]_{\alpha_i})}{\widehat{\sigma}_h^2(s_h^\tau,a_h^\tau;\alpha_i)} \right\|_{\boldsymbol{\Sigma}_h^{-1}(\alpha_i)}}_{\text{iv}}.$$

By similar argument as Lemma G.1, we know there exist $\{\tilde{\alpha}_i\}_{i \in [d]}$ such that

$$
\Bigg| \inf_{P_h(\cdot|s,a) \in \mathcal{U}_h^\rho(s,a;\boldsymbol{\mu}_{h,i}^0)} [\mathbb{P}_h \widehat{V}_{h+1}^\rho](s,a) - \widehat{\inf_{P_h(\cdot|s,a) \in \mathcal{U}_h^\rho(s,a;\boldsymbol{\mu}_{h,i}^0)}} [\mathbb{P}_h \widehat{V}_{h+1}^\rho](s,a) \Bigg|
$$

$$
\leq \lambda \underbrace{\sum_{i=1}^d \|\phi_i(s,a)\mathbf{1}_i\|_{\boldsymbol{\Sigma}_h^{-1}(\tilde{\alpha}_i)} \|\mathbb{E}^{\boldsymbol{\mu}_h^0}[V_{h+1}^{\star,\rho}(s)]_{\alpha_i}\|_{\boldsymbol{\Sigma}_h^{-1}(\tilde{\alpha}_i)}}_{\text{i}}
$$

$$
+ \underbrace{\sum_{i=1}^d \|\phi_i(s,a)\mathbf{1}_i\|_{\boldsymbol{\Sigma}_h^{-1}(\tilde{\alpha}_i)} \Big\| \sum_{\tau=1}^K \frac{\boldsymbol{\phi}_h^\tau \eta_h^\tau([V_{h+1}^{\star,\rho}]_{\alpha_i})}{\widehat{\sigma}_h^2(s_h^\tau, a_h^\tau; \alpha_i)} \Big\|_{\boldsymbol{\Sigma}_h^{-1}(\tilde{\alpha}_i)}}_{\text{ii}}
$$

$$
+ \lambda \underbrace{\sum_{i=1}^d \|\phi_i(s,a)\mathbf{1}_i\|_{\boldsymbol{\Sigma}_h^{-1}(\alpha_i)} \Big\| \mathbb{E}^{\boldsymbol{\mu}_h^0} \big[ [\widehat{V}_{h+1}^\rho(s)]_{\tilde{\alpha}_i} - [V_{h+1}^{\star,\rho}(s)]_{\alpha_i} \big] \Big\|_{\boldsymbol{\Sigma}_h^{-1}(\tilde{\alpha}_i)}}_{\text{iii}}
$$

$$
+ \underbrace{\sum_{i=1}^d \|\phi_i(s,a)\mathbf{1}_i\|_{\boldsymbol{\Sigma}_h^{-1}(\tilde{\alpha}_i)} \Big\| \sum_{\tau=1}^K \frac{\boldsymbol{\phi}_h^\tau \eta_h^\tau([\widehat{V}_{h+1}^\rho(s)]_{\tilde{\alpha}_i} - [V_{h+1}^{\star,\rho}(s)]_{\alpha_i})}{\widehat{\sigma}_h^2(s_h^\tau, a_h^\tau; \alpha_i)} \Big\|_{\boldsymbol{\Sigma}_h^{-1}(\tilde{\alpha}_i)}}_{\text{iv}}.
$$

This concludes the proof. $\qquad\square$

## G.5  Proof of Lemma E.3

*Proof.* By the robust bellman equation (3.1), we know

$$
V_h^{\pi,\rho}(s) = \mathbb{E}_{a \sim \pi(\cdot|s)} \Big[ r(s,a) + \inf_{P_h(\cdot|s,a) \in \mathcal{U}_h^\rho(s,a;\boldsymbol{\mu}_h^0)} [\mathbb{P}_h V_{h+1}^{\pi,\rho}](s,a) \Big]. \tag{G.24}
$$

Then, we can trivially bound $\max_{s \in \mathcal{S}} V_h^{\pi,\rho}(s)$ as

$$
\max_{s \in \mathcal{S}} V_h^{\pi,\rho}(s) \leq \max_{s,a} \Big( 1 + \inf_{P_h(\cdot|s,a) \in \mathcal{U}_h^\rho(s,a;\boldsymbol{\mu}_h^0)} [\mathbb{P}_h V_{h+1}^{\pi,\rho}](s,a) \Big). \tag{G.25}
$$

Further, by the definition of the $d$-rectangular uncertainty set, we have

$$
\inf_{P_h(\cdot|s,a) \in \mathcal{U}_h^\rho(s,a;\boldsymbol{\mu}_h^0)} [\mathbb{P}_h V_{h+1}^{\pi,\rho}](s,a) = \sum_{i=1}^d \phi_i(s,a) \inf_{\mu_{h,i} \in \mathcal{U}_{h,i}^\rho(\mu_{h,i}^0)} \mathbb{E}_{s \sim \mu_{h,i}}[V_{h+1}^{\pi,\rho}(s)]. \tag{G.26}
$$

Denoting $s_{\max} = \operatorname{argmax}_{s \in \mathcal{S}} V_{h+1}^{\pi,\rho}(s)$ and $s_{\min} = \operatorname{argmin}_{s \in \mathcal{S}} V_{h+1}^{\pi,\rho}(s)$, and for all $i \in [d]$, we construct a distribution $\breve{\mu}_{h,i} = (1-\rho)\mu_{h,i} + \rho\delta_{s_{\min}}$, where $\delta_x$ is the Dirac Delta distribution with mass on $x$. Note that $\breve{\mu}_{h,i} \in \mathcal{U}_{h,i}^\rho(\mu_{h,i}^0)$, thus we have

$$
\inf_{\mu_{h,i} \in \mathcal{U}_{h,i}^\rho(\mu_{h,i}^0)} \mathbb{E}_{s \sim \mu_{h,i}}[V_{h+1}^{\pi,\rho}(s)] \leq \mathbb{E}_{s \sim \breve{\mu}_{h,i}}[V_{h+1}^{\pi,\rho}(s)] \leq (1-\rho) \max_{s \in \mathcal{S}} V_{h+1}^{\pi,\rho}(s) + \rho \min_s V_{h+1}^{\pi,\rho}(s). \tag{G.27}
$$

Combining (G.25), (G.26) and (G.27), we have

$$
\max_{s \in \mathcal{S}} V_h^{\pi,\rho}(s) \leq (1-\rho) \max_{s \in \mathcal{S}} V_{h+1}^{\pi,\rho}(s) + \rho \min_{s \in \mathcal{S}} V_{h+1}^{\pi,\rho}(s) + 1. \tag{G.28}
$$

On the other hand, by (G.24), we can trivially bound $\min_s V_h^{\pi,\rho}(s)$ as

$$
\min_s V_h^{\pi,\rho}(s) \geq \min_{s,a} \inf_{P_h(\cdot|s,a) \in \mathcal{U}_h^\rho(s,a;\boldsymbol{\mu}_h^0)} [\mathbb{P}_h V_{h+1}^{\pi,\rho}](s,a). \tag{G.29}
$$

By the fact that

$$
\inf_{\mu_{h,i} \in \mathcal{U}_{h,i}^\rho(\mu_{h,i}^0)} \mathbb{E}_{s \sim \mu_{h,i}}[V_{h+1}^{\pi,\rho}(s)] \geq \min_{s \in \mathcal{S}} V_{h+1}^{\pi,\rho}(s), \tag{G.30}
$$

combining (G.26), (G.29) and (G.30), we have

$$\min_s V_h^{\pi,\rho}(s) \geq \min_{s \in \mathcal{S}} V_{h+1}^{\pi,\rho}(s). \tag{G.31}$$

For any $h \in [H]$, by (G.28) and (G.31), we have

$$\max_{s \in \mathcal{S}} V_h^{\pi,\rho}(s) - \min_{s \in \mathcal{S}} V_h^{\pi,\rho}(s)$$
$$\leq 1 + (1-\rho) \max_{s \in \mathcal{S}} V_{h+1}^{\pi,\rho}(s) - \min_{s \in \mathcal{S}} V_{h+1}^{\pi,\rho}(s) + \rho \min_{s \in \mathcal{S}} V_{h+1}^{\pi,\rho}(s)$$
$$= 1 + (1-\rho) \Big[ \max_{s \in \mathcal{S}} V_{h+1}^{\pi,\rho}(s) - \min_{s \in \mathcal{S}} V_{h+1}^{\pi,\rho}(s) \Big]. \tag{G.32}$$

For step $H$, by the definition of the value function, we have $0 \leq V_H^{\pi,\rho}(s) \leq 1, \forall s \in \mathcal{S}$. Applying (G.32) with $h = H - 1$ leads to $\max_{s \in \mathcal{S}} V_{H-1}^{\pi,\rho}(s) - \min_{s \in \mathcal{S}} V_{H-1}^{\pi,\rho}(s) \leq 1 + (1-\rho) \cdot 1$. We finish the proof by recursively applying (G.32). $\qquad \square$

### G.6   Proof of Lemma F.1

*Proof.* The proof of Lemma F.1 consists of the following two steps:

**Step 1: lower bound the suboptimality by Hamming distance.**   For any $\boldsymbol{\xi} \in \{-1, 1\}^{dH}$, denote $V_{\boldsymbol{\xi}}^{\star,\rho}(s)$ as the optimal robust value function for the MDP instance $M_{\boldsymbol{\xi}}$. For any function $\pi$, denote $V_{\boldsymbol{\xi}}^{\pi,\rho}$ as the robust value function corresponding to a policy $\pi$. Then by definition, we have

$$V_{\boldsymbol{\xi}}^{\star,\rho}(x_1) = \max_{\pi} \inf_{P \in \mathcal{U}^\rho(P^0)} \mathbb{E}^{\pi,P} \big[ r_1(s_1, a_1) + \cdots + r_H(s_H, a_H) | s_1 = x_1 \big],$$
$$V_{\boldsymbol{\xi}}^{\pi,\rho}(x_1) = \inf_{P \in \mathcal{U}^\rho(P^0)} \mathbb{E}^{\pi,P} \big[ r_1(s_1, a_1) + \cdots + r_H(s_H, a_H) | s_1 = x_1 \big].$$

For any given $\boldsymbol{\xi}$, the optimal action at step $h$ is

$$a_h^\star = ((1 + \xi_{h1})/2, \cdots, (1 + \xi_{hd})/2).$$

The worst case transition at the first step is known as

$$\mathbb{P}_1(x_1|x_1, a) = (1 - \rho), \ \mathbb{P}_1(x_2|x_1, a) = \rho, \ \mathbb{P}_1(x_2|x_2, a) = 1, \ \forall a \in \mathcal{A},$$

and from the second step on, the state always stays at $s_2$. With these facts in mind, we have

$$V_{\boldsymbol{\xi}}^{\star,\rho}(x_1)$$
$$= \delta \Big\{ \Big[ \frac{1}{2} + \sum_{i=1}^d \frac{1 + \xi_{1i}}{4d} \Big] + (1-\rho) \Big[ \frac{1}{2} + \sum_{i=1}^d \frac{1 + \xi_{2i}}{4d} \Big] + \cdots + (1-\rho) \Big[ \frac{1}{2} + \sum_{i=1}^d \frac{1 + \xi_{Hi}}{4d} \Big] \Big\}$$
$$= \frac{\delta}{2d} \Big\{ \Big[ d + \sum_{i=1}^d \frac{1 + \xi_{1i}}{2} \Big] + (1-\rho) \Big[ d + \sum_{i=1}^d \frac{1 + \xi_{2i}}{2} \Big] + \cdots + (1-\rho) \Big[ d + \sum_{i=1}^d \frac{1 + \xi_{Hi}}{2} \Big] \Big\},$$

and

$$V_{\boldsymbol{\xi}}^{\pi,\rho}(x_1)$$
$$= \frac{\delta}{2d} \mathbb{E}^\pi \Big\{ \Big[ d + \sum_{i=1}^d \xi_{1i} a_{1i} \Big] + (1-\rho) \Big[ d + \sum_{i=1}^d \xi_{2i} a_{2i} \Big] \cdots + (1-\rho) \Big[ d + \sum_{i=1}^d \xi_{Hi} a_{Hi} \Big] \Big\}.$$

Then we have

$$V_{\boldsymbol{\xi}}^{\star,\rho}(x_1) - V_{\boldsymbol{\xi}}^{\pi,\rho}(x_1)$$
$$= \frac{\delta}{2d} \Big\{ \Big[ \sum_{i=1}^d \frac{1 + \xi_{1i}}{2} - \xi_{1i} \mathbb{E}^\pi a_{1i} \Big] + (1-\rho) \sum_{h=2}^H \sum_{i=1}^d \Big( \frac{1 + \xi_{hi}}{2} - \xi_{hi} \mathbb{E}^\pi a_{hi} \Big) \Big\}$$
$$\geq \frac{\delta}{2d} (1-\rho) \sum_{h=1}^H \sum_{i=1}^d \Big( \frac{1 + \xi_{hi}}{2} - \xi_{hi} \mathbb{E}^\pi a_{hi} \Big)$$

$$= \frac{\delta}{2d}(1-\rho)\sum_{h=1}^{H}\sum_{i=1}^{d}\left(\frac{1}{2} + \xi_{hi}\mathbb{E}^{\pi}\left(\frac{1}{2} - a_{hi}\right)\right)$$

$$= \frac{\delta}{4d}(1-\rho)\sum_{h=1}^{H}\sum_{i=1}^{d}(1 - \xi_{hi}\mathbb{E}^{\pi}(2a_{hi} - 1)). \tag{G.33}$$

Note that for any $(h,i) \in [H] \times [d]$, by design we have $1 = \xi_{hi}^2$, thus

$$\frac{\delta}{4d}(1-\rho)\sum_{h=1}^{H}\sum_{i=1}^{d}(1 - \xi_{hi}\mathbb{E}^{\pi}(2a_{hi} - 1)) = \frac{\delta}{4d}(1-\rho)\sum_{h=1}^{H}\sum_{i=1}^{d}(\xi_{hi} - \mathbb{E}^{\pi}(2a_{hi} - 1))\xi_{hi}$$

$$= \frac{\delta}{4d}(1-\rho)\sum_{h=1}^{H}\sum_{i=1}^{d}|\xi_{hi} - \mathbb{E}^{\pi}(2a_{hi} - 1)|, \tag{G.34}$$

where (G.34) holds due to the fact that $\mathbb{E}^{\pi}(2a_{hi} - 1) \in [-1, 1]$. To continue, we have

$$\frac{\delta}{4d}(1-\rho)\sum_{h=1}^{H}\sum_{i=1}^{d}|\xi_{hi} - \mathbb{E}^{\pi}(2a_{hi} - 1)|$$

$$\geq \frac{\delta}{4d}(1-\rho)\sum_{h=1}^{H}\sum_{i=1}^{d}|\xi_{hi} - \mathbb{E}^{\pi}(2a_{hi} - 1)|\,\mathbb{1}\{\xi_{hi} \neq \mathrm{sign}(\mathbb{E}^{\pi}(2a_{h,i} - 1))\}$$

$$\geq \frac{\delta}{4d}(1-\rho)\sum_{h=1}^{H}\sum_{i=1}^{d}\mathbb{1}\{\xi_{hi} \neq \mathrm{sign}(\mathbb{E}^{\pi}(2a_{h,i} - 1))\}$$

$$\geq \frac{\delta}{4d}(1-\rho)D_H(\boldsymbol{\xi}, \boldsymbol{\xi}^{\pi}), \tag{G.35}$$

where $D_H(\cdot, \cdot)$ is the Hamming distance, $\boldsymbol{\xi}^{\pi} = \{\boldsymbol{\xi}_h^{\pi}\}_{h \in [H]}$, and $\xi_{hi}^{\pi} := \mathrm{sign}(\mathbb{E}^{\pi}(2a_{hi} - 1)), \forall i \in [d]$. Combining (G.33), (G.34), (G.35) and the definition of the suboptimality gap, we have

$$\mathrm{SupOpt}(M_{\boldsymbol{\xi}}, x_1, \pi, \rho) \geq \frac{\delta}{4d}(1-\rho)D_H(\boldsymbol{\xi}, \boldsymbol{\xi}^{\pi}). \tag{G.36}$$

**Step 2: lower bound the hamming distance by testing error.** Applying Assouad's method [46, Lemma 2.12], we have

$$\inf_{\pi}\sup_{\boldsymbol{\xi} \in \Omega}\mathbb{E}_{\boldsymbol{\xi}}\left[D_H(\boldsymbol{\xi}, \boldsymbol{\xi}')\right] \geq \frac{dH}{2}\min_{\substack{\boldsymbol{\xi}, \boldsymbol{\xi}' \in \Omega \\ D_H(\boldsymbol{\xi}, \boldsymbol{\xi}')=1}}\inf_{\psi}\left[\mathbb{Q}_{\boldsymbol{\xi}}(\psi(\mathcal{D}) \neq \boldsymbol{\xi}) + \mathbb{Q}_{\boldsymbol{\xi}'}(\psi(\mathcal{D}) \neq \boldsymbol{\xi}')\right], \tag{G.37}$$

where $\inf_{\psi}$ denotes the infimum over all test functions taking values in $\{\boldsymbol{\xi}, \boldsymbol{\xi}'\}$. We conclude the proof by combining (G.36) and (G.37). $\qquad\square$

### G.7 Proof of Lemma F.2

*Proof.* By the Theorem 2.12 in [46], we lower bound the testing error as follows

$$\min_{\boldsymbol{\xi}, \boldsymbol{\xi}': D_H(\boldsymbol{\xi}, \boldsymbol{\xi}')=1}\inf_{\psi}\left[\mathbb{Q}_{\boldsymbol{\xi}}(\psi(\mathcal{D}) \neq \boldsymbol{\xi}) + \mathbb{Q}_{\boldsymbol{\xi}'}(\psi(\mathcal{D}) \neq \boldsymbol{\xi}')\right]$$

$$\geq 1 - \left(\frac{1}{2}\max_{\boldsymbol{\xi}, \boldsymbol{\xi}': D_H(\boldsymbol{\xi}, \boldsymbol{\xi}')=1}D_{\mathrm{KL}}\left(\mathbb{Q}_{\boldsymbol{\xi}}||\mathbb{Q}_{\boldsymbol{\xi}'}\right)\right)^{1/2},$$

where $D_{\mathrm{KL}}(\cdot||\cdot)$ is the Kullback-Leibler divergence. Then it remains to bound $D_{\mathrm{KL}}\left(\mathbb{Q}_{\boldsymbol{\xi}}||\mathbb{Q}_{\boldsymbol{\xi}'}\right)$. According to the definition of $\mathbb{Q}_{\boldsymbol{\xi}}(\mathcal{D})$, we have

$$\mathbb{Q}_{\boldsymbol{\xi}}(\mathcal{D}) = \prod_{k=1}^{K}\prod_{h=1}^{H}\pi_h^b(a_h^k|s_h^k)P_h(s_{h+1}^k|s_h^k, a_h^k)R(s_h^k, a_h^k; r_h^k),$$

where $R(s_h^k, a_h^k; r_h^k)$ is the density function of $\mathcal{N}(r_h(s_h^k, a_h^k), 1)$ at $r_h^k$. Note that the difference between the two distribution $\mathbb{Q}_{\boldsymbol{\xi}}(\mathcal{D})$ and $\mathbb{Q}_{\boldsymbol{\xi}'}(\mathcal{D})$ lies only in the reward distribution corresponding to the index where $\boldsymbol{\xi}$ and $\boldsymbol{\xi}'$ differ. Then, by the chain rule of Kullback-Leibler divergence, we have

$$D_{\mathrm{KL}}\big(\mathbb{Q}_{\boldsymbol{\xi}}(\mathcal{D}) || \mathbb{Q}_{\boldsymbol{\xi}'}(\mathcal{D})\big) = \sum_{k=1}^{\frac{K}{d+2}} D_{\mathrm{KL}}\Big(\mathcal{N}\big(\frac{d+1}{2d}\delta, 1\big) \big|\big| \mathcal{N}\big(\frac{d-1}{2d}\delta, 1\big)\Big) = \frac{K}{d+2}\frac{\delta^2}{d^2}.$$

Then by our choice of $\delta$, we have

$$\min_{\boldsymbol{\xi}, \boldsymbol{\xi}': D_H(\boldsymbol{\xi}, \boldsymbol{\xi}')=1} \inf_{\psi} \Big[ \mathbb{Q}_{\boldsymbol{\xi}}(\psi(\mathcal{D}) \neq \boldsymbol{\xi}) + \mathbb{Q}_{\boldsymbol{\xi}'}(\psi(\mathcal{D}) \neq \boldsymbol{\xi}') \Big] \geq 1 - \Big( \frac{K\delta^2}{2(d+2)d^2} \Big)^{1/2}$$

$$\geq 1 - \Big( \frac{K\delta^2}{2d^3} \Big)^{1/2}$$

$$= \frac{1}{2}.$$

This completes the proof. $\qquad\qquad\square$

## G.8 Proof of Lemma F.3

*Proof.* Recall that

$$\boldsymbol{\Sigma}_h^{\star-1} = \sum_{k=1}^{K} \frac{\boldsymbol{\phi}_h^\tau \boldsymbol{\phi}_h^{\tau\top}}{[\mathbb{V}_h V_h^{\star,\rho}](s_h^\tau, a_h^\tau)} + \lambda I.$$

We first show that with sufficiently large $K$, the clipped conditional variances of the optimal robust value functions are always 1. Note that $V_h^{\star,\rho}(x_2) = 0, \forall h \in [H]$, and

$$V_H^{\star,\rho}(x_1) = \frac{\delta}{2d}\Big( \sum_{i=1}^{d} \frac{1 + \xi_{Hi}}{2} + d \Big) \leq \delta,$$

$$V_{H-1}^{\star,\rho}(x_1) = \frac{\delta}{2d}\Big( \sum_{i=1}^{d} \frac{1 + \xi_{H-1i}}{2} + d \Big) + V_H^{\star,\rho}(x_1) \leq 2\delta,$$

$$\cdots$$

$$V_2^{\star,\rho}(x_1) = \frac{\delta}{2d}\Big( \sum_{i=1}^{d} \frac{1 + \xi_{2i}}{2} + d \Big) + V_3^{\star,\rho}(x_1) \leq (H-1) \cdot \delta.$$

Then, when $K \geq \Omega(H^2 d^3)$, we have

$$\big[ \mathrm{Var}_1 V_2^{\star,\rho} \big](x_1, a) = \big[ \mathbb{P}_1^0 (V_2^{\star,\rho})^2 \big](x_1, a) - \big( \big[ \mathbb{P}_1^0 (V_2^{\star,\rho})^2 \big](x_1, a) \big)^2 \leq (1-\rho)\rho H^2 \delta^2 \leq 1,$$

and by design we have,

$$[\mathrm{Var}_1 V_2^{\star,\rho}](x_2, a) = 0 \text{ and } [\mathrm{Var}_h V_{h+1}^{\star,\rho}](s, a) = 0, \forall (s, a, h) \in \mathcal{S} \times \mathcal{A} \times [H]/\{1\}.$$

Thus, we have $[\mathbb{V}_h V_h^{\star,\rho}](s_h^\tau, a_h^\tau) = 1$, which implies

$$\boldsymbol{\Sigma}_h^\star = \boldsymbol{\Lambda}_h. \qquad\qquad (G.38)$$

Define

$$\tilde{\boldsymbol{\Lambda}}_h = \mathbb{E}^{\pi^b, P^0}[\boldsymbol{\phi}(s_h, a_h)\boldsymbol{\phi}(s_h, a_h)^\top],$$

then by definition we have

$$\tilde{\boldsymbol{\Lambda}}_h = \frac{1}{d+2} \begin{bmatrix} \frac{1}{d^2} & 0 & \cdots & 0 & \frac{1}{d}(1-\frac{1}{d}) & 0 \\ 0 & 0 & \cdots & 0 & 0 & 0 \\ \vdots & \vdots & \vdots & \vdots & \vdots & \vdots \\ 0 & 0 & \cdots & 0 & 0 & 0 \\ \frac{1}{d}(1-\frac{1}{d}) & 0 & \cdots & 0 & (1-\frac{1}{d})^2 & 0 \\ 0 & 0 & \cdots & 0 & 0 & 0 \end{bmatrix} + \frac{1}{d+2} \begin{bmatrix} 0 & 0 & \cdots & 0 & 0 & 0 \\ 0 & \frac{1}{d^2} & \cdots & 0 & \frac{1}{d}(1-\frac{1}{d}) & 0 \\ \vdots & \vdots & \vdots & \vdots & \vdots & \vdots \\ 0 & 0 & \cdots & 0 & 0 & 0 \\ 0 & \frac{1}{d}(1-\frac{1}{d}) & \cdots & 0 & (1-\frac{1}{d})^2 & 0 \\ 0 & 0 & \cdots & 0 & 0 & 0 \end{bmatrix}$$

$$+ \cdots + \frac{1}{d+2} \begin{bmatrix} 0 & 0 & \cdots & 0 & 0 & 0 \\ 0 & 0 & \cdots & 0 & 0 & 0 \\ \vdots & \vdots & & \vdots & \vdots & \vdots \\ 0 & 0 & \cdots & \frac{1}{d^2} & \frac{1}{d}(1-\frac{1}{d}) & 0 \\ 0 & 0 & \cdots & \frac{1}{d}(1-\frac{1}{d}) & (1-\frac{1}{d})^2 & 0 \\ 0 & 0 & \cdots & 0 & 0 & 0 \end{bmatrix} + \frac{1}{d+2} \begin{bmatrix} 0 & 0 & \cdots & 0 & 0 & 0 \\ 0 & 0 & \cdots & 0 & 0 & 0 \\ \vdots & \vdots & & \vdots & \vdots & \vdots \\ 0 & 0 & \cdots & 0 & 0 & 0 \\ 0 & 0 & \cdots & 0 & 1 & 0 \\ 0 & 0 & \cdots & 0 & 0 & 0 \end{bmatrix}$$

$$+ \frac{1}{d+2} \begin{bmatrix} 0 & 0 & \cdots & 0 & 0 & 0 \\ 0 & 0 & \cdots & 0 & 0 & 0 \\ \vdots & \vdots & & \vdots & \vdots & \vdots \\ 0 & 0 & \cdots & 0 & 0 & 0 \\ 0 & 0 & \cdots & 0 & 0 & 0 \\ 0 & 0 & \cdots & 0 & 0 & 1 \end{bmatrix}$$

$$= \frac{d}{d+2} \begin{bmatrix} \frac{1}{d^3} & 0 & \cdots & 0 & \frac{1}{d^2}(1-\frac{1}{d}) & 0 \\ 0 & \frac{1}{d^3} & \cdots & 0 & \frac{1}{d^2}(1-\frac{1}{d}) & 0 \\ \vdots & \vdots & & \vdots & \vdots & \vdots \\ 0 & 0 & \cdots & \frac{1}{d^3} & \frac{1}{d^2}(1-\frac{1}{d}) & 0 \\ \frac{1}{d^2}(1-\frac{1}{d}) & \frac{1}{d^2}(1-\frac{1}{d}) & \cdots & \frac{1}{d^2}(1-\frac{1}{d}) & (1-\frac{1}{d})^2+\frac{1}{d} & 0 \\ 0 & 0 & \cdots & 0 & 0 & \frac{1}{d} \end{bmatrix}.$$

Denote

$$D = \begin{bmatrix} \frac{1}{d^3} & 0 & \cdots & 0 & \frac{1}{d^2}(1-\frac{1}{d}) \\ 0 & \frac{1}{d^3} & \cdots & 0 & \frac{1}{d^2}(1-\frac{1}{d}) \\ \vdots & \vdots & & \vdots & \vdots \\ 0 & 0 & \cdots & \frac{1}{d^3} & \frac{1}{d^2}(1-\frac{1}{d}) \\ \frac{1}{d^2}(1-\frac{1}{d}) & \frac{1}{d^2}(1-\frac{1}{d}) & \cdots & \frac{1}{d^2}(1-\frac{1}{d}) & (1-\frac{1}{d})^2+\frac{1}{d} \end{bmatrix},$$

then by Gaussian elimination, we have

$$D^{-1} = \begin{bmatrix} 2d^3-2d^2+d & d^3-2d^2+d & \cdots & d^3-2d^2+d & d-d^2 \\ d^3-2d^2+d & 2d^3-2d^2+d & \cdots & d^3-2d^2+d & d-d^2 \\ \vdots & \vdots & & \vdots & \vdots \\ d^3-2d^2+d & d^3-2d^2+d & \cdots & 2d^3-2d^2+d & d-d^2 \\ d-d^2 & d-d^2 & \cdots & d-d^2 & d \end{bmatrix}.$$

Note that

$$\tilde{\Lambda}_h = \frac{d}{d+2} \begin{bmatrix} D & 0 \\ 0 & \frac{1}{d} \end{bmatrix},$$

then we have

$$\tilde{\Lambda}_h^{-1} = \frac{d+2}{d} \begin{bmatrix} D^{-1} & 0 \\ 0 & d \end{bmatrix}.$$

Note that $\lambda_{\min}(D) = O(1/d^3)$, thus $\|\tilde{\Lambda}_h^{-1}\| = O(d^3)$. Then when $K > \tilde{O}(d^6)$, for any $(s,a,i,h) \in \mathcal{S} \times \mathcal{A} \times [d] \times [H]$, with probability at least $1-\delta$, we have

$$\|\phi_i(s,a)\mathbf{1}_i\|_{\Lambda_h^{-1}} \le \frac{2}{\sqrt{K}}\|\phi_i(s,a)\mathbf{1}_i\|_{\tilde{\Lambda}_h^{-1}}. \tag{G.39}$$

With this in mind, we have

$$\sup_{P \in \mathcal{U}^\rho(P^0)} \sum_{h=1}^{H} \mathbb{E}^{\pi^\star, P}\Big[ \sum_{i=1}^{d} \|\phi_i(s,a)\mathbf{1}_i\|_{\Sigma_h^{\star-1}} | s_1 = x_1 \Big]$$

$$= \sup_{P \in \mathcal{U}^\rho(P^0)} \sum_{h=1}^{H} \mathbb{E}^{\pi^\star, P}\Big[ \sum_{i=1}^{d} \|\phi_i(s_h,a_h)\mathbf{1}_i\|_{\Lambda_h^{-1}} | s_1 = x_1 \Big]$$

$$\leq \sup_{P \in \mathcal{U}^\rho(P^0)} \sum_{h=1}^{H} \mathbb{E}^{\pi^\star, P}\Big[\frac{2}{\sqrt{K}} \sum_{i=1}^{d} \|\phi_i(s_h, a_h)\mathbf{1}_i\|_{\tilde{\mathbf{\Lambda}}_h^{-1}} |s_1 = x_1\Big] \qquad \text{(G.40)}$$

$$= \sup_{P \in \mathcal{U}^\rho(P^0)} \sum_{h=1}^{H} \mathbb{E}^{\pi^\star, P}\Big[\frac{2}{\sqrt{K}} \sum_{i=1}^{d} \phi_i(s_h, a_h)\big(\tilde{\mathbf{\Lambda}}_h^{-1}\big)_{ii}^{1/2} |s_1 = x_1\Big]$$

$$\leq \frac{4Hd^{3/2}}{\sqrt{K}},$$

where (G.40) is due to (G.39). This concludes the proof. $\qquad\square$

# H  Proof of Supporting Lemmas

## H.1  Proof of Lemma G.1

To prove Lemma G.1, we need the following proposition on the dual formulation under the TV uncertainty set.

**Proposition H.1.** (Strong duality for TV [42, Lemma 4]). Given any probability measure $\mu^0$ over $\mathcal{S}$, a fixed uncertainty level $\rho$, the uncertainty set $\mathcal{U}^\rho(\mu^0) = \{\mu : \mu \in \Delta(\mathcal{S}), D_{TV}(\mu||\mu^0) \leq \rho\}$, and any function $V : \mathcal{S} \to [0, H]$, we obtain

$$\inf_{\mu \in \mathcal{U}^\rho(\mu^0)} \mathbb{E}_{s \sim \mu} V(s) = \max_{\alpha \in [V_{\min}, V_{\max}]} \big\{\mathbb{E}_{s \sim \mu^0}[V(s)]_\alpha - \rho\big(\alpha - \min_{s'}[V(s')]_\alpha\big)\big\}, \qquad \text{(H.1)}$$

where $[V(s)]_\alpha = \min\{V(s), \alpha\}$, $V_{\min} = \min_s V(s)$ and $V_{\max} = \max_s V(s)$. Notably, the range of $\alpha$ can be relaxed to $[0, H]$ without impacting the optimization.

*Proof of Lemma G.1.* By Assumption 3.1 and Proposition H.1, we have

$$\inf_{P_h(\cdot|s,a) \in \mathcal{U}_h^\rho(s,a;\boldsymbol{\mu}_{h,i}^0)} [\mathbb{P}_h \widehat{V}_{h+1}^\rho](s,a) - \widehat{\inf_{P_h(\cdot|s,a) \in \mathcal{U}_h^\rho(s,a;\boldsymbol{\mu}_{h,i}^0)}} [\mathbb{P}_h \widehat{V}_{h+1}^\rho](s,a)$$

$$= \sum_{i=1}^{d} \phi_i(s,a)\Big[\max_{\alpha \in [0,H]}\{\mathbb{E}^{\mu_{h,i}^0}[\widehat{V}_{h+1}^\rho(s)]_\alpha - \rho(\alpha - \min_{s'}[\widehat{V}_{h+1}^\rho(s')]_\alpha)\}$$

$$- \max_{\alpha \in [0,H]}\{\widehat{\mathbb{E}}^{\mu_{h,i}^0}[\widehat{V}_{h+1}^\rho(s)]_\alpha - \rho(\alpha - \min_{s'}[\widehat{V}_{h+1}^\rho(s')]_\alpha)\}\Big].$$

Denote $\alpha_i = \text{argmax}_{\alpha \in [0,H]}\{\mathbb{E}^{\mu_{h,i}^0}[\widehat{V}_{h+1}^\rho(s)]_\alpha - \rho(\alpha - \min_{s'}[\widehat{V}_{h+1}^\rho(s')]_\alpha)\}$, then we have

$$\inf_{P_h(\cdot|s,a) \in \mathcal{U}_h^\rho(s,a;\boldsymbol{\mu}_{h,i}^0)} [\mathbb{P}_h \widehat{V}_{h+1}^\rho](s,a) - \widehat{\inf_{P_h(\cdot|s,a) \in \mathcal{U}_h^\rho(s,a;\boldsymbol{\mu}_{h,i}^0)}} [\mathbb{P}_h \widehat{V}_{h+1}^\rho](s,a)$$

$$\leq \sum_{i=1}^{d} \phi_i(s,a)\big(\mathbb{E}^{\mu_{h,i}^0}[\widehat{V}_{h+1}^\rho(s)]_{\alpha_i} - \widehat{\mathbb{E}}^{\mu_{h,i}^0}[\widehat{V}_{h+1}^\rho(s)]_{\alpha_i}\big)$$

$$= \sum_{i=1}^{d} \phi_i(s,a)\big[\mathbf{1}_i^\top \mathbb{E}^{\boldsymbol{\mu}_h^0}[\widehat{V}_{h+1}^\rho(s)]_{\alpha_i} - \mathbf{1}_i^\top \widehat{\mathbb{E}}^{\boldsymbol{\mu}_h^0}[\widehat{V}_{h+1}^\rho(s)]_{\alpha_i}\big].$$

Here we do reference-advantage decomposition by using the optimal robust value function as the reference function. Specifically, we have

$$\inf_{P_h(\cdot|s,a) \in \mathcal{U}_h^\rho(s,a;\boldsymbol{\mu}_{h,i}^0)} [\mathbb{P}_h \widehat{V}_{h+1}^\rho](s,a) - \widehat{\inf_{P_h(\cdot|s,a) \in \mathcal{U}_h^\rho(s,a;\boldsymbol{\mu}_{h,i}^0)}} [\mathbb{P}_h \widehat{V}_{h+1}^\rho](s,a)$$

$$\leq \sum_{i=1}^{d} \phi_i(s,a)\big[\mathbf{1}_i^\top\big(\mathbb{E}^{\boldsymbol{\mu}_h^0}\big[[\widehat{V}_{h+1}^\rho(s)]_{\alpha_i} - [V_{h+1}^{\star,\rho}(s)]_{\alpha_i} + [V_{h+1}^{\star,\rho}(s)]_{\alpha_i}\big]\big)$$

$$- \mathbf{1}_i^\top\big(\widehat{\mathbb{E}}^{\boldsymbol{\mu}_h^0}\big[[\widehat{V}_{h+1}^\rho(s)]_{\alpha_i} - [V_{h+1}^{\star,\rho}(s)]_{\alpha_i} + [V_{h+1}^{\star,\rho}(s)]_{\alpha_i}\big]\big)\big]$$

$$= \underbrace{\sum_{i=1}^{d} \phi_i(s,a) \mathbf{1}_i^\top \big( \mathbb{E}^{\boldsymbol{\mu}_h^0}[V_{h+1}^{\star,\rho}(s)]_{\alpha_i} - \widehat{\mathbb{E}}^{\boldsymbol{\mu}_h^0}[V_{h+1}^{\star,\rho}(s)]_{\alpha_i} \big)}_{\text{reference uncertainty}}$$

$$+ \underbrace{\sum_{i=1}^{d} \phi_i(s,a) \mathbf{1}_i^\top \big( \mathbb{E}^{\boldsymbol{\mu}_h^0}\big[[\widehat{V}_{h+1}^{\rho}(s)]_{\alpha_i} - [V_{h+1}^{\star,\rho}(s)]_{\alpha_i}\big] - \widehat{\mathbb{E}}^{\boldsymbol{\mu}_h^0}\big[[\widehat{V}_{h+1}^{\rho}(s)]_{\alpha_i} - [V_{h+1}^{\star,\rho}(s)]_{\alpha_i}\big] \big)}_{\text{advantage uncertainty}}.$$

(H.2)

**The Reference Uncertainty.** First, we bound the reference uncertainty. Specifically, we have

$$\sum_{i=1}^{d} \phi_i(s,a) \mathbf{1}_i^\top \big( \mathbb{E}^{\boldsymbol{\mu}_h^0}[V_{h+1}^{\star,\rho}(s)]_{\alpha_i} - \widehat{\mathbb{E}}^{\boldsymbol{\mu}_h^0}[V_{h+1}^{\star,\rho}(s)]_{\alpha_i} \big)$$

$$= \sum_{i=1}^{d} \phi_i(s,a) \mathbf{1}_i^\top \Big( \mathbb{E}^{\boldsymbol{\mu}_h^0}[V_{h+1}^{\star,\rho}(s)]_{\alpha_i} - \boldsymbol{\Lambda}_h^{-1} \sum_{\tau=1}^{K} \boldsymbol{\phi}_h^\tau \big[ \mathbb{P}_h^0[V_{h+1}^{\star,\rho}]_{\alpha_i} \big](s_h^\tau, a_h^\tau)$$

$$+ \boldsymbol{\Lambda}_h^{-1} \sum_{\tau=1}^{K} \boldsymbol{\phi}_h^\tau \big[ \mathbb{P}_h^0[V_{h+1}^{\star,\rho}]_{\alpha_i} \big](s_h^\tau, a_h^\tau) - \boldsymbol{\Lambda}_h^{-1} \sum_{\tau=1}^{K} \boldsymbol{\phi}_h^\tau [V_{h+1}^{\star,\rho}(s_{h+1}^\tau)]_{\alpha_i} \Big)$$

$$= \sum_{i=1}^{d} \phi_i(s,a) \mathbf{1}_i^\top \Big( \mathbb{E}^{\boldsymbol{\mu}_h^0}[V_{h+1}^{\star,\rho}(s)]_{\alpha_i} - \boldsymbol{\Lambda}_h^{-1} \sum_{\tau=1}^{K} \boldsymbol{\phi}_h^\tau \boldsymbol{\phi}_h^{\tau\top} \mathbb{E}^{\boldsymbol{\mu}_h^0}[V_{h+1}^{\star,\rho}(s)]_{\alpha_i}$$

$$+ \boldsymbol{\Lambda}_h^{-1} \sum_{\tau=1}^{K} \boldsymbol{\phi}_h^\tau \big( \big[ \mathbb{P}_h^0[V_{h+1}^{\star,\rho}]_{\alpha_i} \big](s_h^\tau, a_h^\tau) - [V_{h+1}^{\star,\rho}(s_{h+1}^\tau)]_{\alpha_i} \big) \Big).$$

For any function $f : \mathcal{S} \to [0, H-1]$, we define $\eta_h^\tau([f]_{\alpha_i}) = \big( \big[ \mathbb{P}_h^0[f]_{\alpha_i} \big](s_h^\tau, a_h^\tau) - [f(s_{h+1}^\tau)]_{\alpha_i} \big)$. Then, we have

$$\sum_{i=1}^{d} \phi_i(s,a) \mathbf{1}_i^\top \big( \mathbb{E}^{\boldsymbol{\mu}_h^0}[V_{h+1}^{\star,\rho}(s)]_{\alpha_i} - \widehat{\mathbb{E}}^{\boldsymbol{\mu}_h^0}[V_{h+1}^{\star,\rho}(s)]_{\alpha_i} \big)$$

$$= \lambda \sum_{i=1}^{d} \phi_i(s,a) \mathbf{1}_i^\top \boldsymbol{\Lambda}_h^{-1} \mathbb{E}^{\boldsymbol{\mu}_h^0}[V_{h+1}^{\star,\rho}(s)]_{\alpha_i} + \sum_{i=1}^{d} \phi_i(s,a) \mathbf{1}_i^\top \boldsymbol{\Lambda}_h^{-1} \sum_{\tau=1}^{K} \boldsymbol{\phi}_h^\tau \eta_h^\tau([V_{h+1}^{\star,\rho}]_{\alpha_i})$$

$$\leq \underbrace{\lambda \sum_{i=1}^{d} \|\phi_i(s,a)\mathbf{1}_i\|_{\boldsymbol{\Lambda}_h^{-1}} \|\mathbb{E}^{\boldsymbol{\mu}_h^0}[V_{h+1}^{\star,\rho}(s)]_{\alpha_i}\|_{\boldsymbol{\Lambda}_h^{-1}}}_{\text{i}} + \underbrace{\sum_{i=1}^{d} \|\phi_i(s,a)\mathbf{1}_i\|_{\boldsymbol{\Lambda}_h^{-1}} \Big\| \sum_{\tau=1}^{K} \boldsymbol{\phi}_h^\tau \eta_h^\tau([V_{h+1}^{\star,\rho}]_{\alpha_i}) \Big\|_{\boldsymbol{\Lambda}_h^{-1}}}_{\text{ii}}.$$

(H.3)

**The Advantage Uncertainty.** Next, we bound the advantage uncertainty. By similar argument in bounding the reference uncertainty, we have

$$\sum_{i=1}^{d} \phi_i(s,a) \mathbf{1}_i^\top \big( \mathbb{E}^{\boldsymbol{\mu}_h^0}\big[[\widehat{V}_{h+1}^{\rho}(s)]_{\alpha_i} - [V_{h+1}^{\star,\rho}(s)]_{\alpha_i}\big] - \widehat{\mathbb{E}}^{\boldsymbol{\mu}_h^0}\big[[\widehat{V}_{h+1}^{\rho}(s)]_{\alpha_i} - [V_{h+1}^{\star,\rho}(s)]_{\alpha_i}\big] \big)$$

$$\leq \underbrace{\lambda \sum_{i=1}^{d} \|\phi_i(s,a)\mathbf{1}_i\|_{\boldsymbol{\Lambda}_h^{-1}} \Big\| \mathbb{E}^{\boldsymbol{\mu}_h^0}\big[[\widehat{V}_{h+1}^{\rho}(s)]_{\alpha_i} - [V_{h+1}^{\star,\rho}(s)]_{\alpha_i}\big] \Big\|_{\boldsymbol{\Lambda}_h^{-1}}}_{\text{iii}}$$

$$+ \underbrace{\sum_{i=1}^{d} \|\phi_i(s,a)\mathbf{1}_i\|_{\boldsymbol{\Lambda}_h^{-1}} \Big\| \sum_{\tau=1}^{K} \boldsymbol{\phi}_h^\tau \eta_h^\tau([\widehat{V}_{h+1}^{\rho}(s)]_{\alpha_i} - [V_{h+1}^{\star,\rho}(s)]_{\alpha_i}) \Big\|_{\boldsymbol{\Lambda}_h^{-1}}}_{\text{iv}}.$$

(H.4)

Combining (H.2), (H.3) and (H.4), we have

$$\inf_{P_h(\cdot|s,a)\in\mathcal{U}_h^\rho(s,a;\boldsymbol{\mu}_{h,i}^0)}[\mathbb{P}_h\widehat{V}_{h+1}^\rho](s,a) - \widehat{\inf_{P_h(\cdot|s,a)\in\mathcal{U}_h^\rho(s,a;\boldsymbol{\mu}_{h,i}^0)}}[\mathbb{P}_h\widehat{V}_{h+1}^\rho](s,a)$$

$$\leq \lambda\underbrace{\sum_{i=1}^d\|\phi_i(s,a)\mathbf{1}_i\|_{\boldsymbol{\Lambda}_h^{-1}}\|\mathbb{E}^{\boldsymbol{\mu}_h^0}[V_{h+1}^{\star,\rho}(s)]_{\alpha_i}\|_{\boldsymbol{\Lambda}_h^{-1}} + \sum_{i=1}^d\|\phi_i(s,a)\mathbf{1}_i\|_{\boldsymbol{\Lambda}_h^{-1}}\Big\|\sum_{\tau=1}^K\phi_h^\tau\eta_h^\tau([V_{h+1}^{\star,\rho}]_{\alpha_i})\Big\|_{\boldsymbol{\Lambda}_h^{-1}}}_{\text{i}} \underbrace{\phantom{x}}_{\text{ii}}$$

$$+ \lambda\underbrace{\sum_{i=1}^d\|\phi_i(s,a)\mathbf{1}_i\|_{\boldsymbol{\Lambda}_h^{-1}}\Big\|\mathbb{E}^{\boldsymbol{\mu}_h^0}\big[[\widehat{V}_{h+1}^\rho(s)]_{\alpha_i} - [V_{h+1}^{\star,\rho}(s)]_{\alpha_i}\big]\Big\|_{\boldsymbol{\Lambda}_h^{-1}}}_{\text{iii}}$$

$$+ \underbrace{\sum_{i=1}^d\|\phi_i(s,a)\mathbf{1}_i\|_{\boldsymbol{\Lambda}_h^{-1}}\Big\|\sum_{\tau=1}^K\phi_h^\tau\eta_h^\tau([\widehat{V}_{h+1}^\rho(s)]_{\alpha_i} - [V_{h+1}^{\star,\rho}(s)]_{\alpha_i})\Big\|_{\boldsymbol{\Lambda}_h^{-1}}}_{\text{iv}}.$$

On the other hand, we can similarly deduce

$$\widehat{\inf_{P_h(\cdot|s,a)\in\mathcal{U}_h^\rho(s,a;\boldsymbol{\mu}_{h,i}^0)}}[\mathbb{P}_h\widehat{V}_{h+1}^\rho](s,a) - \inf_{P_h(\cdot|s,a)\in\mathcal{U}_h^\rho(s,a;\boldsymbol{\mu}_{h,i}^0)}[\mathbb{P}_h\widehat{V}_{h+1}^\rho](s,a)$$

$$\leq \lambda\underbrace{\sum_{i=1}^d\|\phi_i(s,a)\mathbf{1}_i\|_{\boldsymbol{\Lambda}_h^{-1}}\|\mathbb{E}^{\boldsymbol{\mu}_h^0}[V_{h+1}^{\star,\rho}(s)]_{\alpha_i'}\|_{\boldsymbol{\Lambda}_h^{-1}} + \sum_{i=1}^d\|\phi_i(s,a)\mathbf{1}_i\|_{\boldsymbol{\Lambda}_h^{-1}}\Big\|\sum_{\tau=1}^K\phi_h^\tau\eta_h^\tau([V_{h+1}^{\star,\rho}]_{\alpha_i'})\Big\|_{\boldsymbol{\Lambda}_h^{-1}}}_{\text{i}} \underbrace{\phantom{x}}_{\text{ii}}$$

$$+ \lambda\underbrace{\sum_{i=1}^d\|\phi_i(s,a)\mathbf{1}_i\|_{\boldsymbol{\Lambda}_h^{-1}}\Big\|\mathbb{E}^{\boldsymbol{\mu}_h^0}\big[[\widehat{V}_{h+1}^\rho(s)]_{\alpha_i'} - [V_{h+1}^{\star,\rho}(s)]_{\alpha_i'}\big]\Big\|_{\boldsymbol{\Lambda}_h^{-1}}}_{\text{iii}}$$

$$+ \underbrace{\sum_{i=1}^d\|\phi_i(s,a)\mathbf{1}_i\|_{\boldsymbol{\Lambda}_h^{-1}}\Big\|\sum_{\tau=1}^K\phi_h^\tau\eta_h^\tau([\widehat{V}_{h+1}^\rho(s)]_{\alpha_i'} - [V_{h+1}^{\star,\rho}(s)]_{\alpha_i'})\Big\|_{\boldsymbol{\Lambda}_h^{-1}}}_{\text{iv}},$$

where $\alpha_i' = \mathrm{argmax}_{\alpha\in[0,H]}\{\widehat{\mathbb{E}}^{\boldsymbol{\mu}_{h,i}^0}[\widehat{V}_{h+1}^\rho(s)]_\alpha - \rho(\alpha - \min_{s'}[\widehat{V}_{h+1}^\rho(s')]_\alpha)\}$. Then for all $i \in [d]$, there exist $\tilde{\alpha}_i \in \{\alpha_i, \alpha_i'\}$, such that

$$\Big|\inf_{P_h(\cdot|s,a)\in\mathcal{U}_h^\rho(s,a;\boldsymbol{\mu}_{h,i}^0)}[\mathbb{P}_h\widehat{V}_{h+1}^\rho](s,a) - \widehat{\inf_{P_h(\cdot|s,a)\in\mathcal{U}_h^\rho(s,a;\boldsymbol{\mu}_{h,i}^0)}}[\mathbb{P}_h\widehat{V}_{h+1}^\rho](s,a)\Big|$$

$$\leq \lambda\underbrace{\sum_{i=1}^d\|\phi_i(s,a)\mathbf{1}_i\|_{\boldsymbol{\Lambda}_h^{-1}}\|\mathbb{E}^{\boldsymbol{\mu}_h^0}[V_{h+1}^{\star,\rho}(s)]_{\tilde{\alpha}_i}\|_{\boldsymbol{\Lambda}_h^{-1}} + \sum_{i=1}^d\|\phi_i(s,a)\mathbf{1}_i\|_{\boldsymbol{\Lambda}_h^{-1}}\Big\|\sum_{\tau=1}^K\phi_h^\tau\eta_h^\tau([V_{h+1}^{\star,\rho}]_{\tilde{\alpha}_i})\Big\|_{\boldsymbol{\Lambda}_h^{-1}}}_{\text{i}} \underbrace{\phantom{x}}_{\text{ii}}$$

$$+ \lambda\underbrace{\sum_{i=1}^d\|\phi_i(s,a)\mathbf{1}_i\|_{\boldsymbol{\Lambda}_h^{-1}}\Big\|\mathbb{E}^{\boldsymbol{\mu}_h^0}\big[[\widehat{V}_{h+1}^\rho(s)]_{\tilde{\alpha}_i} - [V_{h+1}^{\star,\rho}(s)]_{\tilde{\alpha}_i}\big]\Big\|_{\boldsymbol{\Lambda}_h^{-1}}}_{\text{iii}}$$

$$+ \underbrace{\sum_{i=1}^d\|\phi_i(s,a)\mathbf{1}_i\|_{\boldsymbol{\Lambda}_h^{-1}}\Big\|\sum_{\tau=1}^K\phi_h^\tau\eta_h^\tau([\widehat{V}_{h+1}^\rho(s)]_{\tilde{\alpha}_i} - [V_{h+1}^{\star,\rho}(s)]_{\tilde{\alpha}_i})\Big\|_{\boldsymbol{\Lambda}_h^{-1}}}_{\text{iv}},$$

This concludes the proof. □

## H.2 Proof of Lemma G.2

The proof of Lemma G.2 will use the following fact.

**Lemma H.2.** [14, Lemma D.1] Let $\boldsymbol{\Lambda}_t = \lambda \mathbf{I} + \sum_{i=1}^{t} \boldsymbol{\phi}_i \boldsymbol{\phi}_i^\top$, where $\boldsymbol{\phi}_i \in \mathbb{R}^d$ and $\lambda > 0$. Then:

$$\sum_{i=1}^{t} \boldsymbol{\phi}_i^\top (\boldsymbol{\Lambda}_t)^{-1} \boldsymbol{\phi}_i \leq d.$$

*Proof of Lemma G.2.* The proof of Lemma G.2 is similar to that of Lemma E.1 in [20]. Denote $\alpha_i = \mathrm{argmax}_{\alpha \in [0,H]}\{\hat{z}_{h,i}(\alpha) - \rho(\alpha - \min_{s'}[\widehat{V}_{h+1}^\rho(s')]_\alpha)\}, i \in [d]$. For any vector $\boldsymbol{v} \in \mathbb{R}^d$, we have

$$\left| \boldsymbol{v}^\top \boldsymbol{w}_h^\rho \right| = \left| \boldsymbol{v}^\top \boldsymbol{\theta}_h + \boldsymbol{v}^\top \left[ \max_{\alpha \in [0,H]} \{\hat{z}_{h,i}(\alpha) - \rho(\alpha - \min_{s'}[\widehat{V}_{h+1}^\rho(s')]_\alpha)\} \right]_{i \in [d]} \right|$$

$$\leq \left| \boldsymbol{v}^\top \boldsymbol{\theta}_h \right| + \left| \boldsymbol{v}^\top \left[ \max_{\alpha \in [0,H]} \{\hat{z}_{h,i}(\alpha) - \rho(\alpha - \min_{s'}[\widehat{V}_{h+1}^\rho(s')]_\alpha)\} \right]_{i \in [d]} \right|$$

$$\leq \sqrt{d} \|\boldsymbol{v}\|_2 + H\|\boldsymbol{v}\|_1 + \left| \boldsymbol{v}^\top \left[ \mathbf{1}_i^\top \left( \boldsymbol{\Lambda}_h^{-1} \sum_{\tau=1}^{K} \boldsymbol{\phi}_h^\tau [\max_a \widehat{Q}_{h+1}^\rho(s_{h+1}^\tau, a)]_{\alpha_i} \right) \right]_{i \in [d]} \right| \quad \text{(H.5)}$$

$$\leq \sqrt{d} \|\boldsymbol{v}\|_2 + H\sqrt{d}\|\boldsymbol{v}\|_2 + \sqrt{\left[ \sum_{\tau=1}^{K} \boldsymbol{v}^\top \boldsymbol{\Lambda}_h^{-1} \boldsymbol{v} \right] \left[ \sum_{\tau=1}^{K} (\boldsymbol{\phi}_h^\tau)^\top \boldsymbol{\Lambda}_h^{-1} \boldsymbol{\phi}_h^\tau \right] \cdot H} \quad \text{(H.6)}$$

$$\leq 2H\|\boldsymbol{v}\|_2 \sqrt{dK/\lambda}. \quad \text{(H.7)}$$

We note that the term $[(\boldsymbol{\Lambda}_h^{-1} \sum_{\tau=1}^{K} \boldsymbol{\phi}_h^\tau [\max_a \widehat{Q}_{h+1}^\rho(s_{h+1}^\tau, a)]_{\alpha_i})_i]_{i \in [d]}$ in (H.5) is constructed by first taking out the $i$-th coordinate of the ridge solution vector, $\boldsymbol{\Lambda}_h^{-1} \sum_{\tau=1}^{K} \boldsymbol{\phi}_h^\tau [\max_a \widehat{Q}_{h+1}^\rho(s_{h+1}^\tau, a)]_{\alpha_i} \in \mathbb{R}^d$, $\forall i \in [d]$, and then concatenating all $d$ values into a vector. Inequality (H.5) is due to the fact that $\rho \leq 1$, (H.6) is due to the fact that $\widehat{Q}_{h+1}^\rho \leq H$, and (H.7) is due to Lemma H.2 with $t = K$ and the fact that the minimum eigenvalue of $\boldsymbol{\Lambda}_h$ is lower bounded by $\lambda$. The remainder of the proof follows from the fact that $\|\boldsymbol{w}_h^\rho\|_2 = \max_{\boldsymbol{v}:\|\boldsymbol{v}\|_2=1} |\boldsymbol{v}^\top \boldsymbol{w}_h^\rho|$.

$\square$

## H.3  Proof of Lemma G.4

The proof of Lemma G.4 will use the following fact.

**Lemma H.3.** [14, Covering Number of Euclidean Ball] For any $\epsilon > 0$, the $\epsilon$-covering number of the Euclidean ball in $\mathbb{R}^d$ with radius $R > 0$ is upper bounded by $(1 + 2R/\epsilon)^d$.

*Proof of Lemma G.4.* The proof is similar to the proof of Lemma E.3 in [20]. Denote $\boldsymbol{A} = \beta^2 \boldsymbol{\Sigma}_h^{-1}$, so we have

$$\mathcal{V}_h(\cdot) = \max_{a \in \mathcal{A}} \left\{ \boldsymbol{\phi}(s,a)^\top \boldsymbol{\theta} - \sum_{i=1}^{d} \sqrt{\phi_i(s,a) \mathbf{1}_i^\top \boldsymbol{A} \phi_i(s,a) \mathbf{1}_i} \right\}_{[0,H-h+1]}, \quad \text{(H.8)}$$

for $\|\boldsymbol{\theta}\| \leq L$, $\|\boldsymbol{A}\| \leq B^2 \lambda^{-1}$. For any two functions $V_1, V_2 \in \mathcal{V}$, let them take the form in (H.8) with parameters $(\boldsymbol{\theta}_1, \boldsymbol{A}_1)$ and $(\boldsymbol{\theta}_2, \boldsymbol{A}_2)$, respectively. Then since both $\{\cdot\}_{[0,H-h+1]}$ and $\max_a$ are contraction maps, we have

$$\mathrm{dist}(V_1, V_2) \leq \sup_{x,a} \left| \left[ \boldsymbol{\theta}_1^\top \boldsymbol{\phi}(x,a) - \sum_{i=1}^{d} \sqrt{\phi_i(x,a) \mathbf{1}_i^\top \boldsymbol{A}_1 \phi_i(x,a) \mathbf{1}_i} \right] \right.$$

$$\left. - \left[ \boldsymbol{\theta}_2^\top \boldsymbol{\phi}(x,a) - \sum_{i=1}^{d} \sqrt{\phi_i(x,a) \mathbf{1}_i^\top \boldsymbol{A}_2 \phi_i(x,a) \mathbf{1}_i} \right] \right|$$

$$\leq \sup_{\boldsymbol{\phi}:\|\boldsymbol{\phi}\| \leq 1} \left| \left[ \boldsymbol{\theta}_1^\top \boldsymbol{\phi} - \sum_{i=1}^{d} \sqrt{\phi_i \mathbf{1}_i^\top \boldsymbol{A}_1 \phi_i \mathbf{1}_i} \right] - \left[ \boldsymbol{\theta}_2^\top \boldsymbol{\phi} - \sum_{i=1}^{d} \sqrt{\phi_i \mathbf{1}_i^\top \boldsymbol{A}_2 \phi_i \mathbf{1}_i} \right] \right|$$

$$\leq \sup_{\boldsymbol{\phi}:\|\boldsymbol{\phi}\| \leq 1} \left| (\boldsymbol{\theta}_1 - \boldsymbol{\theta}_2)^\top \boldsymbol{\phi} \right| + \sup_{\boldsymbol{\phi}:\|\boldsymbol{\phi}\| \leq 1} \sum_{i=1}^{d} \sqrt{\phi_i \mathbf{1}_i^\top (\boldsymbol{A}_1 - \boldsymbol{A}_2) \phi_i \mathbf{1}_i} \quad \text{(H.9)}$$

$$\leq \|\boldsymbol{\theta}_1 - \boldsymbol{\theta}_2\| + \sqrt{\|\boldsymbol{A}_1 - \boldsymbol{A}_2\|} \sup_{\boldsymbol{\phi}:\|\boldsymbol{\phi}\|\leq 1} \sum_{i=1}^d \|\phi_i \mathbf{1}_i\|$$

$$\leq \|\boldsymbol{\theta}_1 - \boldsymbol{\theta}_2\| + \sqrt{\|\boldsymbol{A}_1 - \boldsymbol{A}_2\|_F}, \tag{H.10}$$

where (H.9) follows from triangular inequality and the fact that $|\sqrt{x} - \sqrt{y}| \leq \sqrt{|x-y|}$, $\forall x, y \geq 0$. For matrices, $\|\cdot\|$ and $\|\cdot\|_F$ denote the matrix operator norm and Frobenius norm respectively.

Let $\mathcal{C}_{\boldsymbol{\theta}}$ be an $\epsilon/2$-cover of $\{\boldsymbol{\theta} \in \mathbb{R}^d | \|\boldsymbol{\theta}\|_2 \leq L\}$ with respect to the 2-norm, and $\mathcal{C}_A$ be an $\epsilon^2/4$-cover of $\{A \in \mathbb{R}^{d\times d} | \|A\|_F \leq d^{1/2}B^2\lambda^{-1}\}$ with respect to the Frobenius norm. By Lemma H.3, we know:

$$\left|\mathcal{C}_{\boldsymbol{\theta}}\right| \leq \left(1 + 4L/\epsilon\right)^d, \quad \left|\mathcal{C}_A\right| \leq \left[1 + 8d^{1/2}B^2/(\lambda\epsilon^2)\right]^{d^2}.$$

By (H.10), for any $V_1 \in \mathcal{V}$, there exists $\boldsymbol{\theta}_2 \in \mathcal{C}_{\boldsymbol{\theta}}$ and $A_2 \in \mathcal{C}_A$ such that $V_2$ parametrized by $(\boldsymbol{\theta}_2, A_2)$ satisfies $\mathrm{dist}(V_1, V_2) \leq \epsilon$. Hence, it holds that $\mathcal{N}_\epsilon \leq |\mathcal{C}_{\boldsymbol{\theta}}| \cdot |\mathcal{C}_A|$, which leads to

$$\log \mathcal{N}_\epsilon \leq \log |\mathcal{C}_{\mathbf{w}}| + \log |\mathcal{C}_A| \leq d \log(1 + 4L/\epsilon) + d^2 \log \left[1 + 8d^{1/2}B^2/(\lambda\epsilon^2)\right].$$

This concludes the proof. $\qquad\square$

### H.4 Proof of Theorem B.1

In this section, we give the proof of Theorem B.1, which largely follows the proof of Theorem 5.2, only with minor modifications of the argument of the variance estimation.

The following lemma bounds the estimation error by reference-advantage decomposition.

**Lemma H.4** (Modified Variance-Aware Reference-Advantage Decomposition). *There exist $\{\alpha_i\}_{i\in[d]}$, where $\alpha_i \in [0, H], \forall i \in [d]$, such that*

$$\left| \inf_{P_h(\cdot|s,a)\in\mathcal{U}_h^\rho(s,a;\boldsymbol{\mu}_{h,i}^0)} [\mathbb{P}_h \widehat{V}_{h+1}^\rho](s,a) - \widehat{\inf_{P_h(\cdot|s,a)\in\mathcal{U}_h^\rho(s,a;\boldsymbol{\mu}_{h,i}^0)} [\mathbb{P}_h \widehat{V}_{h+1}^\rho]}(s,a) \right|$$

$$\leq \underbrace{\lambda \sum_{i=1}^d \|\phi_i(s,a)\mathbf{1}_i\|_{\boldsymbol{\Sigma}_h^{-1}} \|\mathbb{E}^{\boldsymbol{\mu}_h^0}[V_{h+1}^{\star,\rho}(s)]_{\alpha_i}\|_{\boldsymbol{\Sigma}_h^{-1}}}_{\text{i}} + \underbrace{\sum_{i=1}^d \|\phi_i(s,a)\mathbf{1}_i\|_{\boldsymbol{\Sigma}_h^{-1}} \left\| \sum_{\tau=1}^K \frac{\phi_h^\tau \eta_h^\tau([V_{h+1}^{\star,\rho}]_{\alpha_i})}{\widehat{\sigma}_h^2(s_h^\tau, a_h^\tau)} \right\|_{\boldsymbol{\Sigma}_h^{-1}}}_{\text{ii}}$$

$$+ \underbrace{\lambda \sum_{i=1}^d \|\phi_i(s,a)\mathbf{1}_i\|_{\boldsymbol{\Sigma}_h^{-1}} \left\| \mathbb{E}^{\boldsymbol{\mu}_h^0} \left[ [\widehat{V}_{h+1}^\rho(s)]_{\alpha_i} - [V_{h+1}^{\star,\rho}(s)]_{\alpha_i} \right] \right\|_{\boldsymbol{\Sigma}_h^{-1}}}_{\text{iii}}$$

$$+ \underbrace{\sum_{i=1}^d \|\phi_i(s,a)\mathbf{1}_i\|_{\boldsymbol{\Sigma}_h^{-1}} \left\| \sum_{\tau=1}^K \frac{\phi_h^\tau \eta_h^\tau([\widehat{V}_{h+1}^\rho(s)]_{\alpha_i} - [V_{h+1}^{\star,\rho}(s)]_{\alpha_i})}{\widehat{\sigma}_h^2(s_h^\tau, a_h^\tau)} \right\|_{\boldsymbol{\Sigma}_h^{-1}}}_{\text{iv}},$$

*where $\eta_h^\tau([f]_{\alpha_i}) = \left([\mathbb{P}_h^0[f]_{\alpha_i}](s_h^\tau, a_h^\tau) - [f(s_{h+1}^\tau)]_{\alpha_i}\right)$, for any function $f: \mathcal{S} \to [0, H-1]$.*

*Proof of Theorem B.1.* To prove this theorem, we bound the estimation error by $\Gamma_h(s,a)$, then invoke Lemma D.1 to get the results. First, we bound terms i-iv in Lemma H.4 at each step $h \in [H]$ respectively to deduce $\Gamma_h(s,a)$.

**Bound i and iii:** We set $\lambda = 1/H^2$ to ensure that for all $(s,a,h) \in \mathcal{S} \times \mathcal{A} \times [H]$, we have

$$\text{i} + \text{iii} \leq \sqrt{\lambda}\sqrt{d}H \sum_{i=1}^d \|\phi_i(s,a)\mathbf{1}_i\|_{\boldsymbol{\Sigma}_h^{-1}} = \sqrt{d} \sum_{i=1}^d \|\phi_i(s,a)\mathbf{1}_i\|_{\boldsymbol{\Sigma}_h^{-1}}. \tag{H.11}$$

**Bound ii:** For all $(s,a,\alpha) \in \mathcal{S} \times \mathcal{A} \times [0, H]$, by definition we have $\widehat{\sigma}_h(s,a) \geq 1$. Thus, for all $(h,\tau,i) \in [H] \times [K] \times [d]$, we have $\eta_h^\tau([V_{h+1}^{\star,\rho}]_{\alpha_i})/\widehat{\sigma}_h(s_h^\tau, a_h^\tau) \leq H$. Note that $V_{H+1}^{\star,\rho}$ is independent of $\mathcal{D}$, we can directly apply Bernstein-type self-normalized concentration inequality

Lemma I.2 and a union bound to obtain the upper bound. In concrete, we define the filtration $\mathcal{F}_{\tau-1,h} = \sigma(\{(s_h^j, a_h^j)\}_{j=1}^{\tau} \cup \{s_{h+1}^j\}_{j=1}^{\tau-1})$. Since $V_{h+1}^{\star,\rho}$ and $\widehat{\sigma}_h(s,a)$ are independent of $\mathcal{D}$, thus $\eta_h^\tau([V_{h+1}^{\star,\rho}]_{\alpha_i})/\widehat{\sigma}_h(s_h^\tau, a_h^\tau)$ is mean-zero conditioned on the filtration $\mathcal{F}_{\tau-1,h}$. By Lemma E.1 with $\alpha = H$, we have

$$\big[\mathbb{V}_h V_{h+1}^{\star,\rho}\big](s,a) - \tilde{O}\Big(\frac{dH^3}{\sqrt{K\kappa}}\Big) \leq \widehat{\sigma}_h^2(s,a) \leq \big[\mathbb{V}_h V_{h+1}^{\star,\rho}\big](s,a), \tag{H.12}$$

thus, for any $\alpha_i \in [0, H]$, we have

$$\big[\mathbb{V}_h[V_{h+1}^{\star,\rho}]_{\alpha_i}\big](s,a) - \tilde{O}\Big(\frac{dH^3}{\sqrt{K\kappa}}\Big) \leq \big[\mathbb{V}_h V_{h+1}^{\star,\rho}\big](s,a) - \tilde{O}\Big(\frac{dH^3}{\sqrt{K\kappa}}\Big) \leq \widehat{\sigma}_h^2(s,a). \tag{H.13}$$

Further, we have

$$\mathbb{E}\Big[\Big(\frac{\eta_h^\tau([V_{h+1}^{\star,\rho}]_{\alpha_i})}{\widehat{\sigma}_h(s_h^\tau, a_h^\tau)}\Big)^2\Big|\mathcal{F}_{\tau-1,h}\Big] = \frac{[\mathrm{Var}[V_{h+1}^{\star,\rho}]_{\alpha_i}](s_h^\tau, a_h^\tau)}{\widehat{\sigma}_h^2(s_h^\tau, a_h^\tau)} \tag{H.14}$$

$$\leq \frac{[\mathbb{V}[V_{h+1}^{\star,\rho}]_{\alpha_i}](s_h^\tau, a_h^\tau)}{\widehat{\sigma}_h^2(s_h^\tau, a_h^\tau)}$$

$$= \frac{[\mathbb{V}[V_{h+1}^{\star,\rho}]_{\alpha_i}](s_h^\tau, a_h^\tau) - \tilde{O}(dH^3/\sqrt{K\kappa})}{\widehat{\sigma}_h^2(s_h^\tau, a_h^\tau)} + \frac{\tilde{O}(dH^3/\sqrt{K\kappa})}{\widehat{\sigma}_h^2(s_h^\tau, a_h^\tau)}$$

$$\leq 1 + \frac{\tilde{O}(dH^3/\sqrt{K\kappa})}{\widehat{\sigma}_h^2(s_h^\tau, a_h^\tau) - \tilde{O}(dH^3/\sqrt{K\kappa})} \tag{H.15}$$

$$\leq 1 + 2\tilde{O}\Big(\frac{dH^3}{\sqrt{K\kappa}}\Big), \tag{H.16}$$

where (H.14) holds by the fact that $\widehat{\sigma}_h^2(\cdot, \cdot)$ is independent of $\mathcal{D}$ and $(s_h^\tau, a_h^\tau)$ is $\mathcal{F}_{\tau-1,h}$ measurable. (H.15) holds by (H.13), and (H.16) holds by setting $K \geq \tilde{\Omega}(d^2 H^6/\kappa)$ such that $\widehat{\sigma}_h^2(s_h^\tau, a_h^\tau) - \tilde{O}(dH^3/\sqrt{K\kappa}) \geq 1 - \tilde{O}(dH^3/\sqrt{K\kappa}) \geq 1/2$. Further, by (H.16), our choice of $K$ also ensures that $\mathbb{E}\big[(\eta_h^\tau([V_{h+1}^{\star,\rho}]_{\alpha_i}))^2|\mathcal{F}_{\tau-1,h}\big] = O(1)$. Then by Lemma I.2, we have

$$\Big\|\sum_{\tau=1}^K \frac{\phi_h^\tau \eta_h^\tau([V_{h+1}^{\star,\rho}]_{\alpha_i})}{\widehat{\sigma}_h^2(s_h^\tau, a_h^\tau)}\Big\|_{\Sigma_h^{-1}} \leq \tilde{O}(\sqrt{d}).$$

This implies

$$\text{ii} \leq \tilde{O}(\sqrt{d}) \sum_{i=1}^d \|\phi_i(s,a)\mathbf{1}_i\|_{\Sigma_h^{-1}}. \tag{H.17}$$

**Bound iv:** Following the same induction analysis procedure, we know that $\|[\widehat{V}_{h+1}^\rho]_{\alpha_i} - [V_{h+1}^{\star,\rho}]_{\alpha_i}\| \leq \tilde{O}(\sqrt{d}H^2/\sqrt{K\kappa})$. Using standard $\epsilon$-covering number argument and Lemma I.1, we have

$$\text{iv} \leq \tilde{O}\Big(\frac{d^{3/2}H^2}{\sqrt{K\kappa}}\Big) \sum_{i=1}^d \|\phi_i(s,a)\mathbf{1}_i\|_{\Sigma_h^{-1}}. \tag{H.18}$$

To make it non-dominant, we require $K \geq \tilde{\Omega}(d^2 H^4/\kappa)$. By (H.12), we have $\widehat{\sigma}_h^2(s_h^\tau, a_h^\tau) \leq [\mathbb{V}_h V_{h+1}^\star](s_h^\tau, a_h^\tau)$, which implies that

$$\Big(\sum_{\tau=1}^K \frac{\phi_h^\tau \phi_h^{\tau\top}}{\widehat{\sigma}_h^2(s_h^\tau, a_h^\tau)} + \lambda I\Big)^{-1} \preceq \Big(\sum_{\tau=1}^K \frac{\phi_h^\tau \phi_h^{\tau\top}}{[\mathbb{V}_h V_{h+1}^\star](s_h^\tau, a_h^\tau)} + \lambda I\Big)^{-1}.$$

Combining (H.11), (H.17) and (H.18), we have

$$\Big|\inf_{P_h(\cdot|s,a)\in\mathcal{U}_h^\rho(s,a;\boldsymbol{\mu}_{h,i}^0)}[\mathbb{P}_h\widehat{V}_{h+1}^\rho](s,a) - \widehat{\inf_{P_h(\cdot|s,a)\in\mathcal{U}_h^\rho(s,a;\boldsymbol{\mu}_{h,i}^0)}}[\mathbb{P}_h\widehat{V}_{h+1}^\rho](s,a)\Big|$$

$$\leq \tilde{O}(\sqrt{d}) \sum_{i=1}^d \|\phi_i(s,a)\mathbf{1}_i\|_{\Sigma_h^{\star-1}}.$$

Define $\Gamma_h(s,a) = \tilde{O}(\sqrt{d}) \sum_{i=1}^d \|\phi_i(s,a)\mathbf{1}_i\|_{\Sigma_h^{\star-1}}$, we concludes the proof by invoking Lemma D.1.

$\square$

# I   Auxiliary Lemmas

**Lemma I.1** (Concentration of Self-Normalized Processes). [1, Theorem 1] Let $\{\epsilon_t\}_{t=1}^{\infty}$ be a real-valued stochastic process with corresponding filtration $\{\mathcal{F}_t\}_{t=0}^{\infty}$. Let $\epsilon_t | \mathcal{F}_{t-1}$ be mean-zero and $\sigma$-subGaussian; i.e. $\mathbb{E}[\epsilon_t | \mathcal{F}_{t-1}] = 0$, and

$$\forall \lambda \in \mathbb{R}, \quad \mathbb{E}[e^{\lambda \epsilon_t} | \mathcal{F}_{t-1}] \leq e^{\lambda^2 \sigma^2 / 2}.$$

Let $\{\phi_t\}_{t=1}^{\infty}$ be an $\mathbb{R}^d$-valued stochastic process where $\phi_t$ is $\mathcal{F}_{t-1}$ measurable. Assume $\boldsymbol{\Lambda}_0$ is a $d \times d$ positive definite matrix, and let $\boldsymbol{\Lambda}_t = \boldsymbol{\Lambda}_0 + \sum_{s=1}^{t} \phi_s \phi_s^{\top}$. Then for any $\delta > 0$, with probability at least $1 - \delta$, we have for all $t \geq 0$:

$$\left\| \sum_{s=1}^{t} \phi_s \epsilon_s \right\|_{\boldsymbol{\Lambda}_t^{-1}}^{2} \leq 2\sigma^2 \log \left[ \frac{\det(\boldsymbol{\Lambda}_t)^{1/2} \det(\boldsymbol{\Lambda}_0)^{-1/2}}{\delta} \right].$$

**Lemma I.2** (Bernstein inequality for self-normalized martingales). [65, Theorem 2] Let $\{\eta_t\}_{t=1}^{\infty}$ be a real-valued stochastic process. Let $\{\mathcal{F}_t\}_{t=0}^{\infty}$ be a filtration, such that $\eta_t$ is $\mathcal{F}_t$-measurable. Assume $\eta_t$ also satisfies

$$|\eta_t| \leq R, \mathbb{E}[\eta_t | \mathcal{F}_{t-1}] = 0, \mathbb{E}[\eta_t^2 | \mathcal{F}_{t-1}] \leq \sigma^2.$$

Let $\{\boldsymbol{x}_t\}_{t=1}^{\infty}$ be an $\mathbb{R}^d$-valued stochastic process where $\boldsymbol{x}_t$ is $\mathcal{F}_{t-1}$ measurable and $\|\boldsymbol{x}_t\| \leq L$. Let $\boldsymbol{\Lambda}_t = \lambda \mathbf{I}_d + \sum_{s=1}^{t} \boldsymbol{x}_s \boldsymbol{x}_s^{\top}$. Then for any $\delta > 0$, with probability at least $1 - \delta$, for all $t > 0$,

$$\left\| \sum_{s=1}^{t} \boldsymbol{x}_s \eta_s \right\|_{\boldsymbol{\Lambda}_t^{-1}} \leq 8\sigma \sqrt{d \log \left( 1 + \frac{tL^2}{\lambda d} \right) \cdot \log \left( \frac{4t^2}{\delta} \right)} + 4R \log \left( \frac{4t^2}{\delta} \right).$$

**Lemma I.3.** [27, Lemma H.5] Let $\phi : \mathcal{S} \times \mathcal{A} \to \mathbb{R}^d$ satisfying $\|\phi(x, a)\| \leq C$ for all $(x, a) \in \mathcal{S} \times \mathcal{A}$. For any $K > 0$ and $\lambda > 0$, define $\overline{\mathbb{G}}_K = \sum_{k=1}^{K} \phi(x_k, a_k)\phi(x_k, a_k)^{\top} + \lambda \mathbf{I}_d$ where $(x_k, a_k)$ 's are i.i.d. samples from some distribution $\nu$ over $\mathcal{S} \times \mathcal{A}$. Let $\mathbb{G} = \mathbb{E}_v[\phi(x, a)\phi(x, a)^{\top}]$. Then, for any $\delta \in (0, 1)$, if $K$ satisfies that

$$K \geq \max \left\{ 512 C^4 \|\mathbb{G}^{-1}\|^2 \log \left( \frac{2d}{\delta} \right), 4\lambda \|\mathbb{G}^{-1}\| \right\},$$

then with probability at least $1 - \delta$, it holds simultaneously for all $u \in \mathbb{R}^d$ that

$$\|u\|_{\overline{\mathbb{G}}_K^{-1}} \leq \frac{2}{\sqrt{K}} \|u\|_{\mathbb{G}^{-1}}.$$

