# OpenReview forum: "Minimax Optimal and Computationally Efficient Algorithms for Distributionally Robust Offline Reinforcement Learning"
_NeurIPS.cc/2024/Conference — NeurIPS 2024 poster_

### Official Review · Reviewer_d55R · 2024-06-17

**Soundness:** 3
**Presentation:** 2
**Contribution:** 3
**Rating:** 6
**Confidence:** 3

**Summary:**

This paper studies robust offline RL with function approximation under the specific setting of $d$-rectangular linear DRMDPs, where the nominal environment is a linear MDP with simplex feature space. The authors propose two learning algorithms and establish instance-dependent upper bounds for the suboptimality. The derived information-theoretic lower bound shows that the proposed VA-DRPVI algorithm is near-optimal up to a factor of $\widetilde{O}(\sqrt{d})$.

**Strengths:**

(+) The paper derives an information-theoretic lower bound, which depends on the novel uncertainty function $\Phi((\Sigma_h^*)^{-1},s)$ and could be of independent interest to the community.

(+) The suboptimality for the proposed VA-DRPVI algorithm nearly matches the lower bound.

**Weaknesses:**

(-) The derived suboptimality bounds (in Theorems 3.4 and 4.2) and the lower bound (in Theorem 5.1) require the number $K$ of the trajectories in the offline dataset to scale with $\text{poly}(d,H)$, which could be quite restrictive in the offline setting.

**Questions:**

1. Is it possible to establish suboptimality bounds and the lower bound for universal $K$? I guess there would be some "burn-in" terms.
2. Ma et al. (2023) consider a similar setting, except that the model uncertainty is measured by the KL divergence instead of the TV divergence in this paper. Can you point out the differences between this work and theirs? Also, are there any specific reasons for considering the TV divergence, and what are the unique challenges in the analysis?
3. The paper considers linear MDPs with simplex feature space. Can you explain the challenges in extending the results to general linear MDPs?
4. A minor suggestion on wording: In part 3 of the contribution section (the last paragraph on p.g. 2), the paper claims that "...which implies that VA-DRPVI is minimax optimal in the sense of information theory". However, as parameter $\beta_2$ scales with $\widetilde{O}(\sqrt{d})$, it can be better to switch to the word *near-optimal* instead of *minimax optimal*.

**Limitations:**

Limitations have been addressed in the paper.

---

> ### Author Rebuttal · Authors · 2024-08-05
>
> We thank the reviewer for your positive feedback on our work. We hope our response fully addresses all your questions.
>
> ---
> ### 1. Requirement of the number of trajectories $K$ to be poly(d,H).
>
> This is an interesting question. We acknowledge that our current analysis requires the sample size $K$ has a large order dependence on $d$ and $H$. Similar requirements are also necessary for standard offline RL with linear function approximation (Yin et al., 2022, Xiong et al. 2022). It is still unsure if we could construct upper and lower bounds for full-range $K$ in the robust linear function approximation setting. We will leave this problem for future study.
>
> ---
> ### 2. Comparison with Ma et al. (2022).
>
> We acknowledge that Ma et al. (2022)’s work is the most closely related to ours. However, we note that there are several technical flaws in their paper. We summarize the main issues here:
>
> * The proofs of main lemmas (Lemma D.1 and Lemma D.2 in the arXiv version) related to suboptimality decomposition and the proof of theorems are incorrect.
>
> * The sample complexity results for the two algorithms are both incorrect on the order of $d$.
>
> * The Assumption 4.4 on the dual variable of the dual formulation of the KL-divergence is too strong to be realistic.
>
> * The concept of mis-specification in Section 5.2 is not well-defined, which makes all results in Section 5 vacuous.
>
> Given these technique flaws, we believe the fundamental challenges of $d$-rectangular linear DRMDPs are not properly addressed in their work. It is also unsure whether $d$-rectangular linear DRMDPs with KL-divergence uncertainty set are solvable by their current methodologies. Moreover, in our work, we start with the setting of $d$-rectangular linear DRMDPs with TV-divergence uncertainty set. We address the essential challenges and explore the  fundamental limit of $d$-rectangular linear DRMDPs.
>
> ---
> ### 3. Reasons for considering TV-divergence and challenges in the analysis.
>
> There are two main reasons we consider the TV-divergence: 1) TV-divergence in a commonly studied distance in the field of DRMDPs, it serves as a proper “start” setting to address the fundamental challenges of DRMDPs with linear function approximation; 2) We also tried the KL-divergence uncertainty set, however the analysis heavily relies on an unrealistic assumption on the dual variable of the duality for KL. Thus, it requires further effort to figure out if the linear DRMDP with KL-divergence uncertainty set is both theoretically and practically solvable.
>
> Thanks to the simple format of the duality for TV, we can simply focus on the essential challenges of the linear DRMDP setting. To see the main challenge of the analysis, we first note that existing analysis of standard linear MDPs highly relies on the linear dependency of the Bellman equation on the nominal kernel. However, the consideration of model uncertainty of DRMDPs disrupts the linear dependency (see (2.1a)) of the robust Bellman equation on the nominal kernel, where the offline dataset is collected.
> This nonlinearity leads to unique challenges in both upper bound and lower bound analysis, and we are the first to solve those challenges.
>
> ---
> ### 4. Extending the results to general linear MDPs.
>
> Generalizing the results to standard linear MDPs presents challenges primarily due to the construction of  uncertainty sets. In particular, the design of the $d$-rectangular uncertainty set relies on the fact that the factor uncertainty sets are defined around probability distributions, since the total variation distance $D_{\text{TV}}(\cdot||\cdot)$ is a distance measure for probability distributions. Further, it is non-trivial to define uncertainty sets for the standard linear MDP to ensure (1) it contains valid probability distributions and (2) the optimization among the uncertainty set can be efficiently solved (by duality for example). We leave this extension for future research.
>
> ---
> ### 5. Wording suggestion.
> Thanks, we will take your advice and revise the “minimax optimal” to “near-optimal”.
>
> ---
> We hope we have addressed all of your questions/concerns. If you have any further questions, we would be more than happy to answer them and if you don’t, would you kindly consider increasing your score?
>
> ---
> **References**
>
> [1] Yin, Ming, Yaqi Duan, Mengdi Wang, and Yu-Xiang Wang. "Near-optimal Offline Reinforcement Learning with Linear Representation: Leveraging Variance Information with Pessimism." In International Conference on Learning Representation. 2022.
>
> [2] Xiong, Wei, Han Zhong, Chengshuai Shi, Cong Shen, Liwei Wang, and Tong Zhang. Nearly Minimax Optimal Offline Reinforcement Learning with Linear Function Approximation: Single-Agent MDP and Markov Game. In International Conference on Learning Representations (ICLR). 2023.

---

> > ### Comment · Reviewer_d55R · 2024-08-12
> > **Thank you**
> >
> > Thank you for your detailed responses. I tend to maintain my score. Responding:
> >
> > After reading references [1] and [2], I can see that the lower bound on $K$ results from the concentration properties of variance estimators. Hence, it seems quite confusing that a suboptimality upper bound for uniform $K$ cannot be derived for Algorithm 1 as it does not involve such estimators.

---

> > > ### Author Response · Authors · 2024-08-12
> > >
> > > Thank you for your reply and for acknowledging our responses. We answer your further questions as follows.
> > >
> > > ---
> > > ### Suboptimality upper bound for uniform $K$.
> > >
> > > We would like to acknowledge that your intuition is correct. For Algorithm 1, if we adopt the Suboptimality Decomposition Lemma C.1 proposed in our current paper, as well as the uniform concentration argument for analyzing the estimation error developed by Jin et al. (2021), we can get an instance-dependent suboptimality upper bound for uniform $K$ for Algorithm 1 without Assumption 3.3. However, this would lead to an additional $O(\sqrt{d})$ in the order of the upper bound of Algorithm 1.
> > >
> > > More importantly, the focus of our work is to explore the fundamental limit and intrinsic characteristics of the offline d-rectangular linear DRMDP. We proposed Algorithm 1 to use it as the backbone of the more advanced Algorithm 2, which could achieve the nearly minimax optimal suboptimality bound. Therefore, we analyze Algorithm 1 using the method presented in our paper so that it serves as a warm-start for us to work out the theoretical analysis pipeline needed to achieve our goal (near-optimal instance-dependent upper and lower bounds). Further, we would like to clarify that it is unsure if we can derive an upper bound for uniform $K$ for **Algorithm 2** due to the reason we state in the sequel.
> > >
> > > Note that Jin et al. (2021)’s analysis framework does not incorporate the reference-decomposition technique, and thus cannot be leveraged to achieve the near-optimal suboptimality in the $d$-rectangular linear DRMDP setting. Consequently, adapting Jin et al. (2021)’s analysis framework mentioned in the above paragraph for analyzing Algorithm 1 would be a distraction/detour from our main goal. Nevertheless, we will add a remark to discuss more on this in the next version of our manuscript.
> > >
> > > We would be more than happy to discuss more if you have further questions.
> > >
> > >
> > > **References**
> > >
> > > [1] Ying Jin, Zhuoran Yang, and Zhaoran Wang. Is pessimism provably efficient for offline RL? In International Conference on Machine Learning, pages 5084–5096. PMLR, 2021.

---

> > > > ### Comment · Reviewer_d55R · 2024-08-13
> > > > **Thank you**
> > > >
> > > > I appreciate the detailed explanation from the authors. I have no further questions.

---

### Official Review · Reviewer_nZV5 · 2024-07-11

**Soundness:** 2
**Presentation:** 2
**Contribution:** 2
**Rating:** 5
**Confidence:** 4

**Summary:**

This paper proposes minimax optimal and computationally efficient algorithms with linear function approximation in the context of distributionally robust offline RL. The authors incorporate multiple new techniques in theoretical analysis, e.g., variance information, suboptimality and estimation uncertainty decomposition, robust value function shrinkage, and family of hard instances.

**Strengths:**

1. The paper is technically sound with most claims supported sufficiently.
2. The theoretical analysis seems novel.

**Weaknesses:**

Quality:
1. In Line 175, the expression requires calculating the minimum value of the entire space. This operation is computationally intractable when considering continuous state space, which violates the "computationally efficient" claim made by authors.
2. In Line 202, I don't find Assumption 4.1 and Remark 4.2 in the paper. Additionally, the "fail-state" assumption doesn't necessarily hold in multiple MuJoCo environments, i.e., some environments have negative minimum values.
3. There are no experimental results provided in the paper. Although it is a theoretical paper, it is better to include some toy experiments to verify the proposed algorithm.

Clarity:
1. Since the authors only focus on linear function approximation, it is better to clarify it in the open questions (Lines 51-52).
2. (Lines 97-98) Assoud's method -> Assouad's method? Additionally, it is better to provide references about Assouad's method.
3. (Lines 201-203) It is better to provide references about the Nelder-Mead method.

Significance:
There are several works providing theoretical guarantees for general function approximation in online robust RL (e.g., [58]) and offline RL (e.g., [Chen and Jiang, 2019]). Some discussions are missing about why general function approximation is not considered here for offline robust RL settings.

[Chen and Jiang, 2019] Information-Theoretic Considerations in Batch Reinforcement Learning, ICML 2019.

**Questions:**

Please see the details in "weakness".

**Limitations:**

There is no potential negative social impact of this work.

---

> ### Author Rebuttal · Authors · 2024-08-05
>
> We thank the reviewer for your valuable time and effort in providing detailed feedback on our work. We hope our response will fully address all your questions.
>
> ---
> ### 1. Computational tractability.
>
> We would like to note that we discussed the computational tractability of our algorithm on Line 178 - Line 203. Specifically, if the ‘fail-state’ assumption holds, then the minimal value is trivially equal to zero hence no computation is needed. If the ‘fail-state’ does not hold, we can use some heuristic methods, such as the Nelder-Mead method, to search for the minimal value. Thus, our algorithms are in general computationally tractable. We have also conducted numerical experiments to show the computational tractability of our algorithm.
>
> In terms of computation efficiency, 1) our algorithms leverage the linear function approximation, and do not need to estimate the robust value at each state-action pair as those methods developed for the tabular DRMDP setting; 2) Moreover, thanks to the structure of linear DRMDPs, we do not need to solve the dual problem at each state-action pair. Thus, our algorithm is computationally efficient compared to those methods developed for tabular DRMDPs and general $(s, a)$-rectangular DRMDPs. This is consistent with the claim of computationally efficient algorithms in theoretical RL (Jin et al., 2019, Xiong et al. 2022, Yin et al. 2022), which stating the runtime and the sample complexity should not depend on the number of states, but should depend instead on an intrinsic complexity measure of the function class.
>
>
>
>
> ---
> ### 2. In Line 202, I don't find Assumption 4.1 and Remark 4.2 in the paper. Additionally, the "fail-state" assumption doesn't necessarily hold in multiple MuJoCo environments, i.e., some environments have negative minimum values.
>
> We believe there is a misunderstanding. Here we are discussing previous work’s assumption 4.1 and remark 4.2 (Liu and Xu, 2024:  https://proceedings.mlr.press/v238/liu24d/liu24d.pdf).
> We highlight that our work does **not** need the fail-state assumption. For environments that the fail-state does not hold, we can 1) either modify the reward of the environment to rescale the minimal value (of an absorbing state) to zero, which creates a fail-state, or 2) use heuristic methods to search for/approximate the minimal value.
>
> ---
> ### 3.  Numerical experiments
>
> Thanks for your suggestion. We have done some experiments to verify our proposed algorithm. Please see the overall response for more details.
>
> ---
> ### 4. Clarify the linear function approximation in the open question.
> We will revise the open question as follows: Is it possible to design a computationally efficient and minimax optimal algorithm for robust offline RL with linear function approximation?
>
> ---
> ### 5. Typos and references.
> Thanks for pointing out these issues. We have revised the typo and added the references.
>
> ---
> ### 6. Justification of why linear function approximation rather than the general function approximation is considered in this work.
> In this work, we focus on offline DRMDPs with linear function approximation under the setting of $d$-rectangular linear DRMDPs with TV-divergence uncertainty sets, which is an understudied area with unique fundamental challenges compared to standard offline RL. It would be an interesting future research direction to incorporate general function approximation techniques, such as those in Chen and Jiang (2019) into offline DRMDPs. We would add the following paragraph in the introduction section to motivate our setting:
>
> Although standard offline MDPs based on linear function approximation have exhibited theoretical success (Jin et al., 2021; Yin et al., 2022; Xiong et al., 2022), DRMDPs with linear function approximation remain understudied. In particular, DRMDP encounters unique difficulties when applying linear function approximations, even when the source domain transition kernel is linear. The dual formulation in worst-case analyses induces extra nonlinearity for the function approximation. Consequently, the theoretical understanding of offline DRMDPs with function approximation remains elusive, even when the approximation is linear. In this work, we aim to address the essential challenges and explore the fundamental limit of this setting.
>
> In the conclusion section, we will add the following sentences to explore future research directions:
>
> Moreover, we would like to extend the pessimism principle utilized in this work beyond linear function approximation to explore general function approximation in DRMDPs. Leveraging the techniques for general function approximation in standard offline RL (Chen and Jiang, 2021), we would explore the unique challenges and fundamental limits of practically solving DRMDPs with general function approximation.
>
> ---
> We hope we have addressed all of your questions/concerns. If you have any further questions, we would be more than happy to answer them and if you don’t, would you kindly consider increasing your score?
>
> ---
> **References**
>
> [1] Zhishuai Liu, and Pan Xu. "Distributionally robust off-dynamics reinforcement learning: Provable efficiency with linear function approximation." In International Conference on Artificial Intelligence and Statistics, pp. 2719-2727. PMLR, 2024.
>
> [2] Yin, Ming, Yaqi Duan, Mengdi Wang, and Yu-Xiang Wang. (2022). "Near-optimal Offline Reinforcement Learning with Linear Representation: Leveraging Variance Information with Pessimism."
>
> [4] Ying Jin, Zhuoran Yang, and Zhaoran Wang. (2021). Is pessimism provably efficient for offline RL?
>
> [4] Xiong, Wei, Han Zhong, Chengshuai Shi, Cong Shen, Liwei Wang, and Tong Zhang. (2022). Nearly Minimax Optimal Offline Reinforcement Learning with Linear Function Approximation: Single-Agent MDP and Markov Game.
>
> [5] Jin, Chi, Zhuoran Yang, Zhaoran Wang, and Michael I. Jordan. (2019). Provably efficient reinforcement learning with linear function approximation.

---

> > ### Comment · Reviewer_nZV5 · 2024-08-12
> >
> > Thank you for your clarification! I have increased my score, but I still have two more questions.
> > 1. As I mentioned, the "fail-state" assumption doesn't necessarily hold in multiple MuJoCo environments, while the Nelder–Mead technique is a heuristic search method that cannot guarantee convergence to global optimal. In these cases, I think the suboptimality gap cannot hold anymore as shown in Theorem 4.2.
> > 2. I admit that there are some challenges in the setting of d-rectangular linear DRMDPs. However, given existing online robust RL [58] and offline robust RL [26] have shown the theoretical guarantees under general function approximation, it is still not clear why authors don't consider general function approximation directly. In the authors' rebuttal, this part is somewhat dodged by highlighting the difficulty of d-rectangular linear DRMDPs.

---

> > > ### Author Response · Authors · 2024-08-12
> > >
> > > Thank you for your reply. We answer your further questions as follows.
> > >
> > > ---
> > > ### Q1.  About the "fail-state" assumption and Nelder–Mead technique.
> > >
> > > In theory, we would like to clarify that Theorem 4.2 does not rely on the “fail-state” assumption. The minimization over $s$ can be treated as an oracle, a common approach in theoretical RL analysis [1, 2, 3, 4]. Therefore, **our theoretical results are valid for any setting that satisfies the $d$-rectangular linear DRMDP structure**. Specifically, [1, 2, 3, 4] focus on standard linear MDPs and propose computationally efficient methods with the help of an oracle. For example, in the Bellman operator, the maximization over the action space $\mathcal{A}$, especially when continuous, is treated as an oracle. Similarly, in our work, assuming access to an oracle for minimization over the (potentially continuous) state space allows us to concentrate on the fundamental limits and challenges of $d$-rectangular linear DRMDPs with TV-divergence uncertainty sets.
> > >
> > > In practice, the minimization/maximization can be approximated. We acknowledge that in RL, a gap often exists between theoretical guarantees and actual performance in practice. However, our focus in this work is on providing a rigorous theoretical understanding of the offline $d$-rectangular DRMDP setting, which aligns with the approach of other theoretical research and meets NeurIPS standards.
> > >
> > > Furthermore, we plan to explore ways to address the issue of minimization over the state space in future work, which arises from the strong duality for TV distances (see Proposition G.1). As an alternative, we could consider replacing the TV-divergence uncertainty set with a Kullback-Leibler uncertainty set [6] or a $\chi^2$ uncertainty set [5], both leading to strong duality without the minimization term. We anticipate these approaches will lead to more practical algorithms for $d$-rectangular linear DRMDPs, though they are beyond the scope of this current paper.
> > >
> > > ---
> > > ### Q2. Why not consider general function approximation directly.
> > >
> > > We would like to further elaborate on this issue from the perspectives of uncertainty sets and algorithm design.
> > >
> > > **Panaganti et al. (2022) ([26] in our paper)**: [26] study offline $(s,a)$-rectangular DRMDPs with general function approximation. However, the $(s,a)$-rectangular uncertainty set can lead to overly conservative policies, particularly when the transition probabilities exhibit inherent structure [7,8]. To address this issue, our work focuses on structured uncertainty sets, specifically the $d$-rectangular linear DRMDP. This structured uncertainty set allows for a linear representation of robust value functions, resulting in less conservative policies compared to the general $(s,a)$-rectangular uncertainty set. This is why we do not follow [26] in studying general function approximation. In fact, our study of function approximation under structured uncertainty sets represents a significant contribution of our work, distinguishing it from [26].
> > >
> > > **Zhou et al. (2023) ([58] in our paper)**: [58] explores the Double-Sampling uncertainty set and the Integral Probability Metric uncertainty set. They focus on scenarios with a simulator, proposing a robust Natural Actor-Critic algorithm with finite-time convergence guarantees, supported by extensive numerical experiments to demonstrate robustness. However, their method cannot be easily extended to address the $d$-rectangular uncertainty set of linear DRMDPs due to fundamental differences in uncertainty set formulations and algorithm design (value iteration-based versus policy gradient-based). We acknowledge that the method in [58] and our work are orthogonal. This distinction highlights potential avenues for complementary research in the future.
> > >
> > > We will add the above discussion in our revision. If you have further questions, we are more than happy to discuss with you.
> > >
> > > ---
> > > **References**
> > >
> > > [1]  Jin et al. Provably efficient reinforcement learning with linear function approximation. In COLT 2020.
> > >
> > > [2] He et al. "Nearly minimax optimal reinforcement learning for linear markov decision processes." In ICML 2023.
> > >
> > > [3] Xiong et al. Nearly Minimax Optimal Offline Reinforcement Learning with Linear Function Approximation: Single-Agent MDP and Markov Game. In ICLR 2023.
> > >
> > > [4] Yin et al. "Near-optimal Offline Reinforcement Learning with Linear Representation: Leveraging Variance Information with Pessimism." In ICLR 2022.
> > >
> > > [5] Shi et al. The curious price of distributional robustness in reinforcement learning with a generative model. arXiv preprint, 2023.
> > >
> > > [6] Zhou et al. Finite-sample regret bound for distributionally robust offline tabular reinforcement learning. In AISTATS 2021.
> > >
> > > [7] Goyal et al. Robust markov decision processes: Beyond rectangularity. Mathematics of Operations Research, 48(1):203–226, 2023.
> > >
> > > [8] Ma et al. Distributionally robust offline reinforcement learning with linear function approximation. arXiv preprint, 2022.

---

> > > > ### Comment · Reviewer_nZV5 · 2024-08-13
> > > >
> > > > Thank you for your further clarification! I have increased one more point.

---

### Official Review · Reviewer_Qkxw · 2024-07-13

**Soundness:** 3
**Presentation:** 3
**Contribution:** 3
**Rating:** 6
**Confidence:** 3

**Summary:**

This paper considers the distributionally robust Markov decision process (or sometimes robust MDP in the literature). In particular, it considers the linear RMDP with $d$-rectangularity TV uncertainty set, which decouples the uncertainty set from the state-action pair. Two algorithms are proposed to solve this, with the second algorithm being an improvement to the first. Extensive analysis is provided.

**Strengths:**

1. The paper is well-written and easy-to-follow
2. Results are strong and thorough
3. Problem is well-motivated by introducing a hard instance (lower bound results)

**Weaknesses:**

1. Comparison with other existing works is a bit unclear

**Questions:**

1. Can authors provide more comparison with existing linear MDP with $d$-rectangularity? So besides the difference in the type of uncertainty set used, can you compare your work with [1] in details, especially the results and methodologies?
2. Continuing the first question, in particular, I noticed that your coverage assumption is stronger than [1]. It seems that you need the behavioral policy to cover states and actions that the optimal policy seldom visits as well. [1] states that with pessimism, they are able to achieve single-policy concentrability (w.r.t. only optimal policy). Could you share some insights?

[1] Xiaoteng Ma, Zhipeng Liang, Li Xia, Jiheng Zhang, Jose Blanchet, Mingwen Liu, Qianchuan Zhao, and Zhengyuan Zhou. Distributionally robust offline reinforcement learning with linear function approximation. arXiv preprint arXiv:2209.06620, 2022.

Minor things
1. Question marks on line 618

**Limitations:**

No other limitation I'd like to bring up

---

> ### Author Rebuttal · Authors · 2024-08-05
>
> We thank the reviewer for your positive feedback on our work. We hope our response will fully address all your questions.
>
> ---
> ### 1. Comparison with Ma et al. (2022).
>
> We acknowledge that Ma et al. (2022)’s work is most closely related to ours. However, we note that there are several technical flaws in their paper. We summarize the main issues here:
>
>   (i) The proofs of their main lemmas (Lemma D.1 and Lemma D.2 in the arXiv version) related to suboptimality decomposition and the proof of theorems are incorrect.
>
>   (ii) The sample complexity results for the two algorithms are both incorrect on the order of $d$.
>
>   (iii) Assumption 4.4 on the dual variable of the dual formulation of the KL-divergence is too strong to be realistic.
>
>   (iv) The concept of mis-specification in Section 5.2 is not well-defined, which makes results in Section 5 vacuous.
>
> Given these technique flaws, we believe the fundamental challenges of $d$-rectangular linear DRMDPs are not properly addressed in their work. It is also unsure whether $d$-rectangular linear DRMDPs with KL-divergence uncertainty set are solvable by their current methodologies.
> Moreover, in our work, we start with the setting of $d$-rectangular linear DRMDPs with TV-divergence uncertainty sets. We address the essential challenges and explore the fundamental limit of $d$-rectangular linear DRMDPs. In the next version of our manuscript, we plan to add a paragraph discussing the comparison with Ma et al. (2022) in detail and further elaborate on our contributions.
>
> ---
> ### 2. Stronger coverage assumption.
>
> First of all, Assumption 5.1 in Ma et al. (2022) is problematic, since the right hand side has an additional dependence on $d$, which makes the right hand side unreasonably large. The assumption 5.1 should be in this form: there exists some absolute constant $c>0$, such that for any $(i, h, s, P) \in [d]\times[H] \times \mathcal{S} \times \mathcal{U}^{\rho}(P^0)$, we have
> $$
> \Lambda_h \succeq \lambda I + K\cdot c \cdot \mathbb{E}^{\pi^{\star},P}\big[(\phi_i(s,a)\mathbf{1}_i)(\phi_i(s,a)\mathbf{1}_i)^\top |s_1=s \big],
> $$
> which is exactly the robust partial coverage assumption in Blanchet et al. (2023, Assumption 6.3).
> On the one hand, a closer examination of Assumption 3.3 in our paper reveals that it guarantees a weaker version of the robust partial coverage assumption: there exists some constant $c^{\dagger}>0$, such that for any $(i, h, s, P)\in[d]\times[H]\times \mathcal{S} \times \mathcal{U}^{\rho}(P^0)$, we have
> \begin{align}
>      \Lambda_h \succeq \lambda I + K\cdot c^{\dagger}/d \cdot \mathbb{E}^{\pi^{\star},P}\big[(\phi_i(s,a)\mathbf{1}_i)(\phi_i(s,a)\mathbf{1}_i)^\top |s_1=s \big].
> \end{align}
> Nevertheless, Assumption 3.3 does not directly imply the robust partial coverage assumption. On the other hand, assumption 3.3 is essential in achieving the minimax optimality of our algorithm. A similar phenomenon also appears for the non-robust offline RL with linear function approximation. Specifically, leveraging the pessimism principle, Jin et al. (2021) show that the partial coverage assumption is enough for the provable efficiency of offline RL with linear function approximation. However, to achieve minimax optimality, Yin et al. (2022) and Xiong et al. (2022) show that a stronger full-type coverage is needed.  Lastly, in a concurrent work (Wang et al, 2024) of ours, the same assumption is also required to achieve tighter dependences on $d$ and $H$ (see Assumption 4 and Theorem 2 of their paper).
>
> ---
> ### 3. Question marks on line 618.
>
> Thanks for pointing out this typo. We have fixed it in our manuscript.
>
>
> ---
> We hope we have addressed all of your questions/concerns. If you have any further questions, we would be more than happy to answer them and if you don’t, would you kindly consider increasing your score?
>
> ---
>
> **References**
>
> [1] Ying Jin, Zhuoran Yang, and Zhaoran Wang. Is pessimism provably efficient for offline rl? In International Conference on Machine Learning, pages 5084–5096. PMLR, 2021.
>
> [2] Xiong, Wei, Han Zhong, Chengshuai Shi, Cong Shen, Liwei Wang, and Tong Zhang. Nearly Minimax Optimal Offline Reinforcement Learning with Linear Function Approximation: Single-Agent MDP and Markov Game. In International Conference on Learning Representations (ICLR). 2023.
>
> [3] Blanchet, Jose, Miao Lu, Tong Zhang, and Han Zhong. Double Pessimism is Provably Efficient for Distributionally Robust Offline Reinforcement Learning: Generic Algorithm and Robust Partial Coverage. In Thirty-seventh Conference on Neural Information Processing Systems. 2023.
>
> [4] Wang, He, Laixi Shi, and Yuejie Chi. Sample complexity of offline distributionally robust linear Markov decision processes. arXiv preprint arXiv:2403.12946 (2024).

---

> > ### Comment · Reviewer_Qkxw · 2024-08-12
> >
> > Dear authors,
> >
> > Thank you for responding to my questions. I am satisfied with your answer. I decided to increase my rating. Good luck!

---

> > > ### Author Response · Authors · 2024-08-12
> > >
> > > Thank you very much for your positive feedback! We will revise our paper according to your constructive reviews.
> > >
> > > Best,
> > >
> > > Authors

---

### Official Review · Reviewer_mtm1 · 2024-07-13

**Soundness:** 3
**Presentation:** 3
**Contribution:** 3
**Rating:** 6
**Confidence:** 5

**Summary:**

This paper presents a theoretical study of distributionally robust MDPs. Their findings show that function approximation is both different and harder in robust RL as compared to offline RL. They show matching information theoretic lower bounds for their novel algorithm.

**Strengths:**

1. The paper has positioned itself and it's improvement over past work really well (Table 1).
2. The paper's result about the information theoretic lower bound has a lot of theoretical value.
3. The appendix is well written and the proofs are sound.

**Weaknesses:**

1. More motivation on practical scenarios where DRMDPs are applicable will be useful.
2. Small simulation examples would further strengthen the paper to ensure that the constants in the sample complexities are reasonable.

**Questions:**

There is also literature on theoretical multi-task RL, which while not focussed on the distributional robustness aspect, still tackle the problem of planning in an environment with uncertain dynamics. For example see : https://arxiv.org/pdf/2402.12570 and follow up works. I'd like the authors to cite and write the differences to Theorem 1 of the referenced paper, which use function approximation for the representation as well, and come up with uncertainty gaps for different state action pairs of the transition model, and still give a suboptimality gap on the policy planning in the downstream target task (see Theorem 2).

**Limitations:**

There are no significant technical limitations of this paper.

---

> ### Author Rebuttal · Authors · 2024-08-05
>
> We thank the reviewer for your positive feedback on our work. We hope our response fully addresses all of your questions.
>
> ---
> ### 1. More motivation on practical scenarios where DRMDPs are applicable.
>
> In many real-world applications, the agent only has access to a **single** source domain, which can be different from target domains where the policy is deployed, assuming the same task is being performed. Importantly the target domains are **unknown** and can vary during deployment. Given the limited information and large uncertainty, one may hold a pessimistic perspective hoping the learned policy is robust enough that would perform well even under the worst-case transition. This basic idea leads to the DRMDP framework, which models the unknown target domain in an uncertainty set defined around the source domain.
>
> We take the control of infectious diseases for an example. Suppose we have access to a simulator (source domain), which emulates one specific real-world disease infection process (target domain). However, the simulator does not have perfect knowledge about the complex real-world environment, but it is reasonable to assume that the real-world environment is close to the simulator, thus lying in an uncertainty set around it. Notably, the real-world environment can be any environment in the uncertainty set, and the DRMDP framework provides a worst-case guarantee for policies learned merely through the simulator.
>
> ---
> ### 2. Sample simulation examples.
> Thanks for your suggestion. We have done some experiments to verify our proposed algorithm. Please see the overall response for more details.
>
> ---
> ### 3. Literature on theoretical multi-task RL.
>
> Thank you for bringing this work to our attention. We will add the following paragraphs to the related work section.
>
> Besides the distributionally robust perspective to solve the planning problem in a nearly unknown target environment, another line of work focuses on transfer learning in low-rank MDPs (Cheng et al., 2022; Lu et al., 2022; Agarwal et al., 2023; Bose et al., 2024). Specifically, the problem setup assumes that the agent has access to information of several source tasks. The agent learns a common representation from the source domains and then leverages the learned representation to learn a policy performing well in the target tasks with limited information. This setting is in stark contrast to the setting of DRMDP, where the agent only has access to the information of a single source domain, without any available information of the target domain, assuming the same task is being performed. This motivates the pessimistic attitude of the distributionally robust perspective.
>
> Among the aforementioned works, Bose et al. (2024) studied the offline multi-task RL, which is the most closely related to our setting. In particular, they investigate the representation transfer error in their Theorem 1, stating that the learned representation can lead to a transition kernel that is close to the target kernel in terms of the TV divergence. Note that the uncertainty is induced by the representation estimation error, which is different from our setting assuming that the uncertainty comes from perturbations on underlying factor distributions. Nevertheless, this work provides evidence that TV divergence is a reasonable measure to quantify the uncertainty in transition kernels and motivates a future research direction in learning robust policies that are robust to the uncertainty induced by the representation estimation error.
>
>
> ---
> We hope we have addressed all of your questions/concerns. If you have any further questions, we would be more than happy to answer them and if you don’t, would you kindly consider increasing your score?
>
> ---
>
> **References**
>
> [1] Rui Lu, Andrew Zhao, Simon S Du, and Gao Huang. Provable general function class representation learning in multitask bandits and mdp. Advances in Neural Information Processing Systems, 35:11507–11519, 2022.
>
> [2] Yuan Cheng, Songtao Feng, Jing Yang, Hong Zhang, and Yingbin Liang. Provable benefit of multitask representation learning in reinforcement learning. Advances in Neural Information Processing Systems, 35: 31741–31754, 2022.
>
> [3] Alekh Agarwal, Yuda Song, Wen Sun, Kaiwen Wang, Mengdi Wang, and Xuezhou Zhang. Provable benefits of representational transfer in reinforcement learning. In The Thirty Sixth Annual Conference on Learning Theory, pages 2114–2187. PMLR, 2023.
>
> [4] Bose, Avinandan, Simon Shaolei Du, and Maryam Fazel. "Offline multi-task transfer RL with representational penalization." arXiv preprint arXiv:2402.12570 (2024).
>
> [5] Zhishuai Liu, and Pan Xu. "Distributionally robust off-dynamics reinforcement learning: Provable efficiency with linear function approximation." In International Conference on Artificial Intelligence and Statistics, pp. 2719-2727. PMLR, 2024.
>
> [6] Ying Jin, Zhuoran Yang, and Zhaoran Wang. Is pessimism provably efficient for offline rl? In International Conference on Machine Learning, pages 5084–5096. PMLR, 2021.

---

> > ### Comment · Reviewer_mtm1 · 2024-08-12
> >
> > I thank the authors for their detailed response. I request to maintain my positive rating.

---

### Author Rebuttal · Authors · 2024-08-05

## Overall Response

We would like to thank all reviewers for their insightful and detailed reviews and comments. We have addressed the comments from the reviewers and revised the manuscript accordingly. In the following, we would like to provide overall responses to several common questions raised by reviewers.

### 1. Experimentation

Per reviewers’ requirements, we conduct numerical experiments to illustrate the performances of our two algorithms. In the attached PDF, we present additional experimental results. Specifically, we leverage the simulated linear MDP (Figure 1) proposed by Liu and Xu (2024) and adapt it to the offline setting. In particular, we set the behavior policy as the random policy that chooses actions uniformly at random at each state-action pair $(s, a)$ and stage $h$. The number of trajectories (i.e., sample size) of the offline dataset is set to 100. We compare our algorithms with their non-robust counterpart, PEVI, proposed by Jin et al. (2021). Figure 2 shows the performances of the learned policies of three algorithms. We conclude that both of our proposed algorithms are robust to environmental perturbation compared to the non-robust PEVI. Furthermore, VA_DRPVI slightly outperforms DRPVI in most settings. These numerical results are consistent with our theoretical findings.


### 2. Comparison with Ma et al. (2022)

We acknowledge that the work of Ma et al. (2022) is the most closely related to ours. However, we note that there are several technical flaws in their paper. We summarize the main issues here:

* The proofs of main lemmas (Lemma D.1 and Lemma D.2 in the arXiv version) related to suboptimality decomposition and the proof of theorems are incorrect.

* The sample complexity results for the two algorithms are both incorrect on the order of $d$.

* Assumption 4.4 on the dual variable of the dual formulation of the KL-divergence is too strong to be realistic.

* The concept of mis-specification in Section 5.2 is not well-defined, which makes all results in Section 5 vacuous.

Given these technique flaws, we believe the fundamental challenges of $d$-rectangular linear DRMDPs are not properly addressed in their work. It is also unsure whether $d$-rectangular linear DRMDPs with KL-divergence uncertainty set are solvable by their current methodologies. Moreover, in our work, we start with the setting of $d$-rectangular linear DRMDPs with TV-divergence uncertainty sets. We address the essential challenges and explore the fundamental limit of this setting.

---
**References**

[1] Liu, Zhishuai, and Pan Xu. "Distributionally robust off-dynamics reinforcement learning: Provable efficiency with linear function approximation." In International Conference on Artificial Intelligence and Statistics, pp. 2719-2727. PMLR, 2024.

[2] Ying Jin, Zhuoran Yang, and Zhaoran Wang. Is pessimism provably efficient for offline RL? In International Conference on Machine Learning, pages 5084–5096. PMLR, 2021.

[3] Ma, Xiaoteng, Zhipeng Liang, Jose Blanchet, Mingwen Liu, Li Xia, Jiheng Zhang, Qianchuan Zhao, and Zhengyuan Zhou. "Distributionally robust offline reinforcement learning with linear function approximation." arXiv preprint arXiv:2209.06620 (2022).

---

### Decision · Program_Chairs · 2024-09-25

**Decision:**

Accept (poster)

**Comment:**

The paper provides instance dependent upper and lower bounds under the setting of offline linear DRMDPs. The reviewers all agree with the contribution of the paper. I recommend acceptance.